# Unlocking the Power of Rehearsal in Continual Learning: A Theoretical Perspective

**Junze Deng** [1]  **Qinhang Wu** [2]  **Peizhong Ju** [3]  **Sen Lin** [4]  **Yingbin Liang** [1]  **Ness Shroff** [1 2]

## Abstract

Rehearsal-based methods have shown superior performance in addressing catastrophic forgetting in continual learning (CL) by storing and training on a subset of past data alongside new data in current task. While such a concurrent rehearsal strategy is widely used, it remains unclear if this approach is always optimal. Inspired by human learning, where sequentially revisiting tasks helps mitigate forgetting, we explore whether sequential rehearsal can offer greater benefits for CL compared to standard concurrent rehearsal. To address this question, we conduct a theoretical analysis of rehearsal-based CL in overparameterized linear models, comparing two strategies: 1) Concurrent Rehearsal, where past and new data are trained together, and 2) Sequential Rehearsal, where new data is trained first, followed by revisiting past data sequentially. By explicitly characterizing forgetting and generalization error, we show that sequential rehearsal performs better when tasks are less similar. These insights further motivate a novel Hybrid Rehearsal method, which trains similar tasks concurrently and revisits dissimilar tasks sequentially. We characterize its forgetting and generalization performance, and our experiments with deep neural networks further confirm that the hybrid approach outperforms standard concurrent rehearsal. This work provides the first comprehensive theoretical analysis of rehearsal-based CL.

## 1. Introduction

Continual learning (CL) (Parisi et al., 2019) seeks to build an agent that can learn a sequence of tasks continuously without access to old task data, resembling human's capability of lifelong learning. One of the major challenges therein is the so-called *catastrophic forgetting* (Kirkpatrick et al., 2017), i.e., the agent can easily forget the knowledge of old tasks when learning new tasks. A large amount of studies have been proposed to address this issue, among which *rehearsal-based* approaches (Rolnick et al., 2019) have demonstrated the state-of-the-art performance. The main idea behind is to store a subset of old task data in the memory and revisit them for new task learning, where a widely adopted strategy for training is **concurrent rehearsal** (Goldfarb et al., 2024), i.e., train the model *concurrently* on new data and past data.

While the concurrent rehearsal strategy seems very natural and has shown successful performance to address catastrophic forgetting, it is indeed questionable whether this strategy is always the right way for rehearsal in CL as we consider the following aspects. 1) *From the perspective of human learning.* In daily life, a common strategy to prevent forgetting is to review old knowledge. For example, suppose a student needs to learn a series of topics over a semester before taking an exam, and each topic corresponds to one task in CL. Intuitively, if these topics are highly related to each other, incorporating the knowledge of old topics into learning a new topic can be an effective strategy to strengthen the new learning and simultaneously reduce the forgetting of old knowledge, which is analogous to *concurrent rehearsal*. However, if the topics are very different from each other, a common practice is to learn new topics first and then go over old topics to mitigate forgetting. Here, such a *sequential rehearsal* may lead to better outcome in the exam. 2) *From the perspective of multi-task learning.* Learning multiple tasks all at once may lead to poor learning performance due to the potential interference among gradients of different tasks (Yu et al., 2020), whereas standard CL without regularization and rehearsal may even achieve less forgetting for more dissimilar tasks (Lin et al., 2023). Thus motivated, an interesting and open question to ask is:

*Question: Whether sequential rehearsal will serve as an appealing rehearsal strategy to complement the standard*

---

[1]Department of ECE, Ohio State University, Columbus, USA [2]Department of CSE, Ohio State University, Columbus, USA [3]Department of CS, University of Kentucky, Lexington, USA [4]Department of CS, University of Houston, Houston, USA. Correspondence to: Junze Deng <deng.942@osu.edu>.

*Proceedings of the 42$^{nd}$ International Conference on Machine Learning*, Vancouver, Canada. PMLR 267, 2025. Copyright 2025 by the author(s).

*concurrent rehearsal, and when will it be advantageous over concurrent rehearsal for CL?*

To answer this question from a theoretical perspective, we study rehearsal-based CL through the lens of overparameterized linear models to gain useful insights, by following a recent series of theoretical studies in CL (Lin et al., 2023; Evron et al., 2022; Ding et al., 2024; Li et al., 2024). However, none of those previous studies analyzed the rehearsal-based methods. The only theoretical work that studied the rehearsal-based methods is the recent concurrent work (Banayeeanzade et al., 2024). But this work considered only the standard *concurrent* rehearsal method, not from the new perspective of *sequential* rehearsal.

To capture the idea of sequential rehearsal, we propose a novel rehearsal strategy, in which the agent *sequentially* revisits each old task and trains the model with the corresponding past data after the current task is well learned.

We summarize our main contributions as follows.

(1) First of all, we provide the *first* explicit closed-form expressions for the expected value of forgetting and generalization error for both concurrent rehearsal strategy and sequential rehearsal strategy under an overparameterized linear regression setting. Note that the blending of samples from old tasks in concurrent rehearsal introduces significant intricacies related to task correlation in theoretical analysis. To address this challenge, we partition training data into blocks based on different tasks, which enables us to further calculate the task interference using the properties of block matrix. In particular, our theoretical results demonstrate how the performance of rehearsal-based CL is affected by various factors, including task similarity and memory size.

(2) Secondly, we propose a novel rehearsal strategy, i.e., *sequential rehearsal*, to sequentially revisit old tasks after the current task is fully learned. By characterizing the explicit closed-form expressions for the expected forgetting and generalization error for sequential rehearsal and comparing with the concurrent rehearsal, we give an affirmative answer to the open question above. More importantly, we rigorously characterize the conditions when sequential rehearsal can benefit CL more than concurrent rehearsal, in terms of both forgetting and generalization error, which is also consistent with our motivations above: Sequential rehearsal outperforms concurrent rehearsal if tasks in CL are dissimilar, and the performance improvement is larger when the tasks are more dissimilar. Numerical simulations on linear models further corroborate our theoretical results.

(3) Last but not least, our theoretical insights can indeed go beyond the linear models and guide the practical algorithm design for rehearsal-based CL with deep neural networks (DNNs). More specifically, we merge the idea of sequential rehearsal into standard rehearsal-based CL with concurrent rehearsal, leading to a novel **hybrid rehearsal** approach where 1) old tasks dissimilar to the current task will be revisited by using sequential rehearsal (guided by our theory that suggests more benefit if dissimilar tasks are revisited sequentially) and 2) the past data for the remaining old tasks (that are sufficiently similar to the current task) will still be used concurrently with current task data. Our experiments on real datasets with DNNs verify that our hybrid approach can perform better than concurrent rehearsal and the advantage is more apparent when tasks are more dissimilar.

## 2. Related Work

**Empirical studies in CL.** CL has drawn significant attention in recent years, with numerous empirical approaches developed to mitigate the issue of catastrophic forgetting. Architecture-based approaches combat catastrophic forgetting by dynamically adjusting network parameters (Rusu et al., 2016) or introducing architectural adaptations such as an ensemble of experts (Rypeść et al., 2024). Regularization-based methods constrain model parameter updates to preserve the knowledge of previous tasks (Kirkpatrick et al., 2017; Magistri et al., 2024). Memory-based methods address forgetting by storing information of old tasks in the memory and leveraging the information during current task learning, which can be further divided into orthogonal projection based methods and rehearsal-based methods. The former stores gradient information of old tasks to modify the optimization space for the current task (Saha et al., 2021; Lin et al., 2022b), while the latter stores and reuses a tiny subset of representative data, known as exemplars. Exemplar sampling methods involve reservoir sampling (Chrysakis & Moens, 2020) and an information-theoretic evaluation of exemplar candidates (Sun et al., 2022). Other work such as Shin et al. (2017) retains past knowledge by replaying "pseudo-rehearsal" constructed from input data instead of storing raw input. Rehearsal methods mostly use a concurrent scheme that trains the model using a mix of input data and sampled exemplars (Chaudhry et al., 2018; Dokania et al., 2019; Rebuffi et al., 2017; Garg et al., 2024). Other exemplar utilization methods include Lopez-Paz & Ranzato (2017) and Chaudhry et al. (2018), which use exemplar to impose constraints in the gradient space.

**Theoretical studies in CL.** Compared to the vast amount of empirical studies in CL, the theoretical understanding of CL is very limited but has started to attract much attention very recently. Bennani & Sugiyama (2020); Doan et al. (2021) investigated CL performance for the orthogonal gradient descent approach in NTK models theoretically. Yin et al. (2020) focused on regularization-based methods and proposed a framework, which requires second-order information to approximate loss function. Cao et al. (2022); Li et al. (2022) characterized the benefits of continual represen-

tation learning from a theoretical perspective. Evron et al. (2023) connected regularization-based methods with Projection Onto Convex Sets. Recently, a series of theoretical studies proposed to leverage the tools of overparameterized linear models to facilitate better understanding of CL. Evron et al. (2022) studied the performance of forgetting under such a setup. After that, Lin et al. (2023) characterized the performance of CL, where they discuss the impact of task similarities and the task order. Ding et al. (2024) further characterized the impact of finite gradient descent steps on forgetting of CL. Goldfarb & Hand (2023) illustrated the joint effect of task similarity and overparameterization. Zhao et al. (2024) provided a statistical analysis of regularization-based methods. More recently, Li et al. (2024) theoretically investigated the impact of mixture-of-experts on the performance of CL in linear models.

Different from all these studies, we seek to fill up the theoretical understanding for rehearsal-based CL. Note that one concurrent study (Banayeeanzade et al., 2024) also investigates rehearsal-based CL in linear models with concurrent rehearsal. However, one key difference here is that we propose a novel rehearsal strategy, i.e., the sequential rehearsal, and theoretically show its benefit over concurrent rehearsal for dissimilar tasks. Our theoretical results further motivate a new algorithm design for CL in practice, which demonstrates promising performance on DNNs.

## 3. Problem setting

We consider a common CL setup consisting of $T$ tasks where each task arrives sequentially in time $t \in [T]$. Here $[T] := \{1, 2, ..., T\}$ for any positive integer $T$. Let $\boldsymbol{I}_p$ denote the $p \times p$ identity matrix and let $\|\cdot\|$ denote the $\ell_2$-norm.

**Data Model.** We adopt the setting of linear ground truth which is commonly used in recent theoretical analysis of CL, e.g., (Lin et al., 2023; Li et al., 2024; Banayeeanzade et al., 2024). Specifically, for each task $t \in [T]$, a sample $(\boldsymbol{x}_t, y_t)$ is generated by a linear ground truth model:

$$y_t = \boldsymbol{x}_t^\top \boldsymbol{w}_t^* + z_t, \tag{1}$$

where $\boldsymbol{x}_t \in \mathbb{R}^p$ denotes features, $y_t \in \mathbb{R}$ denotes the output, $\boldsymbol{w}_t^* \in \mathbb{R}^p$ denotes the ground truth parameters, and $z_t \in \mathbb{R}$ denotes the noise.

**Dataset.** For each task $t \in [T]$, there are $n_t$ training samples $(\boldsymbol{x}_{t,i}, y_{t,i})_{i \in [n_t]}$. We stack those samples into matrices/vectors to obtain the dataset $\mathcal{D}_t = \{(\boldsymbol{X}_t, \boldsymbol{Y}_t) \in \mathbb{R}^{p \times n_t} \times \mathbb{R}^{n_t}\}$. By Equation (1), we have

$$\boldsymbol{Y}_t = \boldsymbol{X}_t^\top \boldsymbol{w}_t^* + \boldsymbol{z}_t, \tag{2}$$

where $\boldsymbol{X}_t := [\boldsymbol{x}_{t,1} \ \boldsymbol{x}_{t,2} \ \cdots \ \boldsymbol{x}_{t,n_t}]$, $\boldsymbol{Y}_t := [y_{t,1} \ y_{t,2} \ \cdots \ y_{t,n_t}]^\top$, and $\boldsymbol{z}_t := [z_{t,1} \ z_{t,2} \ \cdots \ z_{t,n_t}]^\top$. We

consider *i.i.d.* Gaussian features and noise, i.e., each element of $\boldsymbol{X}_t$ follows *i.i.d.* standard Gaussian distribution, and $\boldsymbol{z}_t \sim \mathcal{N}(0, \sigma_t^2 \boldsymbol{I}_{n_t})$ where $\sigma_t \geq 0$ denotes the noise level. To make our result easier to interpret, we let $\sigma_t = \sigma$ and $n_t = n$ for all $t \in [T]$.

**Memory.** For any task $t \geq 2$, besides $\mathcal{D}_t$, the agent has an overall memory dataset $\mathcal{M}_t$ that contains separate memory datasets $\mathcal{M}_{t,h}$ for each of the previous tasks $h \in [t-1]$, i.e., $\mathcal{M}_t = \bigcup_{h=1}^{t-1} \mathcal{M}_{t,h}$ where $\mathcal{M}_{t,h} = (\widetilde{\boldsymbol{X}}_{t,h}, \widetilde{\boldsymbol{Y}}_{t,h}) \in \mathbb{R}^{p \times M_{t,h}} \times \mathbb{R}^{M_{t,h}}$ denotes the samples from previous task $h$ and we define $M_{t,h}$ as the number of samples in $\mathcal{M}_{t,h}$. In most CL applications, the memory space is fully utilized and the memory size does not change over time. We denote this memory size by $M$ that does not change with $t$. In this case, we have $\sum_{h=1}^{t-1} M_{t,h} = M$ for any $t \geq 2$. To simplify our theoretical analysis, we focus on the situation in which the memory data are all fresh and have not been used in previous training. We equally allocate the memory to all previous tasks at each time $t$, i.e., $M_{t,h} = \frac{M}{t-1}$ for $h \in [t-1]$. For simplicity, we assume $\frac{M}{t-1}$ is an integer[1] for any $t \in \{2, 3, \cdots, T\}$.

**Performance metrics.** We first introduce the model error of parameter $\boldsymbol{w}$ over task $i$'s ground truth.

$$\mathcal{L}_i(\boldsymbol{w}) = \|\boldsymbol{w} - \boldsymbol{w}_i^*\|^2. \tag{3}$$

We note that this formulation is widely adopted in recent theoretical studies(Evron et al., 2022; Lin et al., 2023). The performance of CL is measured by two key metrics, which are forgetting and generalization error. Let $\boldsymbol{w}_t$ be the parameters of the training result at task $t$.

1. *Forgetting:* It measures the average forgetting of old tasks after learning the new task. In our setup, forgetting at task $T$ w.r.t. previous tasks $[T-1]$ is defined as follows.

$$F_T = \frac{1}{T-1} \sum_{i=1}^{T-1} \mathbb{E}[\mathcal{L}_i(\boldsymbol{w}_T) - \mathcal{L}_i(\boldsymbol{w}_i)]. \tag{4}$$

2. *Generalization error:* It measures the overall model generalization after the final task is learned. In our setup, generalization error is defined as follows.

$$G_T = \frac{1}{T} \sum_{i=1}^{T} \mathbb{E}[\mathcal{L}_i(\boldsymbol{w}_T)]. \tag{5}$$

The definitions are consistent with the standard CL performance measures in experimental studies (Saha et al., 2021).

---

[1] We note that without the assumption of $\frac{M}{t-1} \in \mathbb{Z}$, memory can still be allocated as equally as possible, resulting in only a minor error. Our theoretical results remain of referential significance.

# 4. A Novel Sequential Rehearsal vs. Popular Concurrent Rehearsal

In this section, we first introduce the popular *concurrent rehearsal* strategy that is widely used in current CL applications to mitigate catastrophic forgetting. We will then propose a novel *sequential rehearsal* strategy, which has appealing advantage compared to concurrent rehearsal.

To describe these rehearsal strategies, note that the training result $\boldsymbol{w}_t$ at task $t$ will be used as the initial point for the next task $t + 1$. The initial model parameter of task 1 is set to be $\boldsymbol{0}$, i.e., $\boldsymbol{w}_0 = \boldsymbol{0}$. The training loss for task $t$ is defined by mean-squared-error (MSE). We focus on the over-parameterized case, i.e., $p > n + M$. As shown in (Zhang et al., 2022; Gunasekar et al., 2018), the convergence point of stochastic gradient descent (SGD) for MSE is the feasible point closest to the initial point with respect to the $\ell_2$-norm, i.e., the minimum-norm solution.

**Concurrent rehearsal.** We first introduce the popular concurrent rehearsal strategy as follows. At each task $t \geq 2$, we apply SGD on the current dataset and the memory dataset jointly to update the model parameter. Specifically, as illustrated in Figure 1, at time $t$, we minimize the MSE loss via SGD on the combined dataset $\mathcal{D}_t \bigcup \mathcal{M}_t$ with the initial point $\boldsymbol{w}_{t-1}$ and obtain the convergent point $\boldsymbol{w}_t$ as

$$\boldsymbol{w}_t = \arg\min_{\boldsymbol{w}} \|\boldsymbol{w} - \boldsymbol{w}_{t-1}\|^2$$
$$s.t. \ \ \boldsymbol{X}_t^\top \boldsymbol{w} = \boldsymbol{Y}_t, \ \ \widetilde{\boldsymbol{X}}_{t,h}^\top \boldsymbol{w} = \widetilde{\boldsymbol{Y}}_{t,h}, \forall h \in [t-1].$$

**Novel sequential rehearsal.** In scenarios where previous tasks are very different from the current task, concurrent rehearsal may result in contradicting gradient update directions, and can hurt the knowledge transfer among tasks. Consequently, concurrent rehearsal may not always perform well. This motivates a novel rehearsal strategy that sequentially revisits history tasks one by one after training the current task, analogously to the way how a student reviews previously learned topics to avoid forgetting before exams.

To formally describe the training (see Figure 1 for an illustration), at each task $t \geq 2$, we first train on the current dataset $\mathcal{D}_t$ to learn the new task until the convergence to the initial stopping point $\boldsymbol{w}_t^{(0)}$. Then, for $h = 1, 2, ..., t-1$, we start from the previous stopping point $\boldsymbol{w}_t^{(h-1)}$ and train on the memory dataset $\mathcal{M}_{t,h}$ to converge to the next stopping point. Eventually, $\boldsymbol{w}_t$ is obtained after revisiting all memory sets, i.e., $\boldsymbol{w}_t = \boldsymbol{w}_t^{(t-1)}$. To simplify, we define $\widetilde{\boldsymbol{X}}_{t,0} := \boldsymbol{X}_t$, $\widetilde{\boldsymbol{Y}}_{t,0} := \boldsymbol{Y}_t$ and $\boldsymbol{w}_t^{(-1)} := \boldsymbol{w}_{t-1}$. Then, the training process is equivalent to solve the following optimization problems recursively for $h = 0, 1, ..., t-1$:

$$\boldsymbol{w}_t^{(h)} = \arg\min_{\boldsymbol{w}} \left\| \boldsymbol{w} - \boldsymbol{w}_t^{(h-1)} \right\|^2, \ \ s.t. \ \widetilde{\boldsymbol{X}}_{t,h}^\top \boldsymbol{w} = \widetilde{\boldsymbol{Y}}_{t,h}.$$

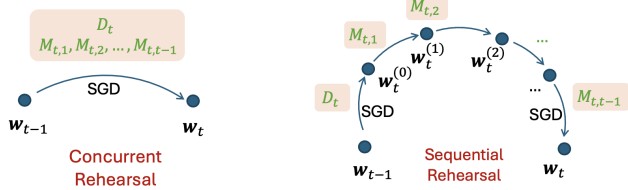

*Figure 1.* An illustration of concurrent and sequential rehearsal.

# 5. Main Results

The main theoretical results consist of two parts. First, we derive closed forms of forgetting and generalization error for both concurrent and sequential rehearsal methods. Second, we compare the performance of these two rehearsal-based schemes, concluding that sequential rehearsal outperforms concurrent rehearsal when tasks are dissimilar.

## 5.1. Characterization of Forgetting and Generalization Error

In rehearsal-based CL methods, the interference among tasks throughout the entire training process is intricate, due to the presence of the memory dataset. This introduces an unavoidable challenge in understanding the impact of memory on the performance of rehearsal-based methods.

Following from the definitions of forgetting and generalization in Equations (4) and (5), the key to evaluating their performance lies in calculating the expected value of model errors over any previous task $i^{\text{th}}$ ground truth after learning the final task (i.e., $\mathbb{E}[\mathcal{L}_i(\boldsymbol{w}_T)]$). Indeed, for a generic $t \leq T$, the explicit expressions of $\mathbb{E}[\mathcal{L}_i(\boldsymbol{w}_t)]$ and $\mathbb{E}[\mathcal{L}_i(\boldsymbol{w}_t) - \mathcal{L}_i(\boldsymbol{w}_i)]$ share the same structure for both rehearsal methods. Thus, further following from the definitions of forgetting and generalization error, we present a common performance structure shared by both concurrent rehearsal and sequential rehearsal methods in the following theorem.

**Theorem 5.1.** *Under the problem setups in this work, the forgetting and the generalization error at time $T \geq 2$ in both rehearsal-based methods take the following forms.*

$$F_T = \frac{1}{T-1} \left[ \sum_{i=1}^{T-1} c_i \|\boldsymbol{w}_i^*\|^2 + \sum_{i=1}^{T-1} \sum_{j,k=1}^{T-1} c_{ijk} \|\boldsymbol{w}_j^* - \boldsymbol{w}_k^*\|^2 \right.$$
$$\left. + \sum_{i=1}^{T-1} (noise_T(\sigma) - noise_i(\sigma)) \right],$$

$$G_T = \frac{1}{T} \left[ d_{0T} \sum_{i=1}^{T} \|\boldsymbol{w}_i^*\|^2 + \sum_{i=1}^{T} \sum_{j,k=1}^{T} d_{ijkT} \|\boldsymbol{w}_j^* - \boldsymbol{w}_k^*\|^2 \right]$$
$$+ noise_T(\sigma), \quad (6)$$

*where coefficients and the noise term depend on $p, n, M$, as provided in Appendix C for both rehearsal-based methods.*

Theorem 5.1 indicates that both rehearsal-based methods share the same high-level performance dependence on the system parameters. It can be seen that both of their forgetting and generalization error consist of the following three components. The first component exhibits the form of $C\|\boldsymbol{w}_i^*\|^2$ for some constant $C$, which arises from the overparameterized linear regression error. The second component captures the impact of task dissimilarities, representing the interference among different tasks during the training process. Extracting central information from this component is particularly useful for understanding how task dissimilarity affects the comparison between the two rehearsal-based methods, which is the focus of Section 5.2. The third part captures the impact of the noise level.

Here, we first provide some basic conclusions for the coefficients in Theorem 5.1. (i) By letting $M = 0$, both training methods yield the same result, which is consistent with the memoryless case shown by Lin et al. (2023). (ii) We can also observe that low task similarity negatively impacts model generalization, as $d_{ijkT}$ (defined in Proposition C.2) are non-negative. When $p \to \infty$, we observe that the value of coefficients $d_{ijkT}$ approaches to $0$ for both rehearsal methods, which implies the negative influence of task dissimilarities will be alleviated if the model has enough capacity (i.e., when $p$ is sufficiently large). (iii) We observe that the forgetting approaches to $0$ when $p \to \infty$. This implies that a model with substantial capacity will facilitate effective learning for each task without forgetting, which can also alleviate the negative impact of task dissimilarity.

**Outline of Proof of Theorem 5.1.** We provide a brief outline of the proof of Theorem 5.1 here, and the detailed proof is given in Appendix C. By the definition of forgetting and generalization error, it is sufficient to analyze $\mathbb{E}[\mathcal{L}_i(\boldsymbol{w}_t)]$ and $\mathbb{E}[\mathcal{L}_i(\boldsymbol{w}_t) - \mathcal{L}_i(\boldsymbol{w}_i)]$. We next explain how to obtain the explicit expression for $\mathbb{E}[\mathcal{L}_i(\boldsymbol{w}_t)]$, and $\mathbb{E}[\mathcal{L}_i(\boldsymbol{w}_i)]$ can be calculated in a similar way by substituting $t$ with $i$.

The derivation of $\mathbb{E}[\mathcal{L}_i(\boldsymbol{w}_t)]$ is carried out through an iterative procedure as follows. We first split $\mathbb{E}[\mathcal{L}_i(\boldsymbol{w}_t)]$ into three terms as follows:

$$\mathbb{E}[\mathcal{L}_i(\boldsymbol{w}_t)] = g_t(\mathbb{E}[\mathcal{L}_i(\boldsymbol{w}_{t-1})]) + \text{term}_2 + \text{term}_{\text{noise}},$$

where $g_t$, $\text{term}_2$ and $\text{term}_{\text{noise}}$ are given in Equation (26) in Appendix C. We then analyze each of those three terms.

The first term $g_t(\mathbb{E}[\mathcal{L}_i(\boldsymbol{w}_{t-1})])$ can be evaluated by iteratively rolling out to the initial term, which can then be derived explicitly. Note that the function $g(\cdot)$ takes a linear form, which simplifies the iteration. The second term captures the interference among different tasks during training process. For concurrent rehearsal method, we further derive it by partitioning the data from different tasks and leveraging the properties of block matrices. For sequential rehearsal, we follow the same idea as the memoryless case (Lin et al.,

2023) since different tasks in the memory dataset as well as the current task are learned one by one. The third term captures the noise, which can be analyzed by applying "trace trick" and the properties of Inverse-Wishart distribution.

## 5.2. Comparison Between Concurrent rehearsal and Sequential rehearsal

The main challenge to compare the performance between the two rehearsal-based methods lies in the complexity of the second term in forgetting and generalization in Theorem 5.1, which captures how the task similarity as well as memory data affect the performance. Here the task similarity is characterized by the distance between the true parameters for two tasks. In this section, we will first study a simple case with two tasks, i.e., when $T = 2$, to build our intuition, and then extend to the case with general $T$ based on the central insight obtained in the simple case.

**Two-task Case ($T = 2$):** Given the noise level $\sigma$, we denote $\text{noise}_t = \text{noise}_t(\sigma)$ for simplification. Following from Theorem 5.1, the performance of both rehearsal methods shares the common form:

$$F_2 = \hat{c}_1 \|\boldsymbol{w}_1^*\|^2 + \hat{c}_2 \|\boldsymbol{w}_1^* - \boldsymbol{w}_2^*\|^2 + \text{noise}_2 - \text{noise}_1,$$
$$G_2 = \hat{d}_1(\|\boldsymbol{w}_1^*\|^2 + \|\boldsymbol{w}_2^*\|^2) + \hat{d}_2 \|\boldsymbol{w}_1^* - \boldsymbol{w}_2^*\|^2 + \text{noise}_2,$$

where the specific expressions of constants $\hat{c}_1, \hat{c}_2, \hat{d}_1, \hat{d}_2$ for both rehearsal methods are provided in Appendix D. To compare between the two rehearsal-based methods, the following lemma captures how their coefficients compare with each other.

**Lemma 5.2.** *Under the problem setups of the two-task case, we have*

$$\hat{c}_1^{(concurrent)} < \hat{c}_1^{(sequential)}, \qquad \hat{c}_2^{(concurrent)} > \hat{c}_2^{(sequential)},$$
$$\hat{d}_1^{(concurrent)} < \hat{d}_1^{(sequential)}, \qquad \hat{d}_2^{(concurrent)} > \hat{d}_2^{(sequential)}.$$

Intuitively, when the task dissimilarities are sufficiently large (i.e., $\|\boldsymbol{w}_1^* - \boldsymbol{w}_2^*\|$ is large), then $\hat{c}_2$ and $\hat{d}_2$ will dominant forgetting and generalization error respectively. Then Lemma 5.2 suggests that sequential rehearsal will have less forgetting and generalization error than concurrent rehearsal. Alternatively, if the tasks are very similar and the noise is small, then $\hat{c}_1$ and $\hat{d}_1$ will dominate the performance, and concurrent rehearsal will yield better performance. The following theorem formally establishes the above observations.

**Theorem 5.3.** *Under the problem setups considered in the work, we have*

$$F_2^{(concurrent)} > F_2^{(sequential)} \quad \textit{iff} \quad \frac{\xi_1 \|\boldsymbol{w}_1^* - \boldsymbol{w}_2^*\|^2 + \xi_2 \sigma^2}{\|\boldsymbol{w}_1^*\|^2} > 1,$$
$$G_2^{(concurrent)} > G_2^{(sequential)} \quad \textit{iff} \quad \frac{\mu_1 \|\boldsymbol{w}_1^* - \boldsymbol{w}_2^*\|^2 + \mu_2 \sigma^2}{\|\boldsymbol{w}_1^*\|^2} > 1,$$

*where $\xi_1, \xi_2, \mu_1, \mu_2$ are positive constants with detailed expressions given in Appendix D.*

Theorem 5.3 provably establishes an intriguing fact that the widely used concurrent rehearsal may not always perform better, and sequential rehearsal can perform better when tasks are more different from each other. We further elaborate our comparison between the two methods for the case with $T = 2$ in Appendix D (where the impact of noise is also considered) and with $T = 3$ in Appendix E. The insights obtained from Theorem 5.3 can also be extended to the general case as follows.

**General Case** ($T \geq 2$)**:** Comparing the performance in two rehearsal methods provided in Theorem 5.1 under general $T$ is significantly more challenging, because the mathematical expression of the coefficients become highly complex. However, our insights obtained from the two-task case can still be useful, i.e., sequential rehearsal tends to performance better when tasks are very different. To formalize such an observation, the following lemma compares the coefficients in forgetting and generalization error between the two rehearsal methods.

**Lemma 5.4.** *Under the problem setups considered in the work, the value of coefficients $c_i, c_{ijk}, d_{0T}, d_{ijkT}$ in Theorem 5.1, as derived from different rehearsal methods, satisfy the following relationship.*

$$c_i^{(concurrent)} < c_i^{(sequential)}, \qquad c_{ijk}^{(concurrent)} \geq c_{ijk}^{(sequential)},$$
$$d_{0T}^{(concurrent)} < d_{0T}^{(sequential)}, \qquad d_{ijkT}^{(concurrent)} \geq d_{ijkT}^{(sequential)}.$$

The above lemma suggests that if the tasks are all very different from each other, then sequential rehearsal will have smaller forgetting and generalization error than concurrent rehearsal because $c_{ijk}^{(concurrent)} > c_{ijk}^{(sequential)}$ and $d_{ijkT}^{(concurrent)} > d_{ijkT}^{(sequential)}$ will dominate the comparison. While it is challenging to provide an exact closed-form characterization of the conditions under which sequential rehearsal outperforms concurrent rehearsal, the following theorem presents an example where sequential rehearsal outperforms concurrent rehearsal, based on the understanding outlined above.

**Theorem 5.5.** *Under the problem setups in this work, suppose the ground truth $w_i^*$ is orthonormal to each other for $i \in [T]$, $M \geq 2$, and $p = \mathcal{O}(T^4 n^2 M^2)$. Then we have:*

$$F_T^{(concurrent)} > F_T^{(sequential)} \quad and \quad G_T^{(concurrent)} > G_T^{(sequential)}.$$

In Theorem 5.5, orthonormal $w_i^*$ is an extreme case to have very different tasks. Typically, since forgetting and generalization error are continuous functions of the task dissimilarity, we expect that in the regime that the tasks are highly different, sequential rehearsal will still be advantageous to enjoy less forgetting and smaller generalization error, and

such an advantage should be more apparent as tasks become more dissimilar. To explain this, we consider the generalization error as an example. Assuming that the norm of ground truth is fixed, a higher level of task dissimilarities exacerbates the generalization error since each coefficient $d_{ijkT}$ (defined in Proposition C.2) is positive for both training methods. However, a weaker dependence on task similarities indicates that the generalization error of sequential rehearsal grows slower than concurrent rehearsal as tasks become more dissimilar, resulting advantage for sequential rehearsal to enjoy smaller generalization error. A similar reason is applicable to the forgetting performance, although it is important to note that $c_{ijk}$ is not always positive.

**Simulation Experiments:** To validate our theoretical investigation, we conduct simulation experiments on CL with overparameterized linear models. Set $T = 5$, $p = 500$, $n = 24$, $\sigma = 0$ and $M = 24$. The construction of ground truth features follows two principles: 1) each feature is drawn from the unit sphere, and 2) the task gap (i.e., $\|w_j^* - w_i^*\|$) between any two features is identical. Under different setups of the task gap, the comparisons between theoretical results and simulation results are shown in Figure 2 in terms of both forgetting and generalization error. Here, the theoretical results are calculated according to Theorem 5.1, and the simulation results are obtained by taking the empirical expectation over $10^3$ iterations.

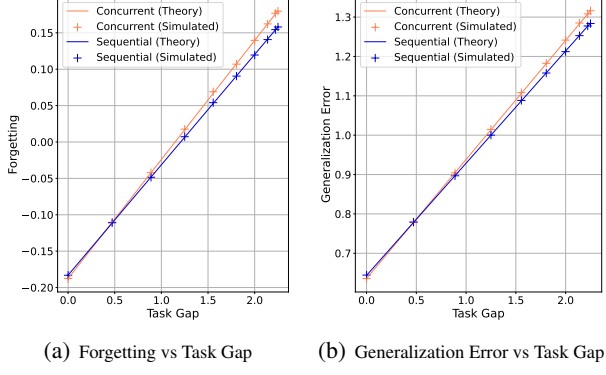

(a) Forgetting vs Task Gap  (b) Generalization Error vs Task Gap

*Figure 2.* Forgetting and Generalization Error vs Task Gap

Several important insights can be immediately obtained from Figure 2: 1) Our theoretical results exactly match with our simulation results, which can clearly corroborate the correctness of our theory. 2) When tasks are similar, i.e., the task gap $\|w_j^* - w_i^*\|^2$ is small than some threshold, concurrent replay is better than sequential replay. However, when tasks become dissimilar, sequential replay starts to outperform concurrent replay in terms of both forgetting and generalization error. And the advantage of sequential replay becomes more significant as the task gap increases, which also aligns with our theoretical results.

**Remark.** It is clear that the order in which old tasks are revisited after current task learning is very important under the framework of sequential rehearsal, which affects both forgetting and generalization errors. Needless to say, the sequential order considered in this work, where tasks are reviewed from the oldest to the newest, is not necessarily the optimal strategy for sequential rehearsal, where however has already demonstrated exciting advantages over concurrent rehearsal. How to design an effective rehearsal order to achieve better performance is a very interesting yet challenging future direction.

## 6. Implications on Practical CL

### 6.1. Hybrid Algorithm Framework

As our theory suggests, sequential rehearsal can benefit CL more than concurrent rehearsal when tasks are dissimilar. Hence, an interesting idea and a potential way to improve the performance is to merge sequential rehearsal into rehearsal-based CL with concurrent rehearsal. Thus inspired, we propose a novel hybrid rehearsal framework, which adapts between concurrent rehearsal and sequential rehearsal for each task based on its similarity with old tasks in the memory. The details are presented in Algorithm 1.

---
**Algorithm 1** *Hybrid Rehearsal* Training Framework

---
**Initialization.** Model parameters $\theta$.
**for** task $t = 1, 2, \ldots, T$ **do**
    Retrieve current data $\mathcal{D}_t$ and memory data $\mathcal{M}_t$
    **if** $\mathcal{M}_t \neq \emptyset$ **then**
        $\mathcal{M}_t^{\text{sim}}, \mathcal{M}_t^{\text{dis}} \leftarrow \text{DIVIDEBUFFER}(\mathcal{M}_t)$
    **end if**
    $\theta \leftarrow \text{CONCURRENTTRAIN}(\mathcal{D}_t \cup \mathcal{M}_t^{\text{sim}})$
    **for** $h : \mathcal{M}_{t,h} \in \mathcal{M}_t^{\text{dis}}$ **do**
        $\theta \leftarrow \text{SEQUENTIALTRAIN}(\mathcal{M}_{t,h})$
    **end for**
    $\mathcal{M}_{t+1} \leftarrow \text{UPDATEMEMORY}(\mathcal{D}_t \cup \mathcal{M}_t)$
**end for**

---

In Algorithm 1, prior to training on task $t$, the overall memory dataset $\mathcal{M}_t$ is first divided into $\mathcal{M}_t^{\text{sim}}$ and $\mathcal{M}_t^{\text{dis}}$, depending on whether previous tasks are similar to the current task or not. Specifically, there are two steps in function DIVIDEBUFFER($\mathcal{M}_t$): we characterize task similarities between the current task $\mathcal{D}_t$ and each previous task $h$ in the memory $\mathcal{M}_{t,h} \in \mathcal{M}_t$, based on their cosine similarity of gradients with respect to the model parameters, i.e., $S_c(\nabla_\theta \mathcal{L}(\mathcal{D}_t, \theta_{t-1}), \nabla_\theta \mathcal{L}(\mathcal{M}_{t,i}), \theta_{t-1})$, where $\mathcal{L}$ denotes training loss and $S_c$ is the cosine similarity function; 2) any previous task for which the similarity score is below a threshold $\tau$ will be regarded as a dissimilar task and its data will be put into $\mathcal{M}_t^{\text{dis}}$. It is important to note that this framework does not rely on very accurate characterizations

of the task similarity. Instead, a heuristic-based estimation should be sufficient, by following the gradient-based similarity characterization as in previous studies (Lopez-Paz & Ranzato, 2017; Lin et al., 2022a;b). To learn task $t$, we first train the model concurrently on the combined dataset consisting of the current task data $\mathcal{D}_t$ and the memory samples from similar tasks $\mathcal{M}_t^{\text{sim}}$. Subsequently, we perform sequential rehearsal by finetuning the learned model on the memory data from each dissimilar task in $\mathcal{M}_t^{\text{dis}}$. The function UPDATEMEMORY represents a general exemplar sampling strategy, such as Reservoir Sampling (Rolnick et al., 2019).

Under the problem setups considered in the work, the **theoretical forgetting and generalization error** of hybrid rehearsal strategy under linear models follow the same general expression as described in Theorem 5.1. The explicit expressions for coefficients are provided in Appendix H. In what follows, we validate the advantage of hybrid rehearsal strategy through experiments conducted on real-world datasets and DNNs.

### 6.2. Hybrid Rehearsal for CL in Practice

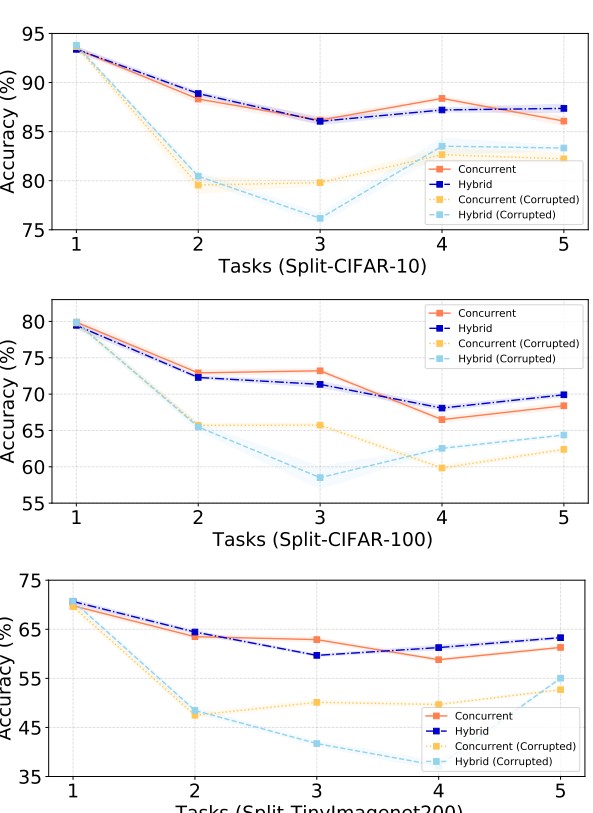

*Figure 3.* Evolution of Accuracy $Acc$ for Concurrent rehearsal and Hybrid rehearsal across three datasets and their corrupted variants, corresponding to the results reported in Table 1.

As an extension of our theoretical results, we verify the

*Table 1.* Averaged Final Accuracy $Acc$ and Averaged Final Forgetting $Fgt$ of different rehearsal methods (concurrent rehearsal vs. hybrid rehearsal) on three datasets and their corrupted variations. "Improvement" shows the $Acc$ or $Fgt$ overhead that hybrid rehearsal achieves over concurrent rehearsal under the same setup. For all datasets, we construct a sequence of 5 tasks for training. The reported results are presented with ± standard deviation. All results are averaged over 10 independent runs.

| Metric | $Acc$ ($\uparrow$) | $Fgt$ ($\downarrow$) | $Acc$ ($\uparrow$) | $Fgt$ ($\downarrow$) | $Acc$ ($\uparrow$) | $Fgt$ ($\downarrow$) |
|---|---|---|---|---|---|---|
| **Dataset** | Split-CIFAR-10 | | Split-CIFAR-100 | | Split-TinyImagenet200 | |
| *Concurrent rehearsal* | $86.20 \pm 0.68$ | $9.73 \pm 0.76$ | $68.43 \pm 0.93$ | $14.56 \pm 1.15$ | $61.10 \pm 0.28$ | $13.28 \pm 0.30$ |
| *Hybrid rehearsal* | $\mathbf{87.27} \pm 0.59$ | $\mathbf{8.25} \pm 0.50$ | $\mathbf{69.92} \pm 0.52$ | $\mathbf{11.29} \pm 0.79$ | $\mathbf{63.29} \pm 0.47$ | $\mathbf{9.62} \pm 0.45$ |
| Improvement | $+1.07$ | $-1.48$ | $+1.49$ | $-3.27$ | $+2.19$ | $-3.66$ |
| **Dataset** | Corrupted Split-CIFAR-10 | | Corrupted Split-CIFAR-100 | | Corrupted Split-TinyImagenet200 | |
| *Concurrent rehearsal* | $82.01 \pm 0.90$ | $11.14 \pm 0.98$ | $62.33 \pm 0.57$ | $17.35 \pm 0.75$ | $52.69 \pm 0.44$ | $15.23 \pm 0.49$ |
| *Hybrid rehearsal* | $\mathbf{83.45} \pm 0.22$ | $\mathbf{6.90} \pm 0.56$ | $\mathbf{64.34} \pm 0.36$ | $\mathbf{9.49} \pm 1.67$ | $\mathbf{55.07} \pm 0.33$ | $\mathbf{1.91} \pm 0.69$ |
| Improvement | $+1.44$ | $-4.24$ | $+2.01$ | $-7.86$ | $+2.38$ | $-13.32$ |

performance of the proposed hybrid framework in Algorithm 1 and compare it with the widely-used concurrent rehearsal. We consider three real-world datasets under a task-incremental CL setup: CIFAR-10, CIFAR-100 (Krizhevsky et al., 2009), and TinyImagenet200 (Le & Yang, 2015). Following recent work (Van de Ven et al., 2022), we randomly partition the classes from each dataset into multiple tasks, yielding Split-CIFAR-10, Split-CIFAR-100, and Split-TinyImagenet200. For example, in Split-CIFAR-10, the sequence of tasks $\{\mathcal{T}_1, \ldots, \mathcal{T}_5\}$ is constructed such that each task contains two distinct classes. The objective for each task $\mathcal{T}_t$ is to classify between $\{\mathcal{Y}_{t,1}, \mathcal{Y}_{t,2}\}$ with the task label explicitly provided during training and testing. We employ a non-pretrained ResNet18 as the backbone model for all three datasets. To evaluate the performance, we denote $a_{k,t}$ as the testing accuracy on task $t$ after training task $k$ and consider the following two metrics: 1) Final Average Accuracy ($Acc$) across all seen tasks after training defined as $Acc = \frac{1}{T} \sum_{t=1}^{T} a_{T,t}$, which can be regarded as generalization error, and 2) Final Average Forgetting ($Fgt$), defined as $Fgt = \frac{1}{T-1} \sum_{t=1}^{T-1} (a_{t,t} - a_{T,t})$, which evaluates the average accuracy drop of old tasks after learning the final task.

Here, to simplify our comparison between hybrid rehearsal and concurrent rehearsal while avoiding extraneous complexities such as rehearsal task ordering (which is beyond the scope of this paper), we adopt a straightforward relaxation: at most one task with the lowest similarity characterization in memory is designated as the "dissimilar task". As shown in Table 1, hybrid rehearsal consistently outperforms concurrent rehearsal across all three datasets in terms of both average accuracy and forgetting. For the original datasets, the most significant improvement is observed on Split-TinyImagenet200, where hybrid rehearsal achieves a **2.19%** higher accuracy than concurrent rehearsal while also reducing forgetting by **3.66**. We further report the perfor-

mance evolution during CL for all settings from Table 1 in Figure 3, where each point shows the average performance of the model after learning task $t$ on all seen tasks so far. While the performance of hybrid rehearsal may degrade during intermediate tasks, it consistently surpasses concurrent rehearsal in terms of final performance, highlighting the effectiveness of sequentially fine-tuning the model on dissimilar task data for better knowledge consolidation.

To demonstrate the potential of hybrid rehearsal, we consider a challenging setup on Split-CIFAR-100 and Split-TinyImagenet200 with a long task sequence, i.e., under a 20-task setting. As shown in Table 2, hybrid rehearsal outperforms concurrent rehearsal by a more pronounced gap, demonstrating the scalability and effectiveness of the hybrid rehearsal strategy in scenarios with a large number of tasks.

Furthermore, following our theoretical insight, the hybrid rehearsal strategy exhibits greater advantages when the task dissimilarity is higher. To validate this, we increase task dissimilarity by applying image corruption to one specific task in each of the three datasets, generating Corrupted Split-CIFAR-10, Corrupted Split-CIFAR-100, and Corrupted Split-TinyImagenet200. Details of the employed image corruption schemes are provided in Appendix A.3. As shown in the second part of Table 1, hybrid rehearsal outperforms concurrent rehearsal in both accuracy and forgetting by a larger margin on all three corrupted dataset variations, aligning with our theoretical insight.

To characterize the impact of task similarity on hybrid rehearsal, we next control the similarity by applying various proportions of label corruption in the task sequence on Split-TinyImagenet200. For each class within a corrupted task, $p_{cor}\%$ of training samples are randomly selected and their labels are uniformly reassigned to other class labels within the same task. Intuitively, the tasks are more dissimilar when a larger proportion of labels are corrupted. In particular, we

*Table 2. Acc and Fgt of different rehearsal methods on Split-CIFAR-100 and Split-TinyImagenet200. For all datasets, we construct a sequence of 20 tasks for training. All results are averaged over 5 independent runs.*

| Dataset | Split-CIFAR-100 | Split-TinyImagenet200 |
|---|---|---|
| *Concurrent* | $67.99 \pm 0.21$ | $62.94 \pm 0.60$ |
| *Hybrid* | $72.31 \pm 0.38$ | $66.46 \pm 0.13$ |
| Improvement | $+4.32$ | $+3.52$ |

*Table 2.1 Acc (↑)*

| Dataset | Split-CIFAR-100 | Split-TinyImagenet200 |
|---|---|---|
| *Concurrent* | $13.91 \pm 0.11$ | $15.05 \pm 0.81$ |
| *Hybrid* | $7.43 \pm 0.18$ | $13.03 \pm 0.79$ |
| Improvement | $-6.48$ | $-2.02$ |

*Table 2.2 Fgt (↓)*

*Table 3. Acc and Fgt of different rehearsal methods on Split-Tiny-Imagenet200, where a subset of training data labels are corrupted in the first two tasks. All results are averaged over 5 independent runs.*

| Setting | $p_{cor} = 5\%$ | $p_{cor} = 10\%$ | $p_{cor} = 20\%$ |
|---|---|---|---|
| *Concurrent* | $57.36 \pm 0.06$ | $53.92 \pm 0.78$ | $49.07 \pm 0.59$ |
| *Hybrid* | $60.02 \pm 0.36$ | $56.88 \pm 0.12$ | $53.73 \pm 0.60$ |
| Improvement | $+2.66$ | $+2.96$ | $+4.66$ |

*Table 3.1 Acc (↑)*

| Setting | $p_{cor} = 5\%$ | $p_{cor} = 10\%$ | $p_{cor} = 20\%$ |
|---|---|---|---|
| *Concurrent* | $14.85 \pm 0.28$ | $17.40 \pm 0.82$ | $17.35 \pm 1.15$ |
| *Hybrid* | $11.26 \pm 0.10$ | $12.73 \pm 1.00$ | $12.48 \pm 0.78$ |
| Improvement | $-3.59$ | $-4.67$ | $-4.87$ |

*Table 3.2 Fgt (↓)*

consider three different settings: $p_{cor} = 5\%, 10\%, 20\%$. It can be seen from Table 3 that hybrid rehearsal consistently outperforms concurrent rehearsal, and more importantly, the performance improvement becomes more significant as tasks are more dissimilar. These results further justify the correctness and usefulness of our theoretical results. Note that the performance of hybrid rehearsal has not been optimized in terms of the rehearsal order and selection of similar tasks, which may further improve the effectiveness of sequential rehearsal. This encouraging result highlights the great potential of exploiting sequential rehearsal in improving the performance of rehearsal-based CL.

## 7. Conclusion

In this work, we took a closer look at the rehearsal strategy in rehearsal-based CL and questioned the effectiveness of the widely used training technique, i.e., concurrent rehearsal.

In particular, we proposed a novel rehearsal strategy, namely sequential rehearsal, which revisits old tasks in the memory sequentially after current task learning. By leveraging over-parameterized linear models with equal memory allocation, we provided the first explicit expressions of both forgetting and generalization errors under two rehearsal methods: concurrent rehearsal and sequential rehearsal. Comparisons between their theoretical performance led to the insight that sequential rehearsal outperforms concurrent rehearsal in terms of forgetting and generalization error when the tasks are less similar, which is consistent with our motivations from human learning and multitask learning. Our simulation results on linear models further corroborated the correctness of our theoretical results. More importantly, based on our theory, we proposed a novel hybrid rehearsal framework for practical CL and experiments on real-world data with DNNs verified the superior performance of this framework over traditional concurrent rehearsal. To the best of our knowledge, our work provides the first comprehensive theoretical study on rehearsal for rehearsal-based CL, which could motivate more principled designs for better rehearsal-based CL.

## Acknowledgment

This work has been supported in part by the U.S. National Science Foundation under the grants: NSF AI Institute (AI-EDGE) 2112471, CNS-2312836, RINGS-2148253, CNS-2112471, 2324052, and ECCS-2413528, Office of Naval Research Grant N000142412729, and was sponsored by the Army Research Laboratory under Cooperative Agreement Number W911NF-23-2-0225. The views and conclusions contained in this document are those of the authors and should not be interpreted as representing the official policies, either expressed or implied, of the Army Research Laboratory or the U.S. Government. The U.S. Government is authorized to reproduce and distribute reprints for Government purposes notwithstanding any copyright notation herein.

## Impact Statement

This paper presents work whose goal is to advance the field of Machine Learning. There are many potential societal consequences of our work, none which we feel must be specifically highlighted here.

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

# Supplementary Materials

## A. DNN Experiments Details

### A.1. Hardware Details

We outline the hardware configuration used to conduct the experiments on DNN:

- Operating system: Red Hat Enterprise Linux Server 7.9 (Maipo)

- Type of CPU: 2.4 GHz 14-Core Intel Xeon E5-2680 v4 (Broadwell)

- Type of GPU: NVIDIA P100 "Pascal" GPU with 16GB memory

### A.2. Implementation Details

**Dataset.** We evaluate our hybrid rehearsal on variations of standard image classification datasets: Split-MNIST, Split-CIFAR-10, Split-CIFAR-100, and Split-TinyImagenet200 (Van de Ven et al., 2022; Guo et al., 2022; Sun et al., 2022).

- MNIST (LeCun et al., 1989) contains $60,000$ $28\times28$ grayscale train images and $10,000$ test images of 10 unique classes. For the experiments reported in Table 5, we randomly split 10 classes into 5 tasks, each containing 2 non-overlapping classes.

- CIFAR-10 (Krizhevsky et al., 2009) contains $50,000$ $32 \times 32$ color train images and $10,000$ test images of 10 unique classes. For the experiments reported in Table 1, we randomly split 10 classes into 5 tasks, each containing 2 non-overlapping classes.

- CIFAR-100 (Krizhevsky et al., 2009) contains $50,000$ $32 \times 32$ color train images and $10,000$ test images of 100 unique classes. For the experiments reported in Table 1, we randomly split 25 classes into 5 tasks, each containing 5 non-overlapping classes. For the experiments reported in Table 2, we randomly split 100 classes into 20 tasks, each containing 5 non-overlapping classes. For the experiments reported in Table 6, we randomly split 100 classes into 10 tasks, each containing 10 non-overlapping classes.

- TinyImagenet200 (Le & Yang, 2015) contains $100,000$ $64 \times 64$ color train images and $20,000$ test images of 200 unique classes. For the experiments reported in Table 1 and Table 3, we randomly split 50 classes into 5 tasks, each containing 10 non-overlapping classes. For the experiments reported in Table 2, we randomly split 200 classes into 20 tasks, each containing 10 non-overlapping classes.

**DNN Architecture and Training Details.** For training on Split-MNIST, we employ a three-layer MLP with two fully connected hidden layers of 400 ReLU units following (Van de Ven et al., 2022). For training on Split-CIFAR-10, Split-CIFAR-100, and Split-TinyImagenet200, we employ a non-pretrained ResNet-18 as our DNN backbone. Following (Van de Ven et al., 2022), we adopt a multi-headed output layer such that each task is assigned its own output layer, consistent with the typical Task Incremental CL setup. During supervised training, we explicitly provide the task identifier (ranging from 1 to 5 for Split-CIFAR-10) alongside the image-label pairs as additional input to the model. For simplicity, we use a reservoir sampling strategy to construct the memory buffer. Our buffer size is $200, 300, 30, 30$ per class for Split-MNIST, Split-CIFAR-10, Split-CIFAR-100, and Split-TinyImagenet200, correspondingly. For experiments not involving image corruption, we didn't apply any data augmentation before training.

For all experiments on *concurrent rehearsal* and *sequential rehearsal*, we use the SGD optimizer with a StepLR learning rate scheduler, which decays the learning rate by a fixed factor at predefined intervals. The detailed parameters for each dataset are listed in Table 4.

**Similarity Threshold**. We report the employed similarity threshold parameter $\tau$ corresponding to different setting in Table 1. For Split-CIFAR-100, and Corrupted Split-CIFAR-100, $\tau = -0.1$. For Split-CIFAR-10, Corrupted Split-CIFAR-10, Split-TinyImagenet200, and Corrupted Split-TinyImagenet200, $\tau = 0$.

*Table 4.* Training parameters for all four datasets.

| Method | Parameter | Split-MNIST | Split-CIFAR-10 | Split-CIFAR-100 | Split-TinyImagenet200 |
|---|---|---|---|---|---|
| *Concurrent rehearsal* | Epoch | 15 | 30 | 30 | 40 |
| | Minibatch Size | 256 | 128 | 128 | 128 |
| | Momentum | 0.9 | 0.9 | 0.9 | 0.9 |
| | Weight Decay | $1e^{-5}$ | $1e^{-4}$ | $1e^{-4}$ | $2e^{-4}$ |
| | Initial LR | 0.05 | 0.05 | 0.05 | 0.05 |
| | LR Decay Factor | 0.1 | 0.1 | 0.1 | 0.1 |
| | LR Step | 10 | 20 | 20 | 25 |
| *Sequential rehearsal* | Epoch | 15 | 30 | 30 | 40 |
| | Minibatch Size | 128 | 64 | 64 | 64 |
| | Momentum | 0.9 | 0.9 | 0.9 | 0.9 |
| | Weight Decay | $1e^{-5}$ | $1e^{-3}$ | $1e^{-3}$ | $2e^{-3}$ |
| | Initial LR | 0.0007 | 0.001 | 0.001 | 0.002 |
| | LR Decay Factor | 0.8 | 0.5 | 0.5 | 0.5 |
| | LR Step | 10 | 12 | 12 | 16 |

### A.3. Task Corruption

For experiments described in Section 6.2, we control the similarity level of the dataset by applying data corruption to a selected task. We provide a list of sample images under different image corruption schemes in Figure 4.

In Table 1, for Corrupted Split-CIFAR-10, we apply Glass Corruption on all $\mathcal{T}_1$ training data. For Corrupted Split-CIFAR-100, we apply Color-swapping, Gaussian Blur, and Rotating Corruption on all $\mathcal{T}_2$ training data. For Corrupted Split-TinyImagenet200, we apply Color-swapping, Gaussian Blur, and Rotating Corruption on all $\mathcal{T}_1$ training data.

In Table 5, for Corrupted Split-MNIST, we apply Rotation on all $\mathcal{T}_2$ training data.

In Table 6, for the scenario "Original Dataset", we don't apply any image corruption. For the scenario "1 Corruption", we apply the Glass corruption on $\mathcal{T}_1$. For the scenario "2 Corruption", we apply Glass corruption on $\mathcal{T}_1$, and rotational color swapping on $\mathcal{T}_2$. For the scenario "3 Corruption", we apply Glass corruption on $\mathcal{T}_1$, rotational color swapping on $\mathcal{T}_3$, and elastic pixelation on $\mathcal{T}_5$.

### A.4. Additional Results

We report the performance of hybrid rehearsal and concurrent rehearsal on Split-MNIST and its corrupted variation in Table 5.

To further demonstrate the effect of task similarity on hybrid rehearsal, we seek to control the similarity by using the number of corrupted tasks (i.e., task with corrupted images) in the task sequence on Split-CIFAR-100. In particular, we consider three different scenarios, '1 Corruption' with 1 corrupted task, '2 Corruption' with 2 corrupted tasks, and '3 Corruption' with 3 corrupted tasks. Intuitively, the tasks are more dissimilar when more tasks are corrupted. It can be seen from Table 6 that hybrid rehearsal consistently outperforms concurrent rehearsal, and more importantly, the performance improvement becomes more significant as tasks are more dissimilar.

*Table 5.* $Acc$ and $Fgt$ of different rehearsal methods (concurrent rehearsal vs. hybrid rehearsal) on Split-MNIST and its corrupted variations. All results are averaged over 5 independent runs.

| Metric | $Acc$ ($\uparrow$) | $Fgt$ ($\downarrow$) | $Acc$ ($\uparrow$) | $Fgt$ ($\downarrow$) |
|---|---|---|---|---|
| **Dataset** | Split-MNIST | | Corrupted Split-MNIST | |
| *Concurrent rehearsal* | $95.24 \pm 0.25$ | $5.33 \pm 0.30$ | $94.29 \pm 0.15$ | $6.48 \pm 0.20$ |
| *Hybrid rehearsal* | $\mathbf{95.39} \pm 0.16$ | $\mathbf{1.65} \pm 1.12$ | $\mathbf{94.81} \pm 0.08$ | $\mathbf{2.57} \pm 0.73$ |
| Improvement | $+0.15$ | $-3.68$ | $+0.52$ | $-3.91$ |

*Table 6.* $Acc$ and $Fgt$ of different rehearsal methods on Split-CIFAR-100 with varying numbers of corrupted tasks. We construct a sequence of 10 tasks for training. "Corruption Number = $s$" indicates that data corruption was applied to $s$ out of 10 tasks, making it more dissimilar than others. All results are averaged over 4 independent runs.

| Corruption Number | 0 | 1 | 2 | 3 |
|---|---|---|---|---|
| *Concurrent rehearsal* | $68.27 \pm 0.33$ | $64.24 \pm 0.23$ | $60.67 \pm 0.23$ | $58.93 \pm 0.11$ |
| *Hybrid rehearsal* | $\mathbf{68.79} \pm 0.43$ | $\mathbf{64.81} \pm 0.16$ | $\mathbf{61.30} \pm 0.17$ | $\mathbf{59.72} \pm 0.15$ |
| Improvement | $+0.52$ | $+0.57$ | $+0.63$ | $+0.79$ |

Table 4.1 $Acc$ ($\uparrow$)

| Corruption Number | 0 | 1 | 2 | 3 |
|---|---|---|---|---|
| *Concurrent rehearsal* | $6.24 \pm 0.11$ | $7.76 \pm 0.09$ | $9.28 \pm 0.16$ | $8.57 \pm 0.08$ |
| *Hybrid rehearsal* | $\mathbf{6.10} \pm 0.22$ | $\mathbf{7.23} \pm 0.10$ | $\mathbf{8.75} \pm 0.09$ | $\mathbf{8.35} \pm 0.10$ |
| Improvement | $-0.14$ | $-0.53$ | $-0.53$ | $-0.22$ |

Table 4.2 $Fgt$ ($\downarrow$)

# B. Supporting Lemmas

Define $P_{\boldsymbol{X}} = \boldsymbol{X}(\boldsymbol{X}^\top \boldsymbol{X})^{-1}\boldsymbol{X}^\top$ and $\boldsymbol{X}^\dagger = \boldsymbol{X}(\boldsymbol{X}^\top \boldsymbol{X})^{-1}$. We first provide some useful lemmas for the derivation of forgetting and generalization error. In the following lemma, we provide the expression of the SGD convergence point when training on a single task.

**Lemma B.1.** *Suppose $\boldsymbol{X} \in \mathbb{R}^{p \times m}$ and $\boldsymbol{Y} \in \mathbb{R}^m$, where $\boldsymbol{Y} = \boldsymbol{X}^\top \boldsymbol{w}^* + \boldsymbol{z}$. Consider the optimization problem:*

$$\boldsymbol{w}_o = \arg\min_{\boldsymbol{w}} \|\boldsymbol{w} - \boldsymbol{w}_s\|_2^2$$
$$s.t. \ \boldsymbol{X}^\top \boldsymbol{w} = \boldsymbol{Y}.$$

*The solution of the above problem can be written as:*

$$\boldsymbol{w}_o = \boldsymbol{w}_s + \boldsymbol{X}^\dagger(\boldsymbol{Y} - \boldsymbol{X}^\top \boldsymbol{w}_{in}),$$

*or equivalently,*

$$\boldsymbol{w}_o = (\boldsymbol{I} - P_{\boldsymbol{X}})\boldsymbol{w}_s + P_{\boldsymbol{X}}\boldsymbol{w}^* + \boldsymbol{X}^\dagger \boldsymbol{z}.$$

*Proof.* The detailed proof refers to Lemma B.1 in Lin et al. (2023). $\qquad\square$

**Lemma B.2.** *Suppose each element of the random matrix $\boldsymbol{X} \in \mathbb{R}^{p \times m}$ follows i.i.d. standard distribution and $v \in \mathbb{R}^p$, then we have:*

$$\mathbb{E}\|P_{\boldsymbol{X}}v\|^2 = \frac{m}{p}\|v\|^2.$$

*Proof.* The detailed proof refers to Proposition 3 in Ju et al. (2023). $\qquad\square$

**Lemma B.3.** *Suppose each element of the random matrix $\boldsymbol{X} \in \mathbb{R}^{p \times m}$ follows i.i.d. standard distribution. Also, the random vector $z \in \mathbb{R}^m$ follows $\mathcal{N}(0, \sigma^2 \boldsymbol{I}_m)$ independently. Then, we have:*

$$\mathbb{E}\|\boldsymbol{X}^\dagger \boldsymbol{z}\|^2 = \frac{m\sigma^2}{p - m - 1}.$$

*Proof.* The proof is completed by applying the "trace trick" as follows.

$$\mathbb{E}\|\boldsymbol{X}^\dagger \boldsymbol{z}\|^2 = \mathbb{E}\left[\boldsymbol{z}^\top \left(\boldsymbol{X}^\top \boldsymbol{X}\right)^{-1} \boldsymbol{z}\right]$$
$$= \mathbb{E}\left[\text{tr}\left[\left(\boldsymbol{X}^\top \boldsymbol{X}\right)^{-1} \boldsymbol{z}\boldsymbol{z}^\top\right]\right]$$
$$\overset{(i)}{=} \text{tr}\left[\mathbb{E}\left[\left(\boldsymbol{X}^\top \boldsymbol{X}\right)^{-1}\right] \mathbb{E}\left[\boldsymbol{z}\boldsymbol{z}^\top\right]\right]$$

$$\overset{(ii)}{=} \sigma^2 \text{tr}\left[\mathbb{E}\left[\left(\boldsymbol{X}^\top \boldsymbol{X}\right)^{-1}\right]\right]$$

$$\overset{(iii)}{=} \frac{m\sigma^2}{p - m - 1},$$

where $(i)$ follows from the independence between $\boldsymbol{X}$ and $\boldsymbol{z}$, $(ii)$ follows from the fact that $\mathbb{E}\left[\boldsymbol{z}\boldsymbol{z}^\top\right] = \sigma^2 \boldsymbol{I}_m$ and $(iii)$ follows from the fact that $\left(\boldsymbol{X}^\top \boldsymbol{X}\right)^{-1}$ follows the inverse-Wishart distribution with identity scale matrix $\boldsymbol{I}_m$ and $p$ degrees-of-freedom. $\square$

**Lemma B.4.** *Suppose $\boldsymbol{X} \in \mathbb{R}^{p \times m}$ and $\boldsymbol{v}_1, \boldsymbol{v}_2 \in \mathbb{R}^p$, then we have:*

$$\langle (\boldsymbol{I} - P_{\boldsymbol{X}})\boldsymbol{v}_1, \boldsymbol{X}^\dagger \boldsymbol{v}_2 \rangle = 0,$$
$$\langle (\boldsymbol{I} - P_{\boldsymbol{X}})\boldsymbol{v}_1, P_{\boldsymbol{X}}\boldsymbol{v}_2 \rangle = 0.$$

*Proof.* The proof follows from the definitions of $P_{\boldsymbol{X}}$ and $\boldsymbol{X}^\dagger$ straightforward. $\square$

Now, we provide useful lemmas in proving Lemma C.1 for the concurrent rehearsal method.

**Lemma B.5.** *Suppose $P \in \mathbb{R}^{p \times p}$ is a projection matrix and each element of the random vector $\boldsymbol{v} \in \mathbb{R}^p$ follows i.i.d. standard Gaussian distribution, then $P\boldsymbol{v}$ and $(\boldsymbol{I} - P)\boldsymbol{v}$ are independent. Moreover, Suppose each element of the random matrix $\boldsymbol{V} \in \mathbb{R}^{p \times m}$ follows i.i.d. standard Gaussian distribution, then $P\boldsymbol{V}$ and $(\boldsymbol{I} - P)\boldsymbol{V}$ are independent*

*Proof.* The proof for vector case is completed in two steps. First, we prove that $P\boldsymbol{v}$ and $(\boldsymbol{I} - P)\boldsymbol{v}$ are jointly Gaussian. Denote $\boldsymbol{z} = \begin{bmatrix} P\boldsymbol{v} \\ (\boldsymbol{I} - P)\boldsymbol{v} \end{bmatrix}$ as the concatenation of the two vectors and $\boldsymbol{w} = \begin{bmatrix} \boldsymbol{w}_1 \\ \boldsymbol{w}_2 \end{bmatrix}$ as an arbitrary vector, where $\boldsymbol{w}_1, \boldsymbol{w}_2 \in \mathbb{R}^p$. Since the linear combination $\boldsymbol{w}^\top \boldsymbol{z} = (\boldsymbol{w}_1^\top P + \boldsymbol{w}_2^\top (\boldsymbol{I} - P))\boldsymbol{v}$ remains Gaussian distribution for any $\boldsymbol{w}$, we conclude that $P\boldsymbol{v}$ and $(\boldsymbol{I} - P)\boldsymbol{v}$ are jointly Gaussian. Next, we prove that they are uncorrelated as follows:

$$\begin{aligned} \text{Cov}(P\boldsymbol{v}, (\boldsymbol{I} - P)\boldsymbol{v}) &= \mathbb{E}\left[P\boldsymbol{v}((\boldsymbol{I} - P)\boldsymbol{v})^\top\right] \\ &= P\mathbb{E}(\boldsymbol{v}\boldsymbol{v}^\top)(\boldsymbol{I} - P) \\ &\overset{(i)}{=} P(\boldsymbol{I} - P) \\ &= 0, \end{aligned}$$

where $(i)$ follows from the fact that $\boldsymbol{v}$ has i.i.d. standard Gaussian elements. By combining these two facts, we conclude that $P\boldsymbol{v}$ and $(\boldsymbol{I} - P)\boldsymbol{v}$ are independent. For the matrix case, we can equivalently consider the vector $\hat{\boldsymbol{v}} \in \mathbb{R}^{pm}$ which is formed by concatenating all the columns of $\boldsymbol{V}$ and the projection matrix $\hat{P} = \text{diag}([P, P, .., P]) \in \mathbb{R}^{pm \times pm}$. $\square$

**Lemma B.6.** *Suppose each element of the random matrix $\boldsymbol{X} \in \mathbb{R}^{p \times m}$ follows i.i.d. standard Gaussian distribution and $\boldsymbol{v} \in \mathbb{R}^p$, then we have:*

$$\mathbb{E}\left[\boldsymbol{X}^\top v v^\top \boldsymbol{X}\right] = \|v\|^2 \cdot \boldsymbol{I}.$$

*Proof.* To clarify, we denote $\boldsymbol{X} = [\boldsymbol{x}_1, ..., \boldsymbol{x}_n]$, where $\boldsymbol{x}_i$ is the $i^{th}$ column of $\boldsymbol{X}$. We also denote $[\cdot]_{i,j}$ as the element of $i^{th}$ row and $j^{th}$ column of a matrix. Due to the independence of elements in $\boldsymbol{X}$, it follows that:

$$\left[\mathbb{E}\left[\boldsymbol{X}^\top v v^\top \boldsymbol{X}\right]\right]_{i,j} = \text{cov}(\boldsymbol{v}^\top \boldsymbol{x}_i, \boldsymbol{v}^\top \boldsymbol{x}_j) = \begin{cases} 0 & \text{if } i \neq j, \\ \|v\|^2 & \text{if } i = j. \end{cases}$$

$\square$

**Lemma B.7.** *Suppose each element of the random matrix $\boldsymbol{X} \in \mathbb{R}^{p \times m}$ follows i.i.d. standard Gaussian distribution and the projection matrix $P \in \mathbb{R}^{p \times p}$ satisfying $\text{rank}(P) = d$, then we have:*

$$\text{tr}\left(\mathbb{E}\left[\left(\boldsymbol{X}^\top (\boldsymbol{I} - P)\boldsymbol{X}\right)^{-1}\right]\right) = \frac{m}{p - d - m - 1}.$$

*Proof.* We first note that $(\boldsymbol{I} - P)$ is a projection matrix with $(p - d)$ many eigenvalues 1 and $d$ many eigenvalues 0. With loss of generalization, we write $(\boldsymbol{I} - P) = U^\top \Sigma U$ where $\Sigma = \mathrm{diag}([1, ..., 1, 0, ..., 0])$ is a diagonal matrix, whose first $(p - d)$ diagonal elements are 1 while others are 0, and $U$ is an orthogonal matrix. We denote $\hat{X} \in \mathbb{R}^{(p-d)\times n}$ as the first $(p - d)$ rows of $\boldsymbol{X}$. Then, we have:

$$
\begin{aligned}
\mathrm{tr}\left(\mathbb{E}\left[\left(\boldsymbol{X}^\top(\boldsymbol{I} - P)\boldsymbol{X}\right)^{-1}\right]\right) &= \mathrm{tr}\left(\mathbb{E}\left[\left(\boldsymbol{X}^\top U^\top \Sigma U \boldsymbol{X}\right)^{-1}\right]\right) \\
&\overset{(i)}{=} \mathrm{tr}\left(\mathbb{E}\left[\left(\boldsymbol{X}^\top \Sigma \boldsymbol{X}\right)^{-1}\right]\right) \\
&= \mathrm{tr}\left(\mathbb{E}\left[\left(\hat{\boldsymbol{X}}^\top \hat{\boldsymbol{X}}\right)^{-1}\right]\right) \\
&\overset{(ii)}{=} \frac{m}{p - d - m - 1}
\end{aligned}
$$

where $(i)$ follows from the rotational symmetry of standard Gaussian distribution, $(ii)$ follows from the fact that $\left(\hat{\boldsymbol{X}}^\top \hat{\boldsymbol{X}}\right)^{-1}$ follows the inverse-Wishart distribution with identity scale matrix $\boldsymbol{I}_m$ and $(p - d)$ degrees-of-freedom.. $\qquad\square$

**Lemma B.8.** *Suppose each element of the random matrices $\boldsymbol{X}_1 \in \mathbb{R}^{p\times m_1}$, $\boldsymbol{X}_2 \in \mathbb{R}^{p\times m_2}$ follows i.i.d. standard Gaussian distribution and $\boldsymbol{v} \in \mathbb{R}^p$. Denote $\boldsymbol{V} = [\boldsymbol{X}_1, \boldsymbol{X}_2]$, then we have:*

$$
\mathbb{E}\left\|\boldsymbol{V}^\dagger \begin{bmatrix} \boldsymbol{X}_1^\top \\ \boldsymbol{0} \end{bmatrix} \boldsymbol{v}\right\|^2 = \frac{m_1}{p}\cdot\left(1 + \frac{m_2}{p - m_1 - m_2 - 1}\right)\|\boldsymbol{v}\|^2
$$

*Proof.* First of all, we partition the matrix $\boldsymbol{V}^\top\boldsymbol{V}$ into four blocks as follows:

$$
\boldsymbol{V}^\top\boldsymbol{V} = \begin{bmatrix} \boldsymbol{X}_1^\top \\ \boldsymbol{X}_2^\top \end{bmatrix}\begin{bmatrix} \boldsymbol{X}_1 & \boldsymbol{X}_2 \end{bmatrix} = \begin{bmatrix} \boldsymbol{X}_1^\top\boldsymbol{X}_1 & \boldsymbol{X}_1^\top\boldsymbol{X}_2 \\ \boldsymbol{X}_2^\top\boldsymbol{X}_1 & \boldsymbol{X}_2^\top\boldsymbol{X}_2 \end{bmatrix}.
$$

Then, we partition the matrix $(\boldsymbol{V}^\top\boldsymbol{V})^{-1}$ following the same partitioning scheme as $\boldsymbol{V}^\top\boldsymbol{V}$:

$$
(\boldsymbol{V}^\top\boldsymbol{V})^{-1} = \begin{bmatrix} A_{1,1} & A_{1,2} \\ A_{2,1} & A_{2,2} \end{bmatrix},
$$

where $A_{1,1} \in \mathbb{R}^{m_1\times m_1}$. More specifically, we have

$$
\begin{aligned}
A_{1,1} &= (\boldsymbol{X}_1^\top\boldsymbol{X}_1)^{-1} - (\boldsymbol{X}_1^\top\boldsymbol{X}_1)^{-1}\boldsymbol{X}_1^\top\boldsymbol{X}_2\left(\boldsymbol{X}_2^\top\boldsymbol{X}_2 - \boldsymbol{X}_2^\top\boldsymbol{X}_1(\boldsymbol{X}_1^\top\boldsymbol{X}_1)^{-1}\boldsymbol{X}_1^\top\boldsymbol{X}_2\right)^{-1}\boldsymbol{X}_2^\top\boldsymbol{X}_1(\boldsymbol{X}_1^\top\boldsymbol{X}_1)^{-1} \\
&= P_{\boldsymbol{X}_1} + P_{\boldsymbol{X}_1}\boldsymbol{X}_2\left(\boldsymbol{X}_2^\top(\boldsymbol{I} - P_{\boldsymbol{X}_1})\boldsymbol{X}_2\right)^{-1}\boldsymbol{X}_2^\top P_{\boldsymbol{X}_1}.
\end{aligned}
$$

Therefore, we have

$$
\begin{aligned}
\mathbb{E}\left\|\boldsymbol{V}^\dagger \begin{bmatrix} \boldsymbol{X}_1^\top \\ \boldsymbol{0} \end{bmatrix} \boldsymbol{v}\right\|^2 &= \mathbb{E}\left[\boldsymbol{v}^\top\left[P_{\boldsymbol{X}_1} + P_{\boldsymbol{X}_1}\boldsymbol{X}_2\left(\boldsymbol{X}_2^\top(\boldsymbol{I} - P_{\boldsymbol{X}_1})\boldsymbol{X}_2\right)^{-1}\boldsymbol{X}_2^\top P_{\boldsymbol{X}_1}\right]\boldsymbol{v}\right] \\
&\overset{(i)}{=} \frac{m_1}{p}\|\boldsymbol{v}\|^2 + \mathbb{E}\left[\boldsymbol{v}^\top\left[P_{\boldsymbol{X}_1}\boldsymbol{X}_2\left(\boldsymbol{X}_2^\top(\boldsymbol{I} - P_{\boldsymbol{X}_1})\boldsymbol{X}_2\right)^{-1}\boldsymbol{X}_2^\top P_{\boldsymbol{X}_1}\right]\boldsymbol{v}\right],
\end{aligned}
\tag{7}
$$

where $(i)$ follows from Lemma B.2. Now, we consider

$$
\begin{aligned}
&\mathbb{E}\left[\boldsymbol{v}^\top\left[P_{\boldsymbol{X}_1}\boldsymbol{X}_2\left(\boldsymbol{X}_2^\top(\boldsymbol{I} - P_{\boldsymbol{X}_1})\boldsymbol{X}_2\right)^{-1}\boldsymbol{X}_2^\top P_{\boldsymbol{X}_1}\right]\boldsymbol{v}\right] \\
&= \mathbb{E}\left[\mathrm{tr}\left(\boldsymbol{X}_2^\top P_{\boldsymbol{X}_1}\boldsymbol{v}\boldsymbol{v}^\top P_{\boldsymbol{X}_1}\boldsymbol{X}_2\left(\boldsymbol{X}_2^\top(\boldsymbol{I} - P_{\boldsymbol{X}_1})\boldsymbol{X}_2\right)^{-1}\right)\right] \\
&\overset{(i)}{=} \mathbb{E}_{X_1}\left[\mathrm{tr}\left(\mathbb{E}_{X_2}\left[\boldsymbol{X}_2^\top P_{\boldsymbol{X}_1}\boldsymbol{v}\boldsymbol{v}^\top P_{\boldsymbol{X}_1}\boldsymbol{X}_2\right]\cdot\mathbb{E}_{X_2}\left[\left(\boldsymbol{X}_2^\top(\boldsymbol{I} - P_{\boldsymbol{X}_1})\boldsymbol{X}_2\right)^{-1}\right]\right)\right] \\
&\overset{(ii)}{=} \mathbb{E}_{X_1}\left[\mathrm{tr}\left(\|P_{\boldsymbol{X}_1}\boldsymbol{v}\|^2\cdot\boldsymbol{I}\cdot\mathbb{E}_{X_2}\left[\left(\widetilde{\boldsymbol{X}}_2^\top(\boldsymbol{I} - P_{\boldsymbol{X}_1})\widetilde{\boldsymbol{X}}_2\right)^{-1}\right]\right)\right]
\end{aligned}
$$

$$
= \mathbb{E}_{X_1}\left[\|P_{X_1}v\|^2 \cdot \mathrm{tr}\left(\mathbb{E}_{X_2}\left[\left(X_2^\top(I - P_{X_1})X_2\right)^{-1}\right]\right)\right]
$$

$$
\overset{(iii)}{=} \mathbb{E}_{X_1}\left[\|P_{X_1}v\|^2 \cdot \frac{m_2}{p - m_1 - m_2 - 1}\right]
$$

$$
\overset{(iv)}{=} \frac{m_2}{p - m_1 - m_2 - 1} \cdot \frac{m_1}{p}\|v\|^2, \tag{8}
$$

where $(i)$ follows from Lemma B.5, $(ii)$ follows from Lemma B.6, $(iii)$ follows from the fact that Lemma B.7 actually holds for any $X_2$ and $(iv)$ follows from Lemma B.2. By combining Equations (7) and (8), we complete the proof. $\qquad\square$

**Lemma B.9.** *Suppose each element of random matrices $X_1 \in \mathbb{R}^{p \times m_1}$, $X_2 \in \mathbb{R}^{p \times m_2}$, $X_3 \in \mathbb{R}^{p \times m_3}$ follows i.i.d. standard Gaussian distribution and $v \in \mathbb{R}^p$. Denote $V = [X_1, X_2, X_3]$, then we have:*

$$
\mathbb{E}\left[v^\top \begin{bmatrix} X_1 & 0 & 0 \end{bmatrix}(V^\top V)^{-1}\begin{bmatrix} 0 \\ X_2^\top \\ 0 \end{bmatrix}v\right] = -\frac{m_1 m_2}{p(p - m_1 - m_2 - m_3 - 1)}\|v\|^2
$$

*Proof.* First of all, we observe that:

$$
2v^\top\begin{bmatrix} X_1 & 0 & 0 \end{bmatrix}(V^\top V)^{-1}\begin{bmatrix} 0 \\ X_2^\top \\ 0 \end{bmatrix}v = \left\|V^\dagger\begin{bmatrix} X_1^\top \\ X_2^\top \\ 0 \end{bmatrix}v\right\|^2 - \left\|V^\dagger\begin{bmatrix} X_1^\top \\ 0 \\ 0 \end{bmatrix}v_1\right\|^2 - \left\|V^\dagger\begin{bmatrix} 0^\top \\ X_2 \\ 0 \end{bmatrix}v\right\|^2.
$$

By taking expectation over both sides of the equation, we have:

$$
2\mathbb{E}\left[v^\top\begin{bmatrix} X_1 & 0 & 0 \end{bmatrix}(V^\top V)^{-1}\begin{bmatrix} 0 \\ X_2^\top \\ 0 \end{bmatrix}v\right]
$$

$$
= \mathbb{E}\left\|V^\dagger\begin{bmatrix} X_1^\top \\ X_2^\top \\ 0 \end{bmatrix}v\right\|^2 - \mathbb{E}\left\|V^\dagger\begin{bmatrix} X_1^\top \\ 0 \\ 0 \end{bmatrix}v\right\|^2 - \mathbb{E}\left\|V^\dagger\begin{bmatrix} 0^\top \\ X_2 \\ 0 \end{bmatrix}v\right\|^2
$$

$$
\overset{(i)}{=} \frac{m_1 + m_2}{p}\cdot\left(1 + \frac{m_3}{p - m_1 - m_2 - m_3 - 1}\right)\|v\|^2 - \frac{m_1}{p}\cdot\left(1 + \frac{m_2 + m_3}{p - m_1 - m_2 - m_3 - 1}\right)\|v\|^2
$$

$$
- \frac{n_2}{p}\cdot\left(1 + \frac{m_1 + m_3}{p - m_1 - m_2 - m_3 - 1}\right)\|v\|^2
$$

$$
= -\frac{2m_1 m_2}{p(p - m_1 - m_2 - m_3 - 1)}\|v\|^2,
$$

where $(i)$ follows from Lemma B.8. By dividing both sides by 2, we complete the proof. $\qquad\square$

**Corollary B.10.** *Suppose each element of random matrices $X_1 \in \mathbb{R}^{p \times m_1}$, $X_2 \in \mathbb{R}^{p \times m_2}$, $X_3 \in \mathbb{R}^{p \times m_3}$ follows i.i.d. standard Gaussian distribution and $v_1, v_2 \in \mathbb{R}^p$. Denote $V = [X_1, X_2, X_3]$, then we have:*

$$
\mathbb{E}\left[v_1^\top\begin{bmatrix} X_1 & 0 & 0 \end{bmatrix}(V^\top V)^{-1}\begin{bmatrix} 0 \\ X_2^\top \\ 0 \end{bmatrix}v_2\right] = \frac{m_1 m_2\left(\|v_1 - v_2\|^2 - \|v_1\|^2 - \|v_2\|^2\right)}{2p(p - m_1 - m_2 - m_3 - 1)}
$$

*Proof.* To simplify the notation, we denote $V_1 = \begin{bmatrix} X_1 & 0 & 0 \end{bmatrix}$ and $V_2 = \begin{bmatrix} 0 & X_2 & 0 \end{bmatrix}$. Then according to Lemma B.9, we first have:

$$
\mathbb{E}\left[(v_1 - v_2)^\top V_1 (V^\top V)^{-1} V_2^\top (v_1 - v_2)\right] = -\frac{m_1 m_2}{p(p - m_1 - m_2 - m_3 - 1)}\|v_1 - v_2\|^2.
$$

On the other hand, we have:

$$
\mathbb{E}\left[(v_1 - v_2)^\top V_1 (V^\top V)^{-1} V_2^\top (v_1 - v_2)\right]
$$

$$= \mathbb{E}\left[\boldsymbol{v}_1^\top \boldsymbol{V}_1 (\boldsymbol{V}^\top \boldsymbol{V})^{-1} \boldsymbol{V}_2^\top \boldsymbol{v}_1\right] + \mathbb{E}\left[\boldsymbol{v}_2^\top \boldsymbol{V}_1 (\boldsymbol{V}^\top \boldsymbol{V})^{-1} \boldsymbol{V}_2^\top \boldsymbol{v}_2\right] - 2\mathbb{E}\left[\boldsymbol{v}_1^\top \boldsymbol{V}_1 (\boldsymbol{V}^\top \boldsymbol{V})^{-1} \boldsymbol{V}_2^\top \boldsymbol{v}_2\right]$$

$$\overset{(i)}{=} -\frac{m_1 m_2 \|\boldsymbol{v}_1\|^2}{p(p - m_1 - m_2 - m_3 - 1)} - \frac{m_1 m_2 \|\boldsymbol{v}_2\|^2}{p(p - m_1 - m_2 - m_3 - 1)} - 2\mathbb{E}\left[\boldsymbol{v}_1^\top \boldsymbol{V}_1 (\boldsymbol{V}^\top \boldsymbol{V})^{-1} \boldsymbol{V}_2^\top \boldsymbol{v}_2\right],$$

where $(i)$ follows from Lemma B.9. By combining the above two equations, we complete the proof. $\qquad\square$

Next, we provide our supporting lemmas that help to prove the advantage of sequential rehearsal as follows.

**Lemma B.11.** *Suppose $n, p, t, M$ are positive integers where $t \geq 2$ and $p > n + M$. For any non-negative integer $l < t - 1$, we have:*

$$\left(1 - \frac{M}{(t - l - 1)p}\right)^{t - l - 1} \left(1 - \frac{n}{p}\right) > 1 - \frac{n + M}{p}.$$

*Proof.* We first note the fact that

$$\left(1 - \frac{M}{(t - l - 1)p}\right)^{t - l - 1} > \left(1 - \frac{(t - l - 1)M}{(t - l - 1)p}\right) = 1 - \frac{M}{p}.$$

Therefore, we have

$$\left(1 - \frac{M}{(t - l - 1)p}\right)^{t - l - 1} \left(1 - \frac{n}{p}\right) > \left(1 - \frac{M}{p}\right) \left(1 - \frac{n}{p}\right) = 1 - \frac{n + M}{p}.$$

$$\square$$

**Lemma B.12.** *Suppose $n, p, t, M$ are positive integers where $t \geq 2$ and $p > \max\{n + M, TM\}$. For any non-negative integer $l < t - 1$, we have:*

$$\left(1 - \frac{M}{(t - l - 1)p}\right)^{t - l - 1} \left(1 - \frac{n}{p}\right) < 1 - \frac{n + M}{p} + \frac{(n + M)M}{p^2}.$$

*Proof.* If $t - l - 1 = 1$ or $t - l - 1 = 2$, the proof is trivial. If $t - l - 1 \geq 3$, according to the binomial theorem, we have:

$$\left(1 - \frac{M}{(t - l - 1)p}\right)^{t - l - 1} \left(1 - \frac{n}{p}\right) = \left(1 - \frac{M}{p} + \sum_{k=2}^{t - l - 1} \binom{t - l - 1}{k} \left(-\frac{M}{(t - l - 1)p}\right)^k\right) \left(1 - \frac{n}{p}\right), \qquad (9)$$

where

$$\sum_{k=2}^{t-l-1} \binom{t-l-1}{k} \left(-\frac{M}{(t-l-1)p}\right)^k = \binom{t-l-1}{2} \left(\frac{M}{(t-l-1)p}\right)^2 + \sum_{k=3}^{t-l-1} \binom{t-l-1}{k} \left(-\frac{M}{(t-l-1)p}\right)^k \quad (10)$$

To simplify the notation, we denote $m = \frac{M}{t-l-1}$. We first discuss if $t - l - 1$ is even. Then, we have:

$$\sum_{k=3}^{t-l-1} \binom{t-l-1}{k} \left(-\frac{M}{(t-l-1)p}\right)^k$$

$$= \sum_{k=3}^{(t-l+1)/2} \left[\binom{t-l-1}{2k-3} \left(-\frac{m}{p}\right)^{2k-3} + \binom{t-l-1}{2k-2} \left(-\frac{m}{p}\right)^{2k-2}\right]$$

$$= \sum_{k=3}^{(t-l+1)/2} \left[\frac{(t-l-1)!}{(2k-3)!(t-l-2k+2)!} \left(-\frac{m}{p}\right)^{2k-3} + \frac{(t-l-1)!}{(2k-2)!(t-l-2k+1)!} \left(-\frac{m}{p}\right)^{2k-2}\right]$$

$$= -\sum_{k=3}^{(t-l+1)/2} \frac{(t-l-1)!}{(2k-3)!(t-l-2k+1)!} \left(\frac{m}{p}\right)^{2k-3} \left[\frac{1}{t-l-2k+2} - \frac{1}{2k-2} \cdot \frac{m}{p}\right]$$

$$\overset{(i)}{<} 0 \tag{11}$$

where $(i)$ follows from the fact that $p > TM$. We then discuss if $t - l - 1$ is odd, we have:

$$\sum_{k=3}^{t-l-1} \binom{t-l-1}{k} \left(-\frac{M}{(t-l-1)p}\right)^k$$

$$= \sum_{k=3}^{(t-l)/2} \left[\binom{t-l-1}{2k-3}\left(-\frac{m}{p}\right)^{2k-3} + \binom{t-l-1}{2k-2}\left(-\frac{m}{p}\right)^{2k-2}\right] + \left(-\frac{m}{p}\right)^{t-l-1}$$

$$\overset{(i)}{<} \sum_{k=3}^{(t-l)/2} \left[\frac{(t-l-1)!}{(2k-3)!(t-l-2k+2)!}\left(-\frac{m}{p}\right)^{2k-3} + \frac{(t-l-1)!}{(2k-2)!(t-l-2k+1)!}\left(-\frac{m}{p}\right)^{2k-2}\right]$$

$$= -\sum_{k=3}^{(t-l)/2} \frac{(t-l-1)!}{(2k-3)!(t-l-2k+1)!}\left(\frac{m}{p}\right)^{2k-3}\left[\frac{1}{t-l-2k+2} - \frac{1}{2k-2}\cdot\frac{m}{p}\right]$$

$$\overset{(ii)}{<} 0 \tag{12}$$

where $(i)$ follows from the fact that $t - l - 1$ is odd and $(ii)$ follows from the fact that $p > TM$. By combing Equations (9) to (12), we conclude:

$$\left(1 - \frac{M}{(t-l-1)p}\right)^{t-l-1}\left(1 - \frac{n}{p}\right)$$

$$< \left(1 - \frac{M}{p} + \binom{t-l-1}{2}\frac{M^2}{(t-l-1)^2p^2}\right)\left(1 - \frac{n}{p}\right)$$

$$= 1 - \frac{n+M}{p} + \frac{nM + \frac{(t-l-1)(t-l-2)}{2}\frac{M^2}{(t-l-1)^2}}{p^2} - \binom{t-l-1}{2}\frac{nM^2}{(t-l-1)^2p^3}$$

$$< 1 - \frac{n+M}{p} + \frac{(n+M)M}{p^2}.$$

which completes the proof. $\qquad\square$

**Lemma B.13.** *Suppose $n, p, t, M, T$ are positive integers where $t \leq T$ and $p > n + M$, then we have:*

$$\left(1 - \frac{n+M}{p} + \frac{(n+M)M}{p^2}\right)^t < \left(1 - \frac{n+M}{p}\right)^t + \frac{T^2(n+M)M}{p^2}.$$

*Proof.* According to the binomial theorem, we have:

$$\left(1 - \frac{n+M}{p} + \frac{(n+M)M}{p^2}\right)^t = \left(1 - \frac{n+M}{p}\right)^t + \sum_{k=0}^{t-1}\underbrace{\binom{t}{k}\left(1 - \frac{n+M}{p}\right)^k\left(\frac{(n+M)M}{p^2}\right)^{t-k}}_{\alpha_k} \tag{13}$$

We further notice that for $k = 0, 1, .., t-2$:

$$\binom{t}{k}\left(1 - \frac{n+M}{p}\right)^k\left(\frac{(n+M)M}{p^2}\right)^{t-k} - \binom{t}{k+1}\left(1 - \frac{n+M}{p}\right)^{k+1}\left(\frac{(n+M)M}{p^2}\right)^{t-k-1}$$

$$= \frac{t!}{k!(t-k-1)!}\left(1 - \frac{n+M}{p}\right)^k\left(\frac{(n+M)M}{p^2}\right)^{t-k-1}\left[\frac{(n+M)M}{(t-k)p^2} - \frac{1}{k+1}\left(1 - \frac{n+M}{p}\right)\right]$$

$$< \frac{t!}{k!(t-k-1)!}\left(1 - \frac{n+M}{p}\right)^k\left(\frac{(n+M)M}{p^2}\right)^{t-k-1}\left[\frac{(n+M)M}{p^2} - \frac{1}{T}\left(1 - \frac{n+M}{p}\right)\right]$$

$$\overset{(i)}{<} 0, \tag{14}$$

where $(i)$ follows from the fact that $p > (n + M)(T + 1)$. We note that Equation (14) shows that the term $\alpha_k$ achieves the maximum at $k = t - 1$. Therefore, we can upper bound Equation (13) by

$$\left(1 - \frac{n+M}{p} + \frac{(n+M)M}{p^2}\right)^t < \left(1 - \frac{n+M}{p}\right)^t + t\binom{t}{t-1}\left(1 - \frac{n+M}{p}\right)^{t-1}\left(\frac{(n+M)M}{p^2}\right)$$

$$< \left(1 - \frac{n+M}{p}\right)^t + \frac{T^2(n+M)M}{p^2},$$

which completes the proof. $\qquad\square$

Here, we present a tighter version of Lemma B.13, which helps us to prove Theorem 5.5 in Section 5.2.

**Lemma B.14.** *Suppose $n, p, t, M, T$ are fixed positive integers where $t \leq T$ and $p > n + M$, then we have:*

$$\left(1 - \frac{n+M}{p} + \frac{(n+M)M}{p^2}\right)^t < \left(1 - \frac{n+M}{p}\right)^t + \frac{t(n+M)M}{p^2} + \frac{T^3(n+M)^2M^2}{2p^4}.$$

*Proof.* According to the binomial theorem, we have:

$$\left(1 - \frac{n+M}{p} + \frac{(n+M)M}{p^2}\right)^t$$

$$= \left(1 - \frac{n+M}{p}\right)^t + \binom{t}{t-1}\left(1 - \frac{n+M}{p}\right)^{t-1}\left(\frac{(n+M)M}{p^2}\right) + \sum_{k=0}^{t-2}\binom{t}{k}\left(1 - \frac{n+M}{p}\right)^k\left(\frac{(n+M)M}{p^2}\right)^{t-k}$$

$$< \left(1 - \frac{n+M}{p}\right)^t + \frac{T(n+M)M}{p^2} + \sum_{k=0}^{t-2}\underbrace{\binom{t}{k}\left(1 - \frac{n+M}{p}\right)^k\left(\frac{(n+M)M}{p^2}\right)^{t-k}}_{\alpha_k} \qquad (15)$$

By the same argument as Equation (14), we know that the term $\alpha_k$ achieves the maximum at $k = t - 2$. Therefore, we can upper bound Equation (15) by

$$\left(1 - \frac{n+M}{p} + \frac{(n+M)M}{p^2}\right)^t$$

$$< \left(1 - \frac{n+M}{p}\right)^t + \frac{t(n+M)M}{p^2} + (t-1)\binom{t}{t-2}\left(1 - \frac{n+M}{p}\right)^{t-2}\left(\frac{(n+M)M}{p^2}\right)^2$$

$$< \left(1 - \frac{n+M}{p}\right)^t + \frac{t(n+M)M}{p^2} + \frac{T^3(n+M)^2M^2}{2p^4},$$

which completes the proof. $\qquad\square$

**Lemma B.15.** *Suppose $n, p, t, M, T$ are positive integers where $M \geq 2$, $t \leq T$ and $p > \max\{n+M, \frac{T(n+M)M}{M-1} + n + M\}$. For any non-negative integer $l < t - 1$, we have:*

$$\left(1 - \frac{n+M}{p} + \frac{(n+M)M}{p^2}\right)^l\left(1 - \frac{M}{(t-l-1)p}\right)^{t-l-1} < \left(1 - \frac{1}{Tp}\right)\left(1 - \frac{n+M}{p}\right)^l.$$

*Proof.* By dividing $\left(1 - \frac{n+M}{p}\right)^l$ on both sides, it is equivalent to prove

$$\left(1 + \frac{(n+M)M}{p^2 - p(n+M)}\right)^l\left(1 - \frac{M}{(t-l-1)p}\right)^{t-l-1} < 1 - \frac{1}{Tp}.$$

According to AM-GM inequality, we have:

$$\left(1 + \frac{(n+M)M}{p^2 - p(n+M)}\right)^l\left(1 - \frac{M}{(t-l-1)p}\right)^{t-l-1} \leq \left[\frac{l\left(1 + \frac{(n+M)M}{p^2 - p(n+M)}\right) + (t-l-1)\left(1 - \frac{M}{(t-l-1)p}\right)}{t-1}\right]^{t-1}$$

$$= \left[ 1 + \frac{\frac{l(n+M)M}{p^2 - p(n+M)} - \frac{M}{p}}{t-1} \right]^{t-1}. \tag{16}$$

When $p > \frac{T(n+M)M}{M-1} + n + M$, we have:

$$\frac{l(n+M)M}{p^2 - p(n+M)} - \frac{M}{p} < \frac{T(n+M)M}{p^2 - p(n+M)} - \frac{M}{p} < -\frac{1}{p}. \tag{17}$$

Therefore, by combining Equations (16) and (17), we have:

$$\left( 1 + \frac{(n+M)M}{p^2 - p(n+M)} \right)^l \left( 1 - \frac{M}{(t-l-1)p} \right)^{t-l} < \left( 1 - \frac{1}{(t-1)p} \right)^{t-1} < 1 - \frac{1}{(t-1)p} < 1 - \frac{1}{Tp},$$

which completes the proof. $\qquad\qquad\qquad\qquad\qquad\qquad\qquad\qquad\qquad\qquad\qquad\qquad\qquad\square$

**Lemma B.16.** *Suppose $n, p, t, M, T$ are positive integers where $t \le T$ and $p > \max\{n + M, 2T^3(n+M)^2\}$, then we have:*

$$\prod_{l=0}^{t-2} \left[ \left( 1 - \frac{M}{(t-l-1)p} \right)^{t-l-1} \left( 1 - \frac{n}{p} \right) \right] - \prod_{l=0}^{i-2} \left[ \left( 1 - \frac{M}{(i-l-1)p} \right)^{i-l-1} \left( 1 - \frac{n}{p} \right) \right]$$
$$> \left[ \left( 1 - \frac{n+M}{p} \right)^{t-1} - \left( 1 - \frac{n+M}{p} \right)^{i-1} \right].$$

*Proof.* To prove this lemma, we first have:

$$\prod_{l=0}^{t-2} \left[ \left( 1 - \frac{M}{(t-l-1)p} \right)^{t-l-1} \left( 1 - \frac{n}{p} \right) \right] - \prod_{l=0}^{i-2} \left[ \left( 1 - \frac{M}{(i-l-1)p} \right)^{i-l-1} \left( 1 - \frac{n}{p} \right) \right]$$

$$= \prod_{l=0}^{t-i-1} \left[ \left( 1 - \frac{M}{(t-l-1)p} \right)^{t-l-1} \left( 1 - \frac{n}{p} \right) \right] \prod_{l=t-i}^{t-2} \left[ \left( 1 - \frac{M}{(t-l-1)p} \right)^{t-l-1} \left( 1 - \frac{n}{p} \right) \right]$$
$$- \prod_{l=0}^{i-2} \left[ \left( 1 - \frac{M}{(i-l-1)p} \right)^{i-l-1} \left( 1 - \frac{n}{p} \right) \right]$$

$$= \prod_{l=0}^{t-i-1} \left[ \left( 1 - \frac{M}{(t-l-1)p} \right)^{t-l-1} \left( 1 - \frac{n}{p} \right) \right] \prod_{l=0}^{i-2} \left[ \left( 1 - \frac{M}{(i-l-1)p} \right)^{i-l-1} \left( 1 - \frac{n}{p} \right) \right]$$
$$- \prod_{l=0}^{i-2} \left[ \left( 1 - \frac{M}{(i-l-1)p} \right)^{i-l-1} \left( 1 - \frac{n}{p} \right) \right]$$

$$= \prod_{l=0}^{i-2} \left[ \left( 1 - \frac{M}{(i-l-1)p} \right)^{i-l-1} \left( 1 - \frac{n}{p} \right) \right] \underbrace{\left\{ \prod_{l=0}^{t-i-1} \left[ \left( 1 - \frac{M}{(t-l-1)p} \right)^{t-l-1} \left( 1 - \frac{n}{p} \right) \right] - 1 \right\}}_{\gamma_1}$$

$$\overset{(i)}{>} \left( 1 - \frac{n+M}{p} + \frac{(n+M)M}{p^2} \right)^{i-1} \left\{ \left[ \left( 1 - \frac{M}{p} \right) \left( 1 - \frac{n}{p} \right) \right]^{t-i} - 1 \right\}$$

$$= \left( 1 - \frac{n+M}{p} + \frac{(n+M)M}{p^2} \right)^{i-1} \left[ \left( 1 - \frac{n+M}{p} + \frac{nM}{p^2} \right)^{t-i} - 1 \right]$$

$$= \left[ \left( 1 - \frac{n+M}{p} \right)^{i-1} + \sum_{k=1}^{i-1} \binom{i-1}{k} \left( \frac{(n+M)M}{p^2} \right)^k \left( 1 - \frac{n+M}{p} \right)^{i-k-1} \right]$$
$$\left[ \left( 1 - \frac{n+M}{p} \right)^{t-i} - 1 + \sum_{k=1}^{t-i} \binom{t-i}{k} \left( \frac{nM}{p^2} \right)^k \left( 1 - \frac{n+M}{p} \right)^{t-i-k} \right]$$

$$> \left(1 - \frac{n+M}{p}\right)^{i-1} \left[\left(1 - \frac{n+M}{p}\right)^{t-i} - 1\right]$$

$$+ \underbrace{\left(1 - \frac{n+M}{p}\right)^{i-1} \sum_{k=1}^{t-i} \binom{t-i}{k} \left(\frac{nM}{p^2}\right)^k \left(1 - \frac{n+M}{p}\right)^{t-i-k}}_{\gamma_2}$$

$$+ \underbrace{\left[\left(1 - \frac{n+M}{p}\right)^{t-i} - 1\right] \sum_{k=1}^{i-1} \underbrace{\binom{i-1}{k} \left(\frac{(n+M)M}{p^2}\right)^k \left(1 - \frac{n+M}{p}\right)^{i-k-1}}_{\gamma_{3,k}}}_{\gamma_3} \tag{18}$$

where $(i)$ follows from Lemma B.12 together with the fact that term $\gamma_1 < 0$ and from the fact that $\left(1 - \frac{M}{(t-l-1)p}\right)^{t-l-1} > 1 - \frac{M}{p}$ for $l = 0, 1, .., t - i - 1$. Now. we need to prove $\gamma_2 + \gamma_3 > 0$. We first focus on $\gamma_2$. We have:

$$\gamma_2 > \left(1 - \frac{n+M}{p}\right)^{i-1} \binom{t-i}{1} \left(\frac{nM}{p^2}\right) \left(1 - \frac{n+M}{p}\right)^{t-i-1}$$

$$> \left(1 - \frac{n+M}{p}\right)^T \frac{nM}{p^2}$$

$$> \left(1 - \frac{T(n+M)}{p}\right) \frac{nM}{p^2} \tag{19}$$

We then focus on term $\gamma_3$. Consider:

$$\binom{i-1}{k} \left(\frac{(n+M)M}{p^2}\right)^k \left(1 - \frac{n+M}{p}\right)^{i-k-1} - \binom{i-1}{k+1} \left(\frac{(n+M)M}{p^2}\right)^{k+1} \left(1 - \frac{n+M}{p}\right)^{i-k-2}$$

$$= \frac{(i-1)!}{k!(i-k-2)!} \left(\frac{(n+M)M}{p^2}\right)^k \left(1 - \frac{n+M}{p}\right)^{i-k-2} \left[\frac{1}{i-k-1}\left(1 - \frac{n+M}{p}\right) - \frac{1}{k+1}\frac{(n+M)M}{p^2}\right]$$

$$\overset{(i)}{>} \frac{(i-1)!}{k!(i-k-2)!} \left(\frac{(n+M)M}{p^2}\right)^k \left(1 - \frac{n+M}{p}\right)^{i-k-2} \left[\frac{1}{T}\left(1 - \frac{n+M}{p}\right) - \frac{(n+M)M}{2p^2}\right]$$

$$\overset{(ii)}{>} 0, \tag{20}$$

where $(i)$ follows from $k \in [i-1]$ and $(ii)$ follows from the fact that $p > 2(n+M)$. This indicates that $\gamma_{3,k}$ achieves maximum at $k = 1$. We recall that $\left[\left(1 - \frac{n+M}{p}\right)^{t-i} - 1\right] < 0$. Therefore, we have:

$$\gamma_3 > \left[\left(1 - \frac{n+M}{p}\right)^{t-i} - 1\right](i-1)\binom{i-1}{1}\left(\frac{(n+M)M}{p^2}\right)\left(1 - \frac{n+M}{p}\right)^{i-2}$$

$$> \left[\left(1 - \frac{n+M}{p}\right)^{t-i} - 1\right]\frac{T^2(n+M)M}{p^2}$$

$$= \left[\sum_{k=1}^{t-i}\binom{t-i}{k}\left(-\frac{n+M}{p}\right)^k\right]\frac{T^2(n+M)M}{p^2}. \tag{21}$$

When $k$ is even and less than or equal to $t - i$ (i.e., $k = 2, 4, 6, ...,$ and $k \le t - i$), we have:

$$\binom{t-i}{k}\left(-\frac{n+M}{p}\right)^k + \binom{t-i}{k+1}\left(-\frac{n+M}{p}\right)^{k+1} = \frac{(t-i)!}{k!(t-i-k-1)!}\left(\frac{n+M}{p}\right)^k\left[\frac{1}{t-i-k} - \frac{n+M}{(k+1)p}\right]$$

$$> \frac{(t-i)!}{k!(t-i-k-1)!}\left(\frac{n+M}{p}\right)^k\left[\frac{1}{T} - \frac{n+M}{3p}\right]$$

$$\overset{(i)}{>} 0, \tag{22}$$

where $(i)$ follows from $p > \frac{(n+M)T}{3}$. By combining Equations (21) and (22) and simply discussing when $t - i$ is odd or even, we can conclude

$$\sum_{k=1}^{t-i} \binom{t-i}{k} \left(-\frac{n+M}{p}\right)^k > \binom{t-i}{1}\left(-\frac{n+M}{p}\right) > -\frac{T(n+M)}{p},$$

which further implies:

$$\gamma_3 > -\frac{T^3(n+M)^2 M}{p^3}. \tag{23}$$

Now, by combining Equations (18), (19) and (23), we have:

$$\prod_{l=0}^{t-2}\left[\left(1-\frac{M}{(t-l-1)p}\right)^{t-l-1}\left(1-\frac{n}{p}\right)\right] - \prod_{l=0}^{i-2}\left[\left(1-\frac{M}{(i-l-1)p}\right)^{i-l-1}\left(1-\frac{n}{p}\right)\right]$$

$$> \left(1-\frac{n+M}{p}\right)^{i-1}\left[\left(1-\frac{n+M}{p}\right)^{t-i}-1\right] + \left(1-\frac{T(n+M)}{p}\right)\frac{nM}{p^2} - \frac{T^3(n+M)^2 M}{p^3}$$

$$\overset{(i)}{>} \left(1-\frac{n+M}{p}\right)^{i-1}\left[\left(1-\frac{n+M}{p}\right)^{t-i}-1\right], \tag{24}$$

where $(i)$ follows from the fact that $p > 2T^3(n+M)^2$. $\qquad\square$

**Lemma B.17.** *Suppose $n, p, t, M, T$ are positive integers where $t \le T$ and $p > (n+M)T$. For any non-negative integer $i < t$, we have:*

$$\prod_{l=0}^{t-2}\left[\left(1-\frac{M}{(t-l-1)p}\right)^{t-l-1}\left(1-\frac{n}{p}\right)\right] - \prod_{l=0}^{i-2}\left[\left(1-\frac{M}{(i-l-1)p}\right)^{i-l-1}\left(1-\frac{n}{p}\right)\right]$$

$$< \left(1-\frac{n+M}{p}\right)^{i-1}\left[\left(1-\frac{n+M}{p}\right)^{t-i}-1\right] + \frac{T^2(n+M)M}{p^2}.$$

*Proof.* To prove this lemma, We consider:

$$\prod_{l=0}^{t-2}\left[\left(1-\frac{M}{(t-l-1)p}\right)^{t-l-1}\left(1-\frac{n}{p}\right)\right] - \prod_{l=0}^{i-2}\left[\left(1-\frac{M}{(i-l-1)p}\right)^{i-l-1}\left(1-\frac{n}{p}\right)\right]$$

$$= \prod_{l=0}^{t-i-1}\left[\left(1-\frac{M}{(t-l-1)p}\right)^{t-l-1}\left(1-\frac{n}{p}\right)\right]\prod_{l=t-i}^{t-2}\left[\left(1-\frac{M}{(t-l-1)p}\right)^{t-l-1}\left(1-\frac{n}{p}\right)\right]$$

$$\qquad - \prod_{l=0}^{i-2}\left[\left(1-\frac{M}{(i-l-1)p}\right)^{i-l-1}\left(1-\frac{n}{p}\right)\right]$$

$$= \prod_{l=0}^{t-i-1}\left[\left(1-\frac{M}{(t-l-1)p}\right)^{t-l-1}\left(1-\frac{n}{p}\right)\right]\prod_{l=0}^{i-2}\left[\left(1-\frac{M}{(i-l-1)p}\right)^{i-l-1}\left(1-\frac{n}{p}\right)\right]$$

$$\qquad - \prod_{l=0}^{i-2}\left[\left(1-\frac{M}{(i-l-1)p}\right)^{i-l-1}\left(1-\frac{n}{p}\right)\right]$$

$$= \prod_{l=0}^{i-2}\left[\left(1-\frac{M}{(i-l-1)p}\right)^{i-l-1}\left(1-\frac{n}{p}\right)\right]\underbrace{\left\{\prod_{l=0}^{t-i-1}\left[\left(1-\frac{M}{(t-l-1)p}\right)^{t-l-1}\left(1-\frac{n}{p}\right)\right]-1\right\}}_{\gamma_1}$$

$$\overset{(i)}{<} \left(1 - \frac{n+M}{p}\right)^{i-1} \left[\left(1 - \frac{n+M}{p} + \frac{(n+M)M}{p^2}\right)^{t-i} - 1\right],$$

$$\overset{(ii)}{<} \left(1 - \frac{n+M}{p}\right)^{i-1} \left[\left(1 - \frac{n+M}{p}\right)^{t-i} - 1 + \frac{T^2(n+M)M}{p^2}\right]$$

$$< \left(1 - \frac{n+M}{p}\right)^{i-1} \left[\left(1 - \frac{n+M}{p}\right)^{t-i} - 1\right] + \frac{T^2(n+M)M}{p^2} \tag{25}$$

where $(i)$ follows from Lemmas B.11 and B.12 and the fact that $\gamma_1 < 0$; $(ii)$ follows from Lemma B.13. $\qquad\square$

Here, we present a tighter version of Lemma B.17, which helps to prove Theorem 5.5 in Section 5.2.

**Lemma B.18.** *Suppose $n, p, t, M, T$ are positive integers where $t \leq T$ and $p > (n+M)T$. For any non-negative integer $i < t$, we have:*

$$\prod_{l=0}^{t-2} \left[\left(1 - \frac{M}{(t-l-1)p}\right)^{t-l-1} \left(1 - \frac{n}{p}\right)\right] - \prod_{l=0}^{i-2} \left[\left(1 - \frac{M}{(i-l-1)p}\right)^{i-l-1} \left(1 - \frac{n}{p}\right)\right]$$

$$< \left(1 - \frac{n+M}{p}\right)^{i-1} \left[\left(1 - \frac{n+M}{p}\right)^{t-i} - 1\right] + \frac{(t-i)(n+M)M}{p^2} + \frac{T^3(n+M)^2 M^2}{p^4}.$$

*Proof.* The proof follows from the same argument as Lemma B.17 but we use Lemma B.14 instead of Lemma B.13. $\qquad\square$

## C. Proof of Theorem 5.1

First of all, we present the detailed iteration formula for the proof outline of Theorem 5.1 in Section 5.1. For both rehearsal methods, we have:

$$\mathbb{E}[\mathcal{L}_i(\boldsymbol{w}_t)] = g_t(\mathbb{E}[\mathcal{L}_i(\boldsymbol{w}_{t-1})]) + \text{term}_2 + \text{term}_{\text{noise}}, \tag{26}$$

where $g_t^{\text{(concurrent)}}(x) = (1 - \frac{n+M}{p})x$, $g_t^{\text{(sequential)}}(x) = (1 - \frac{M}{(t-1)p})^{t-1}(1 - \frac{n}{p})x$, $\text{term}_{\text{noise}}^{\text{(concurrent)}} = \frac{(n+M)\sigma^2}{p-n-M-1}$, $\text{term}_{\text{noise}}^{\text{(sequential)}} = \sum_{j=1}^{t-1} \left(1 - \frac{M}{(t-1)p}\right)^{t-j-1} \frac{\frac{M}{t-1}\sigma^2}{p - \frac{M}{t-1} - 1} + \left(1 - \frac{M}{(t-1)p}\right)^{t-1} \frac{n\sigma^2}{p-n-1}$, and the form of $\text{term}_2$ is given in Equations (29) and (33) for concurrent rehearsal and sequential rehearsal, respectively.

Based on the above iteration formula, we further derive the explicit expressions of $\mathbb{E}[\mathcal{L}_i(\boldsymbol{w}_t)]$ and $\mathbb{E}[\mathcal{L}_i(\boldsymbol{w}_t) - \mathcal{L}_i(\boldsymbol{w}_i)]$, and they share the same structure for both rehearsal methods for any $t \leq T$ which is formally presented in the following lemma.

**Lemma C.1.** *Denote $\boldsymbol{w}_t$ as the parameters of training result at task $t$. Under the problem setups considered in this work, the expected value of the model error $\mathcal{L}_i(\boldsymbol{w}_t)$ in both rehearsal-based methods take the following forms.*

$$\mathbb{E}\mathcal{L}_i(\boldsymbol{w}_t) = d_{0t} \|\boldsymbol{w}_i^*\|^2 + \sum_{j,k=1}^{T-1} d_{ijkt} \|\boldsymbol{w}_j^* - \boldsymbol{w}_k^*\|^2 + noise_t(\sigma),$$

$$\mathbb{E}(\mathcal{L}_i(\boldsymbol{w}_t) - \mathcal{L}_i(\boldsymbol{w}_i)) = c_i \|\boldsymbol{w}_i^*\|^2 + \sum_{j,k=1}^{T-1} c_{ijk} \|\boldsymbol{w}_j^* - \boldsymbol{w}_k^*\|^2 + (noise_t(\sigma) - noise_i(\sigma)).$$

For both concurrent and sequential rehearsal method, Theorem 5.1 follows directly by combining Lemma C.1 and the definitions of $F_T$ and $G_T$. Therefore, it suffices to prove Lemma C.1 to obtain the coefficients in Theorem 5.1. Before we start our proof, we first present the explicit expressions of coefficients as well as the noise term in Theorem 5.1 for both concurrent and sequential rehearsal methods in the following proposition.

**Proposition C.2.** *Under the problem setups considered in this work, the coefficients that express forgetting $F_t$ and generalization error $G_t$ take the following forms.*

$$d_{0t}^{\text{(concurrent)}} = r_0 r_M^{t-1}, \qquad c_i^{\text{(concurrent)}} = d_{0T}^{\text{(concurrent)}} - d_{0i}^{\text{(concurrent)}},$$

$$d_{ijkt}^{(concurrent)} = \begin{cases} (1-r_0)r_M^{t-j-1} + \sum\limits_{l=0}^{t-j-1} r_M^l B_{l,t} + r_M^{t-k}nH_{t-k,t} + \sum\limits_{l=0}^{t-2} pr_M^l B_{l,t}H_{l,t} & \text{if } j \in [t-1], k=i \\ (1-r_0) + r_M^{t-k}nH_{t-k,t} & \text{if } j=t, k=i \\ \sum\limits_{l=0}^{t-2} pr_M^l B_{l,t}H_{l,t} & \text{if } j < k \text{ and } j,k \neq i,t \\ r_M^{t-k}nH_{t-k,t} & \text{if } j < k \text{ and } j,k \neq i \end{cases}$$

$$c_{ijk}^{(concurrent)} = d_{ijkT}^{(concurrent)} - d_{ijki}^{(concurrent)},$$

$$d_{0t}^{(sequential)} = r_0\Delta_t(t-1), \qquad c_i^{(sequential)} = d_{0T}^{(sequential)} - d_{0i}^{(concurrent)},$$

$$d_{ijkt}^{(sequential)} = \begin{cases} (1-r_0)(1-B_{t-j,t})^{j-1}\Delta_t(t-j) + \sum\limits_{l=0}^{t-j-1} \Delta_t(l)(1-B_{l,t})^{t-j-l-1}B_{l,t} & \text{if } j \in [t-1] \text{ and } k=i \\ (1-r_0)(1-B_{0,t})^{t-1} & \text{if } j=t, k=i \end{cases}$$

$$c_{ijk}^{(sequential)} = d_{ijkT}^{(sequential)} - d_{ijki}^{(sequential)},$$

$$noise_t^{(concurrent)}(\sigma) = r_0 r_M^{t-1}\Lambda_{n,\sigma} + \sum_{l=0}^{t-2} r_M^l \Lambda_{n+M,\sigma},$$

$$noise_t^{(sequential)}(\sigma) = \sum_{l=0}^{t-2} \Delta_t(l)\Big[(1-B_{0,t})^{t-1}\Lambda_{n,\sigma} + \sum_{l=1}^{t-1}(1-B_{0,t})^{t-l-1}\Lambda_{\frac{M}{t-1},\sigma}\Big].$$

where $r_a = 1 - \frac{n+a}{p}$, $B_{l,t} = \begin{cases} \frac{M}{(t-l-1)p} & \text{if } l \neq t-1 \\ 0 & o.w. \end{cases}$, $H_{l,t} = \frac{B_{l,t}}{p-n-M-1}$, $\Delta_t(a) = \prod\limits_{l=0}^{a-1}\Big[(1-B_{l,t})^{t-l-1}r_0\Big]$, $\Lambda_{a,\sigma} = \frac{a\sigma^2}{p-a-1}$.

To simplify notation, we omit the tilde notation of the memory data to simplify notations: $\boldsymbol{X}_{t,i} := \widetilde{\boldsymbol{X}}_{t,i}$, $\boldsymbol{Y}_{t,i} := \widetilde{\boldsymbol{Y}}_{t,i}$ and $\boldsymbol{z}_{t,i} := \widetilde{\boldsymbol{z}}_{t,i}$ for $i \in [t-1]$. Similar to Equation (2), for the memory data, we have

$$\boldsymbol{Y}_{t,i} = \boldsymbol{X}_{t,i}^\top \boldsymbol{w}_i^* + \boldsymbol{z}_{t,i}. \tag{27}$$

where $\boldsymbol{z}_{t,i} \sim \mathcal{N}(0, \sigma_i^2 \boldsymbol{I}_p)$ is i.i.d. noise. Since there is no memory data involved for task 1, by combining Lemma B.1 and the fact that $\boldsymbol{w}_0 = \boldsymbol{0}$, we can easily derive the first parameter as

$$\boldsymbol{w}_1 = P_{\boldsymbol{X}_1}\boldsymbol{w}_1^* + \boldsymbol{X}_1^\dagger \boldsymbol{z}_1,$$

Then, the expected value of the model error $\mathbb{E}\mathcal{L}_i(\boldsymbol{w}_1)$ can be derived as follows.

$$\mathbb{E}\|\boldsymbol{w}_1 - \boldsymbol{w}_i^*\|^2 \overset{(i)}{=} \mathbb{E}\|P_{\boldsymbol{X}_1}(\boldsymbol{w}_1^* - \boldsymbol{w}_i^*)\|^2 + \mathbb{E}\|(\boldsymbol{I} - P_{\boldsymbol{X}_1})\boldsymbol{w}_i^*\|^2 + \mathbb{E}\left\|\boldsymbol{X}_1^\dagger \boldsymbol{z}_1\right\|^2$$

$$\overset{(ii)}{=} \frac{n}{p}\|\boldsymbol{w}_1^* - \boldsymbol{w}_i^*\|^2 + \left(1 - \frac{n}{p}\right)\|\boldsymbol{w}_i^*\|^2 + \frac{n\sigma^2}{p-n-1}, \tag{28}$$

where $(i)$ follows from Lemma B.4 and the fact that $\boldsymbol{z}_1$ are independent Gaussian with zero mean and $(ii)$ follows from Lemma B.2 and Lemma B.3. For $t \geq 2$, the two training methods use memory in different ways. We present them in the following two subsections.

## C.1. Proof of Concurrent Rehearsal in Lemma C.1

In this subsection, we prove Lemma C.1 for concurrent rehearsal method. To simplify, we apply the following notations to denote the current data in this subsection: $\boldsymbol{X}_t := \boldsymbol{X}_{t,t}$, $\boldsymbol{Y}_t := \boldsymbol{Y}_{t,t}$ and $\boldsymbol{z}_t := \boldsymbol{z}_{t,t}$. Then, for each task $t$, the SGD convergent point $\boldsymbol{w}_t$ of training loss $\mathcal{L}_t^{tr}(\boldsymbol{w}, \mathcal{D}_t \bigcup \mathcal{M}_t)$ is equivalent to the optimization problem:

$$\boldsymbol{w}_t = \min_{\boldsymbol{w}} \|\boldsymbol{w} - \boldsymbol{w}_{t-1}\|^2 \quad s.t. \ \boldsymbol{X}_{t,j}^\top \boldsymbol{w} = \boldsymbol{Y}_{t,j}, \ j \in [t].$$

Define $V_t = [X_{t,1}, X_{t,2}, ..., X_{t,t}]$ and $\vec{z}_t = [z_{t,1}, z_{t,2}, ..., z_{t,t}]^\top$. According to Lemma B.1, we have

$$
w_t = w_{t-1} + V_t^\dagger \left( \begin{bmatrix} Y_{t,1} \\ Y_{t,2} \\ ... \\ Y_{t,t} \end{bmatrix} - V_t^\top w_{t-1} \right)
$$

$$
= (I - P_{V_t}) w_{t-1} + V_t^\dagger \begin{bmatrix} X_{t,1}^\top w_1^* \\ X_{t,2}^\top w_2^* \\ ... \\ X_{t,t}^\top w_t^* \end{bmatrix} + V_t^\dagger \vec{z}_t.
$$

Consider an arbitrary $i$ s.t. $i \leq T$ and fix it. The expected value of model error $\mathbb{E}\mathcal{L}_i(w_t)$ can be split into the following three parts.

$$
\mathbb{E} \|w_t - w_i^*\|^2 = \mathbb{E} \left\| (I - P_{V_t})(w_{t-1} - w_i^*) + V_t^\dagger \begin{bmatrix} X_{t,1}^\top(w_1^* - w_i^*) \\ X_{t,2}^\top(w_2^* - w_i^*) \\ ... \\ X_{t,t}^\top(w_t^* - w_i^*) \end{bmatrix} + V_t^\dagger \vec{z}_t \right\|^2
$$

$$
\stackrel{(i)}{=} \mathbb{E} \|(I - P_{V_t})(w_{t-1} - w_i^*)\|^2 + \mathbb{E} \left\| V_t^\dagger \begin{bmatrix} X_{t,1}^\top(w_1^* - w_i^*) \\ X_{t,2}^\top(w_2^* - w_i^*) \\ ... \\ X_{t,t}^\top(w_t^* - w_i^*) \end{bmatrix} \right\|^2 + \mathbb{E} \left\| V_t^\dagger \vec{z}_t \right\|^2
$$

$$
\stackrel{(ii)}{=} \left(1 - \frac{n_t + M_t}{p}\right) \mathbb{E} \|w_{t-1} - w_i^*\|^2 + \underbrace{\mathbb{E} \left\| V_t^\dagger \begin{bmatrix} X_{t,1}^\top(w_1^* - w_i^*) \\ X_{t,2}^\top(w_2^* - w_i^*) \\ ... \\ X_{t,t}^\top(w_t^* - w_i^*) \end{bmatrix} \right\|^2}_{\text{term}_2^{\text{(concurrent)}}} + \frac{(n + M)\sigma^2}{p - n - M - 1}, \tag{29}
$$

where $(i)$ follows from Lemma B.4 and the fact that $\vec{z}_t$ are independent Gaussian with zero mean and $(ii)$ follows from Lemma B.2 and Lemma B.3. Denote $V_{t,j}$ as $V_t$ with all zero elements except $X_{t,j}$, i.e., $V_{t,j} = [0, ..., X_{t,j}, ..., 0]$. To further calculate $\text{term}_2^{\text{(concurrent)}}$ in Equation (29), we have:

$$
\mathbb{E} \left\| V_t^\dagger \begin{bmatrix} X_{t,1}^\top(w_1^* - w_i^*) \\ X_{t,2}^\top(w_2^* - w_i^*) \\ ... \\ X_{t,t}^\top(w_t^* - w_i^*) \end{bmatrix} \right\|^2
$$

$$
= \mathbb{E} \left\| \sum_{j=1}^t V_t^\dagger V_{t,j}^\top(w_j^* - w_i^*) \right\|^2
$$

$$
= \sum_{j=1}^{t-1} \mathbb{E} \left\| V_t^\dagger V_{t,j}^\top(w_j^* - w_i^*) \right\|^2 + \sum_{j=1}^t \sum_{k=1, k\neq j}^t (w_j^* - w_i^*)^\top V_{t,j}(V_t^\top V_t)^{-1} V_{t,k}^\top(w_k^* - w_i^*)
$$

$$
\stackrel{(i)}{=} \sum_{j=1}^{t-1} \frac{M_{t,j}}{p}\left(1 + \frac{n_t + M_t - M_{t,j}}{p - n_t - M_t - 1}\right) \|w_j^* - w_i^*\|^2 + \frac{n_t}{p}\left(1 + \frac{M_t}{p - n_t - M_t - 1}\right) \|w_t^* - w_i^*\|^2
$$

$$
+ \sum_{j=1}^{t-2} \sum_{k=j+1}^{t-1} \frac{M_{t,j} M_{t,k}}{p(p - n_t - M_t - 1)} \left( \|w_j^* - w_k^*\|^2 - \|w_j^* - w_i^*\|^2 - \|w_k^* - w_i^*\|^2 \right)
$$

$$
+ \sum_{j=1}^{t-1} \frac{n_t M_{t,j}}{p(p - n_t - M_t - 1)} \left( \|w_j^* - w_t^*\|^2 - \|w_j^* - w_i^*\|^2 - \|w_t^* - w_i^*\|^2 \right) \tag{30}
$$

where $(i)$ follows from Lemma B.8 and Corollary B.10. Recall that $n_t = n$, $M_{t,j} = \frac{M}{t-1}$ and the fact that $M_t = M$. By combining Equations (29) and (30), we have:

$$
\mathbb{E}\left\|\boldsymbol{w}_t - \boldsymbol{w}_i^*\right\|^2 = \left(1 - \frac{n+M}{p}\right)\mathbb{E}\left\|\boldsymbol{w}_{t-1} - \boldsymbol{w}_i^*\right\|^2
$$
$$
+ \sum_{j=1}^{t-1}\frac{M}{(t-1)p}\left(1 + \frac{n + M - \frac{M}{t-1}}{p - n - M - 1}\right)\left\|\boldsymbol{w}_j^* - \boldsymbol{w}_i^*\right\|^2
$$
$$
+ \frac{n}{p}\left(1 + \frac{M}{p - n - M - 1}\right)\left\|\boldsymbol{w}_t^* - \boldsymbol{w}_i^*\right\|^2
$$
$$
+ \sum_{j=1}^{t-2}\sum_{k=j+1}^{t-1}\frac{(\frac{M}{t-1})^2}{p(p - n - M - 1)}\left(\left\|\boldsymbol{w}_j^* - \boldsymbol{w}_k^*\right\|^2 - \left\|\boldsymbol{w}_j^* - \boldsymbol{w}_i^*\right\|^2 - \left\|\boldsymbol{w}_k^* - \boldsymbol{w}_i^*\right\|^2\right)
$$
$$
+ \sum_{j=1}^{t-1}\frac{\frac{nM}{t-1}}{p(p - n - M - 1)}\left(\left\|\boldsymbol{w}_j^* - \boldsymbol{w}_t^*\right\|^2 - \left\|\boldsymbol{w}_j^* - \boldsymbol{w}_i^*\right\|^2 - \left\|\boldsymbol{w}_t^* - \boldsymbol{w}_i^*\right\|^2\right)
$$
$$
+ \frac{(n+M)\sigma^2}{p - n - M - 1}.
$$

By iterating the above equation and combining it with Equation (28), we have:

$$
\mathbb{E}\left\|\boldsymbol{w}_t - \boldsymbol{w}_i^*\right\|^2
$$
$$
= \left(1 - \frac{n+M}{p}\right)^{t-1}\mathbb{E}\left\|\boldsymbol{w}_1 - \boldsymbol{w}_i^*\right\|^2
$$
$$
+ \sum_{l=0}^{t-2}\left(1 - \frac{n+M}{p}\right)^l \sum_{j=1}^{t-l-1}\frac{M}{(t-l-1)p}\left(1 + \frac{n + M - \frac{M}{t-l-1}}{p - n - M - 1}\right)\left\|\boldsymbol{w}_j^* - \boldsymbol{w}_i^*\right\|^2
$$
$$
+ \sum_{l=0}^{t-2}\left(1 - \frac{n+M}{p}\right)^l \frac{n}{p}\left(1 + \frac{M}{p - n - M - 1}\right)\left\|\boldsymbol{w}_{t-l}^* - \boldsymbol{w}_i^*\right\|^2
$$
$$
+ \sum_{l=0}^{t-2}\left(1 - \frac{n+M}{p}\right)^l \sum_{j=1}^{t-l-2}\sum_{k=j+1}^{t-l-1}\frac{(\frac{M}{t-l-1})^2}{p(p - n - M - 1)}\left(\left\|\boldsymbol{w}_j^* - \boldsymbol{w}_k^*\right\|^2 - \left\|\boldsymbol{w}_j^* - \boldsymbol{w}_i^*\right\|^2 - \left\|\boldsymbol{w}_k^* - \boldsymbol{w}_i^*\right\|^2\right)
$$
$$
+ \sum_{l=0}^{t-2}\left(1 - \frac{n+M}{p}\right)^l \sum_{j=1}^{t-l-1}\frac{\frac{nM}{t-l-1}}{p(p - n - M - 1)}\left(\left\|\boldsymbol{w}_j^* - \boldsymbol{w}_{t-l}^*\right\|^2 - \left\|\boldsymbol{w}_j^* - \boldsymbol{w}_i^*\right\|^2 - \left\|\boldsymbol{w}_{t-l}^* - \boldsymbol{w}_i^*\right\|^2\right)
$$
$$
+ \sum_{l=0}^{t-2}\left(1 - \frac{n+M}{p}\right)^l \frac{(n+M)\sigma^2}{p - n - M - 1}
$$
$$
= \left(1 - \frac{n}{p}\right)\left(1 - \frac{n+M}{p}\right)^{t-1}\left\|\boldsymbol{w}_i^*\right\|^2 + \left(1 - \frac{n+M}{p}\right)^{t-1}\frac{n}{p}\mathbb{E}\left\|\boldsymbol{w}_1^* - \boldsymbol{w}_i^*\right\|^2
$$
$$
+ \sum_{l=0}^{t-2}\left(1 - \frac{n+M}{p}\right)^l \sum_{j=1}^{t-l-1}\frac{M}{(t-l-1)p}\left(1 + \frac{n + M - \frac{M}{t-l-1}}{p - n - M - 1}\right)\left\|\boldsymbol{w}_j^* - \boldsymbol{w}_i^*\right\|^2
$$
$$
+ \sum_{l=0}^{t-2}\left(1 - \frac{n+M}{p}\right)^l \frac{n}{p}\left(1 + \frac{M}{p - n - M - 1}\right)\left\|\boldsymbol{w}_{t-l}^* - \boldsymbol{w}_i^*\right\|^2
$$
$$
+ \sum_{l=0}^{t-2}\left(1 - \frac{n+M}{p}\right)^l \sum_{j=1}^{t-l-2}\sum_{k=j+1}^{t-l-1}\frac{(\frac{M}{t-l-1})^2}{p(p - n - M - 1)}\left(\left\|\boldsymbol{w}_j^* - \boldsymbol{w}_k^*\right\|^2 - \left\|\boldsymbol{w}_j^* - \boldsymbol{w}_i^*\right\|^2 - \left\|\boldsymbol{w}_k^* - \boldsymbol{w}_i^*\right\|^2\right)
$$
$$
+ \sum_{l=0}^{t-2}\left(1 - \frac{n+M}{p}\right)^l \sum_{j=1}^{t-l-1}\frac{\frac{nM}{t-l-1}}{p(p - n - M - 1)}\left(\left\|\boldsymbol{w}_j^* - \boldsymbol{w}_{t-l}^*\right\|^2 - \left\|\boldsymbol{w}_j^* - \boldsymbol{w}_i^*\right\|^2 - \left\|\boldsymbol{w}_{t-l}^* - \boldsymbol{w}_i^*\right\|^2\right)
$$

$$+ \left(1 - \frac{n}{p}\right)\left(1 - \frac{n+M}{p}\right)^{t-1} \frac{n\sigma^2}{p-n-1} + \sum_{l=0}^{t-2}\left(1 - \frac{n+M}{p}\right)^l \frac{(n+M)\sigma^2}{p-n-M-1}$$

$$= \left(1 - \frac{n}{p}\right)\left(1 - \frac{n+M}{p}\right)^{t-1}\|\boldsymbol{w}_i^*\|^2$$

$$+ \left\{ \left(1 - \frac{n+M}{p}\right)^{t-1}\frac{n}{p} + \sum_{l=0}^{t-2}\left(1 - \frac{n+M}{p}\right)^l \left[ \frac{M}{(t-l-1)p}\left(1 + \frac{n+M-\frac{M}{t-l-1}}{p-n-M-1}\right)\right.\right.$$
$$\left.\left.- \frac{\frac{nM}{(t-l-1)} + (t-l-2)(\frac{M}{t-l-1})^2}{p(p-n-M-1)} \right]\right\}\|\boldsymbol{w}_1^* - \boldsymbol{w}_i^*\|^2$$

$$+ \sum_{l=0}^{t-2}\left(1 - \frac{n+M}{p}\right)^l \sum_{j=2}^{t-l-1}\left[ \frac{M}{(t-l-1)p}\left(1 + \frac{n+M-\frac{M}{t-l-1}}{p-n-M-1}\right)\right.$$
$$\left.- \frac{\frac{nM}{t-l-1} + (t-l-2)(\frac{M}{t-l-1})^2}{p(p-n-M-1)}\right]\|\boldsymbol{w}_j^* - \boldsymbol{w}_i^*\|^2$$

$$+ \sum_{l=0}^{t-2}\left(1 - \frac{n+M}{p}\right)^l \left[ \frac{n}{p}\left(1 + \frac{M}{p-n-M-1}\right) - \frac{(t-l-1)\frac{nM}{t-l-1}}{p(p-n-M-1)}\right]\|\boldsymbol{w}_{t-l}^* - \boldsymbol{w}_i^*\|^2$$

$$+ \sum_{l=0}^{t-2}\left(1 - \frac{n+M}{p}\right)^l \sum_{j=1}^{t-l-2}\sum_{k=j+1}^{t-l-1} \frac{(\frac{M}{t-l-1})^2}{p(p-n-M-1)}\|\boldsymbol{w}_j^* - \boldsymbol{w}_k^*\|^2$$

$$+ \sum_{l=0}^{t-2}\left(1 - \frac{n+M}{p}\right)^l \sum_{j=1}^{t-l-1} \frac{\frac{nM}{t-l-1}}{p(p-n-M-1)}\|\boldsymbol{w}_j^* - \boldsymbol{w}_{t-l}^*\|^2 + \text{noise}_t^{(\text{concurrent})}(\sigma)$$

$$= \left(1 - \frac{n}{p}\right)\left(1 - \frac{n+M}{p}\right)^{t-1}\|\boldsymbol{w}_i^*\|^2$$

$$+ \sum_{j=1}^{t-1}\left\{ \sum_{l=0}^{t-j-1}\left(1 - \frac{n+M}{p}\right)^l \frac{M}{(t-l-1)p} + \left(1 - \frac{n+M}{p}\right)^{t-j}\frac{n}{p}\right\}\|\boldsymbol{w}_j^* - \boldsymbol{w}_i^*\|^2$$

$$+ \frac{n}{p}\|\boldsymbol{w}_t^* - \boldsymbol{w}_i^*\|^2$$

$$+ \sum_{l=0}^{t-2}\left(1 - \frac{n+M}{p}\right)^l \sum_{j=1}^{t-l-2}\sum_{k=j+1}^{t-l-1} \frac{(\frac{M}{t-l-1})^2}{p(p-n-M-1)}\|\boldsymbol{w}_j^* - \boldsymbol{w}_k^*\|^2$$

$$+ \sum_{l=0}^{t-2}\left(1 - \frac{n+M}{p}\right)^l \sum_{j=1}^{t-l-1} \frac{\frac{nM}{t-l-1}}{p(p-n-M-1)}\|\boldsymbol{w}_j^* - \boldsymbol{w}_{t-l}^*\|^2 + \text{noise}_t^{(\text{concurrent})}(\sigma), \tag{31}$$

where

$$\text{noise}_t^{(\text{concurrent})}(\sigma) = \left(1 - \frac{n}{p}\right)\left(1 - \frac{n+M}{p}\right)^{t-1}\frac{n\sigma^2}{p-n-1} + \sum_{l=0}^{t-2}\left(1 - \frac{n+M}{p}\right)^l \frac{(n+M)\sigma^2}{p-n-M-1}.$$

By rearranging the terms in the above equation, we complete the poof for $d_{0t}^{(\text{concurrent})}$ and $d_{ijkt}^{(\text{concurrent})}$ in Lemma C.1. Furthermore, the expressions of $c_i^{(\text{concurrent})}$ and $c_{ijk}^{(\text{concurrent})}$ can be derived directly based on $d_{0t}^{(\text{concurrent})}$, $d_{ijkt}^{(\text{concurrent})}$ and the definitions of forgetting. The explicit expressions of $c_i^{(\text{concurrent})}$ and $c_{ijk}^{(\text{concurrent})}$ are derived as follows.

$$\left[\mathbb{E}\|\boldsymbol{w}_t - \boldsymbol{w}_i^*\|^2 - \mathbb{E}\|\boldsymbol{w}_i - \boldsymbol{w}_i^*\|^2\right]^{(\text{concurrent})}$$

$$= \left(1 - \frac{n}{p}\right)\left[\left(1 - \frac{n+M}{p}\right)^{t-1} - \left(1 - \frac{n+M}{p}\right)^{i-1}\right]\|\boldsymbol{w}_i^*\|^2$$

$$+ \left\{ \left[ \left(1 - \frac{n+M}{p}\right)^{t-1} - \left(1 - \frac{n+M}{p}\right)^{i-1} \right] \frac{n}{p} + \sum_{l=0}^{t-2} \left(1 - \frac{n+M}{p}\right)^{l} \frac{M}{(t-l-1)p} \right.$$

$$\left. - \sum_{l=0}^{i-2} \left(1 - \frac{n+M}{p}\right)^{l} \frac{M}{(i-l-1)p} \right\} \|\boldsymbol{w}_1^* - \boldsymbol{w}_i^*\|^2$$

$$+ \sum_{j=i}^{t-1} \left\{ \sum_{l=0}^{t-j-1} \left(1 - \frac{n+M}{p}\right)^{l} \frac{M}{(t-l-1)p} + \left(1 - \frac{n+M}{p}\right)^{t-j} \frac{n}{p} \right\} \|\boldsymbol{w}_j^* - \boldsymbol{w}_i^*\|^2$$

$$+ \sum_{j=2}^{i-1} \left\{ \sum_{l=0}^{t-j-1} \left(1 - \frac{n+M}{p}\right)^{l} \frac{M}{(t-l-1)p} + \left(1 - \frac{n+M}{p}\right)^{t-j} \frac{n}{p} \right.$$

$$\left. - \sum_{l=0}^{i-j-1} \left(1 - \frac{n+M}{p}\right)^{l} \frac{M}{(i-l-1)p} - \left(1 - \frac{n+M}{p}\right)^{i-j} \frac{n}{p} \right\} \|\boldsymbol{w}_j^* - \boldsymbol{w}_i^*\|^2$$

$$+ \frac{n}{p} \|\boldsymbol{w}_t^* - \boldsymbol{w}_i^*\|^2$$

$$\left. \begin{array}{l} + \sum_{l=0}^{t-2} \left(1 - \frac{n+M}{p}\right)^{l} \sum_{j=1}^{t-l-2} \sum_{k=j+1}^{t-l-1} \frac{(\frac{M}{t-l-1})^2}{p(p-n-M-1)} \|\boldsymbol{w}_j^* - \boldsymbol{w}_k^*\|^2 \\ - \sum_{l=0}^{i-2} \left(1 - \frac{n+M}{p}\right)^{l} \sum_{j=1}^{i-l-2} \sum_{k=j+1}^{i-l-1} \frac{(\frac{M}{i-l-1})^2}{p(p-n-M-1)} \|\boldsymbol{w}_j^* - \boldsymbol{w}_k^*\|^2 \end{array} \right\} \beta_1$$

$$\left. \begin{array}{l} + \sum_{l=0}^{t-2} \left(1 - \frac{n+M}{p}\right)^{l} \sum_{j=1}^{t-l-1} \frac{\frac{nM}{t-l-1}}{p(p-n-M-1)} \|\boldsymbol{w}_j^* - \boldsymbol{w}_{t-l}^*\|^2 \\ - \sum_{l=0}^{i-2} \left(1 - \frac{n+M}{p}\right)^{l} \sum_{j=1}^{i-l-1} \frac{\frac{nM}{i-l-1}}{p(p-n-M-1)} \|\boldsymbol{w}_j^* - \boldsymbol{w}_{i-l}^*\|^2 . \end{array} \right\} \beta_2$$

$$+ \text{noise}_t^{(\text{concurrent})}(\sigma) - \text{noise}_i^{(\text{concurrent})}(\sigma) \tag{32}$$

We will show that $\beta_1$ consists of terms $\delta_{j,k} \|\boldsymbol{w}_j^* - \boldsymbol{w}_k^*\|^2$ with $\delta_{j,k} > 0$ and $j, k \neq t$ and $\beta_2$ consists of terms $\eta_{j,k} \|\boldsymbol{w}_j^* - \boldsymbol{w}_k^*\|^2$ with $\eta_{j,k} > 0$ for a sufficient large $p$ in Appendix F.2.

## C.2. Proof of Sequential Rehearsal in Lemma C.1

In this subsection, we prove Lemma C.1 for sequential rehearsal method. To simplify, we apply the following notations to denote the current data in this subsection: $\boldsymbol{X}_t := \boldsymbol{X}_{t,0}$, $\boldsymbol{Y}_t := \boldsymbol{Y}_{t,0}$ and $\boldsymbol{z}_t := \boldsymbol{z}_{t,0}$. When $t \geq 2$, the sequence of SGD convergent points $\boldsymbol{w}_t^{(j)}$ is equivalent the sequential optimization problems:

$$\hat{\boldsymbol{w}}_t^{(j)} = \min_{\boldsymbol{w}} \left\| \boldsymbol{w} - \hat{\boldsymbol{w}}_t^{(j-1)} \right\|_2^2 \quad s.t. \ \boldsymbol{X}_{t,j}^\top \boldsymbol{w} = \boldsymbol{Y}_{t,j}, \ j = 0, 1, ..., t-1,$$

where $\hat{\boldsymbol{w}}_t^{(-1)} = \boldsymbol{w}_{t-1}$ and $\boldsymbol{w}_t = \hat{\boldsymbol{w}}_t^{(t-1)}$. Consider an arbitrary $i$ s.t. $i \leq T$ and fix it. During the training process over each task in the memory dataset, the expected of model error can be split into three parts as follows.

$$\mathbb{E} \left\| \hat{\boldsymbol{w}}_t^{(j)} - \boldsymbol{w}_i^* \right\|^2 = \mathbb{E} \left\| (\boldsymbol{I} - P_{\boldsymbol{X}_{t,j}})(\hat{\boldsymbol{w}}_t^{(j-1)} - \boldsymbol{w}_i^*) + P_{\boldsymbol{X}_{t,j}}(\boldsymbol{w}_j^* - \boldsymbol{w}_i^*) + \boldsymbol{X}_{t,j}^\dagger \boldsymbol{z}_{t,j} \right\|^2$$

$$\overset{(i)}{=} \left(1 - \frac{M}{(t-1)p}\right) \mathbb{E} \left\| \hat{\boldsymbol{w}}_t^{(j-1)} - \boldsymbol{w}_i^* \right\|^2 + \frac{M}{(t-1)p} \|\boldsymbol{w}_j^* - \boldsymbol{w}_i^*\|^2 + \frac{\frac{M}{(t-1)}\sigma^2}{p - \frac{M}{(t-1)} - 1},$$

for $j = 1, 2, ..., t-1$, where (i) follows from Lemmas B.1 to B.4. Similarly, for the current dataset, we have:

$$\mathbb{E} \left\| \hat{\boldsymbol{w}}_t^{(0)} - \boldsymbol{w}_i^* \right\|^2 = \mathbb{E} \left\| (\boldsymbol{I} - P_{\boldsymbol{X}_t})(\boldsymbol{w}_{t-1} - \boldsymbol{w}_i^*) + P_{\boldsymbol{X}_t}(\boldsymbol{w}_t^* - \boldsymbol{w}_i^*) \right\|^2$$

$$= \left(1 - \frac{n}{p}\right) \mathbb{E} \left\| \boldsymbol{w}_{t-1} - \boldsymbol{w}_i^* \right\|^2 + \frac{n}{p} \left\| \boldsymbol{w}_t^* - \boldsymbol{w}_i^* \right\|^2 + \frac{n\sigma^2}{p - n - 1}.$$

By combining the above two equations, we have:

$$\mathbb{E} \left\| \boldsymbol{w}_t - \boldsymbol{w}_i^* \right\|^2 = \left(1 - \frac{M}{(t-1)p}\right)^{t-1} \left(1 - \frac{n}{p}\right) \mathbb{E} \left\| \boldsymbol{w}_{t-1} - \boldsymbol{w}_i^* \right\|^2$$

$$+ \underbrace{\sum_{j=1}^{t-1} \left(1 - \frac{M}{(t-1)p}\right)^{t-j-1} \frac{M}{(t-1)p} \mathbb{E} \left\| \boldsymbol{w}_j^* - \boldsymbol{w}_i^* \right\|^2 + \left(1 - \frac{M}{(t-1)p}\right)^{t-1} \frac{n}{p} \left\| \boldsymbol{w}_t^* - \boldsymbol{w}_i^* \right\|^2}_{\text{term}_2^{\text{(sequential)}}}$$

$$+ \sum_{j=1}^{t-1} \left(1 - \frac{M}{(t-1)p}\right)^{t-j-1} \frac{\frac{M}{t-1}\sigma^2}{p - \frac{M}{t-1} - 1} + \left(1 - \frac{M}{(t-1)p}\right)^{t-1} \frac{n\sigma^2}{p - n - 1}. \tag{33}$$

By applying this process recursively, we obtain the expression of the expected value of the model error $\mathbb{E}\mathcal{L}_i(\boldsymbol{w}_t)$ as follows.

$$\mathbb{E} \left\| \boldsymbol{w}_t - \boldsymbol{w}_i^* \right\|^2$$

$$= \prod_{l=0}^{t-2} \left[ \left(1 - \frac{M}{(t-l-1)p}\right)^{t-l-1} \left(1 - \frac{n}{p}\right) \right] \mathbb{E} \left\| \boldsymbol{w}_1 - \boldsymbol{w}_i^* \right\|^2$$

$$+ \sum_{j=1}^{t-1} \left\{ \sum_{l=0}^{t-j-1} \prod_{k=0}^{l-1} \left[ \left(1 - \frac{M}{(t-k-1)p}\right)^{t-k-1} \left(1 - \frac{n}{p}\right) \right] \left(1 - \frac{M}{(t-l-1)p}\right)^{t-j-l-1} \frac{M}{(t-l-1)p} \right\} \left\| \boldsymbol{w}_j^* - \boldsymbol{w}_i^* \right\|^2$$

$$+ \sum_{l=0}^{t-2} \prod_{k=0}^{l-1} \left[ \left(1 - \frac{M}{(t-k-1)p}\right)^{t-k-1} \left(1 - \frac{n}{p}\right) \right] \left(1 - \frac{M}{(t-l-1)p}\right)^{t-l-1} \frac{n}{p} \left\| \boldsymbol{w}_{t-l}^* - \boldsymbol{w}_i^* \right\|^2$$

$$+ \left(1 - \frac{M}{(t-1)p}\right)^{t-1} \frac{n}{p} \left\| \boldsymbol{w}_t^* - \boldsymbol{w}_i^* \right\|^2 + \text{noise}_t^{\text{(sequential)}}(\sigma)$$

$$= \left(1 - \frac{n}{p}\right) \prod_{l=0}^{t-2} \left[ \left(1 - \frac{M}{(t-l-1)p}\right)^{t-l-1} \left(1 - \frac{n}{p}\right) \right] \left\| \boldsymbol{w}_i^* \right\|^2$$

$$+ \left\{ \sum_{l=0}^{t-2} \prod_{k=0}^{l-1} \left[ \left(1 - \frac{M}{(t-k-1)p}\right)^{t-k-1} \left(1 - \frac{n}{p}\right) \right] \left(1 - \frac{M}{(t-l-1)p}\right)^{t-l-2} \frac{M}{(t-l-1)p} \right.$$

$$\left. + \prod_{l=0}^{t-2} \left[ \left(1 - \frac{M}{(t-l-1)p}\right)^{t-l-1} \left(1 - \frac{n}{p}\right) \right] \frac{n}{p} \right\} \left\| \boldsymbol{w}_1^* - \boldsymbol{w}_i^* \right\|^2$$

$$+ \sum_{j=2}^{t-1} \left\{ \sum_{l=0}^{t-j-1} \prod_{k=0}^{l-1} \left[ \left(1 - \frac{M}{(t-k-1)p}\right)^{t-k-1} \left(1 - \frac{n}{p}\right) \right] \left(1 - \frac{M}{(t-l-1)p}\right)^{t-j-l-1} \frac{M}{(t-l-1)p} \right.$$

$$\left. + \prod_{k=0}^{t-j-1} \left[ \left(1 - \frac{M}{(t-l-1)p}\right)^{t-k-1} \left(1 - \frac{n}{p}\right) \right] \left(1 - \frac{M}{(j-1)p}\right)^{j-1} \frac{n}{p} \right\} \left\| \boldsymbol{w}_j^* - \boldsymbol{w}_i^* \right\|^2$$

$$+ \left(1 - \frac{M}{(t-1)p}\right)^{t-1} \frac{n}{p} \left\| \boldsymbol{w}_t^* - \boldsymbol{w}_i^* \right\|^2 + \text{noise}_t^{\text{(sequential)}}(\sigma), \tag{34}$$

where

$$\text{noise}_t^{\text{(sequential)}}(\sigma) = \sum_{l=0}^{t-2} \prod_{k=0}^{l-1} \left[ \left(1 - \frac{M}{(t-k-1)p}\right)^{t-k-1} \left(1 - \frac{n}{p}\right) \right]$$

$$\cdot \left[ \sum_{j=1}^{t-1} \left(1 - \frac{M}{(t-1)p}\right)^{t-j-1} \frac{\frac{M}{t-1}\sigma^2}{p - \frac{M}{t-1} - 1} + \left(1 - \frac{M}{(t-1)p}\right)^{t-1} \frac{n\sigma^2}{p - n - 1} \right].$$

By rearranging the terms, we complete the poof for $d_{0t}^{(\text{sequential})}$ and $d_{ijkt}^{(\text{sequential})}$ in Lemma C.1. Furthermore, the expressions of $c_i^{(\text{sequential})}$ and $c_{ijk}^{(\text{sequential})}$ can be derived directly based on $d_{0t}^{(\text{sequential})}$, $d_{ijkt}^{(\text{sequential})}$ and the definition of forgetting. The explicit expressions of $c_i^{(\text{concurrent})}$ and $c_{ijk}^{(\text{sequential})}$ are derived as follows.

$$
\left[\mathbb{E}\left\|\boldsymbol{w}_t - \boldsymbol{w}_i^*\right\|_2^2 - \mathbb{E}\left\|\boldsymbol{w}_i - \boldsymbol{w}_i^*\right\|_2^2\right]^{(\text{sequential})}
$$

$$
= \left(1 - \frac{n}{p}\right)\left\{\prod_{l=0}^{t-2}\left[\left(1 - \frac{M}{(t-l-1)p}\right)^{t-l-1}\left(1 - \frac{n}{p}\right)\right] - \prod_{l=0}^{i-2}\left[\left(1 - \frac{M}{(i-l-1)p}\right)^{i-l-1}\left(1 - \frac{n}{p}\right)\right]\right\}\left\|\boldsymbol{w}_i^*\right\|^2
$$

$$
+ \left\{\frac{n}{p}\left\{\prod_{l=0}^{t-2}\left[\left(1 - \frac{M}{(t-l-1)p}\right)^{t-l-1}\left(1 - \frac{n}{p}\right)\right] - \prod_{l=0}^{i-2}\left[\left(1 - \frac{M}{(i-l-1)p}\right)^{i-l-1}\left(1 - \frac{n}{p}\right)\right]\right\}\right.
$$

$$
+ \sum_{l=0}^{t-2}\prod_{k=0}^{l-1}\left[\left(1 - \frac{M}{(t-k-1)p}\right)^{t-k-1}\left(1 - \frac{n}{p}\right)\right]\left(1 - \frac{M}{(t-l-1)p}\right)^{t-l-2}\frac{M}{(t-l-1)p}
$$

$$
\left. - \sum_{l=0}^{i-2}\prod_{k=0}^{l-1}\left[\left(1 - \frac{M}{(i-k-1)p}\right)^{i-k-1}\left(1 - \frac{n}{p}\right)\right]\left(1 - \frac{M}{(i-l-1)p}\right)^{i-l-2}\frac{M}{(i-l-1)p}\right\}\left\|\boldsymbol{w}_1^* - \boldsymbol{w}_i^*\right\|^2
$$

$$
+ \sum_{j=i}^{t-1}\left\{\sum_{l=0}^{t-j-1}\prod_{k=0}^{l-1}\left[\left(1 - \frac{M}{(t-k-1)p}\right)^{t-k-1}\left(1 - \frac{n}{p}\right)\right]\left(1 - \frac{M}{(t-l-1)p}\right)^{t-j-l-1}\frac{M}{(t-l-1)p}\right.
$$

$$
\left. + \prod_{k=0}^{t-j-1}\left[\left(1 - \frac{M}{(t-k-1)p}\right)^{t-k-1}\left(1 - \frac{n}{p}\right)\right]\left(1 - \frac{M}{(j-1)p}\right)^{j-1}\frac{n}{p}\right\}\left\|\boldsymbol{w}_j^* - \boldsymbol{w}_i^*\right\|^2
$$

$$
+ \sum_{j=2}^{i-1}\left\{\sum_{l=0}^{t-j-1}\prod_{k=0}^{l-1}\left[\left(1 - \frac{M}{(t-k-1)p}\right)^{t-k-1}\left(1 - \frac{n}{p}\right)\right]\left(1 - \frac{M}{(t-l-1)p}\right)^{t-j-l-1}\frac{M}{(t-l-1)p}\right.
$$

$$
- \sum_{l=0}^{i-j-1}\prod_{k=0}^{l-1}\left[\left(1 - \frac{M}{(i-k-1)p}\right)^{i-k-1}\left(1 - \frac{n}{p}\right)\right]\left(1 - \frac{M}{(i-l-1)p}\right)^{i-j-l-1}\frac{M}{(i-l-1)p}
$$

$$
+ \prod_{k=0}^{t-j-1}\left[\left(1 - \frac{M}{(t-k-1)p}\right)^{t-k-1}\left(1 - \frac{n}{p}\right)\right]\left(1 - \frac{M}{(j-1)p}\right)^{j-1}\frac{n}{p}
$$

$$
\left. - \prod_{k=0}^{i-j-1}\left[\left(1 - \frac{M}{(i-k-1)p}\right)^{i-k-1}\left(1 - \frac{n}{p}\right)\right]\left(1 - \frac{M}{(j-1)p}\right)^{j-1}\frac{n}{p}\right\}\left\|\boldsymbol{w}_j^* - \boldsymbol{w}_i^*\right\|^2
$$

$$
+ \left(1 - \frac{M}{(t-1)p}\right)^{t-1}\frac{n}{p}\left\|\boldsymbol{w}_t^* - \boldsymbol{w}_i^*\right\|^2 + \text{noise}_t^{(\text{sequential})}(\sigma) - \text{noise}_i^{(\text{sequential})}(\sigma) \tag{35}
$$

### C.3. Proof of Theorem 5.1

Theorem 5.1 follows directly from Lemma C.1 and the definitions of $F_T$ and $G_T$.

## D. Proof of Lemma 5.2 and Theorem 5.3

In this section, we will prove Lemma 5.2 and Theorem 5.3, and provide details about constants $\xi_1, \xi_2, \mu_1, \mu_2$. According to Equations (31), (32), (34) and (35), the forgetting and generalization error for $T = 2$ is as follows. For concurrent rehearsal method, the forgetting is provided as follows.

$$
\begin{aligned}
F_2^{(\text{concurrent})} &= \mathbb{E}\left\|\boldsymbol{w}_2 - \boldsymbol{w}_1^*\right\|^2 - \mathbb{E}\left\|\boldsymbol{w}_1 - \boldsymbol{w}_1^*\right\|^2 \\
&= \left(-\frac{n+M}{p}\right)\left(1 - \frac{n}{p}\right)\left\|\boldsymbol{w}_1^*\right\|^2 + \frac{n}{p}\left(1 + \frac{M}{p-n-M-1}\right)\left\|\boldsymbol{w}_1^* - \boldsymbol{w}_2^*\right\|^2 \\
&\qquad + \frac{(n+M)\sigma^2}{p - (n+M) - 1} - \frac{n+M}{p}\cdot\frac{n\sigma^2}{p-n-1}.
\end{aligned} \tag{36}
$$

And also, we provide the generalization error as follows.

$$
\begin{aligned}
G_2^{\text{(concurrent)}} &= \frac{1}{2}\left(\mathbb{E}\left\|\boldsymbol{w}_2 - \boldsymbol{w}_1^*\right\|^2 + \mathbb{E}\left\|\boldsymbol{w}_2 - \boldsymbol{w}_2^*\right\|^2\right)\\
&= \frac{1}{2}\left(1 - \frac{n+M}{p}\right)\left(1 - \frac{n}{p}\right)\left(\|\boldsymbol{w}_1^*\|^2 + \|\boldsymbol{w}_2^*\|^2\right)\\
&\quad + \frac{1}{2}\left(\frac{2n+M}{p} + \frac{2nM}{p(p-n-M-1)} - \frac{n(n+M)}{p^2}\right)\|\boldsymbol{w}_1^* - \boldsymbol{w}_2^*\|^2\\
&\quad + \frac{(n+M)\sigma^2}{p-(n+M)-1} + \left(1 - \frac{n+M}{p}\right)\frac{n\sigma^2}{p-n-1}.
\end{aligned}
\tag{37}
$$

For sequential rehearsal method, the forgetting is provided as follows.

$$
\begin{aligned}
F_2^{\text{sequential}} &= \mathbb{E}\left\|\boldsymbol{w}_2 - \boldsymbol{w}_1^*\right\|^2 - \mathbb{E}\left\|\boldsymbol{w}_1 - \boldsymbol{w}_1^*\right\|^2\\
&= \left(-\frac{n+M}{p} + \frac{nM}{p^2}\right)\left(1 - \frac{n}{p}\right)\|\boldsymbol{w}_1^*\|^2 + \left(1 - \frac{M}{p}\right)\frac{n}{p}\|\boldsymbol{w}_1^* - \boldsymbol{w}_2^*\|^2\\
&\quad + \left(1 - \frac{n+2M}{p} + \frac{nM}{p^2}\right)\frac{n\sigma^2}{p-n-1} + \frac{M\sigma^2}{p-M-1}.
\end{aligned}
\tag{38}
$$

And also, we provide the generalization error as follows.

$$
\begin{aligned}
G_2^{\text{sequential}} &= \frac{1}{2}\left(\mathbb{E}\left\|\boldsymbol{w}_2 - \boldsymbol{w}_1^*\right\|^2 + \mathbb{E}\left\|\boldsymbol{w}_2 - \boldsymbol{w}_2^*\right\|^2\right)\\
&= \frac{1}{2}\left(1 - \frac{M}{p}\right)\left(1 - \frac{n}{p}\right)^2\left(\|\boldsymbol{w}_1^*\|^2 + \|\boldsymbol{w}_2^*\|^2\right) + \frac{1}{2}\left(\frac{2n+M}{p} - \frac{n(n+2M)}{p^2} + \frac{n^2M}{p^3}\right)\|\boldsymbol{w}_1^* - \boldsymbol{w}_2^*\|^2\\
&\quad + \left(1 - \frac{M}{p}\right)\left(2 - \frac{n}{p}\right)\frac{n\sigma^2}{p-n-1} + \frac{M\sigma^2}{p-M-1}.
\end{aligned}
\tag{39}
$$

### D.1. Proof of Coefficients $\hat{c}_1, \hat{c}_1$ in Lemma 5.2 and Forgetting in Theorem 5.3

By observing Equation (36) and Equation (38), the expressions of forgetting for both rehearsal methods share the same structure:
$$
F_2 = \hat{c}_1\|\boldsymbol{w}_1^*\|^2 + \hat{c}_2\|\boldsymbol{w}_1^* - \boldsymbol{w}_2^*\|^2 + \hat{\text{noise}}_F(\sigma).
$$
We first compare the coefficients $\hat{c}_1, \hat{c}_2$ and the noise term $\hat{\text{noise}}_F(\sigma)$. In the below inequalities, the expressions for concurrent rehearsal are provided on the left while the expressions for sequential rehearsal are on the right.

$$
\hat{c}_1: \qquad\qquad\qquad \left(-\frac{n+M}{p}\right)\left(1 - \frac{n}{p}\right) < \left(-\frac{n+M}{p} + \frac{nM}{p^2}\right)\left(1 - \frac{n}{p}\right)
$$

$$
\hat{c}_2: \qquad\qquad\qquad \frac{n}{p}\left(1 + \frac{M}{p-n-M-1}\right) > \left(1 - \frac{M}{p}\right)\frac{n}{p},
$$

$$
\hat{\text{noise}}_F(\sigma): \qquad \frac{(n+M)\sigma^2}{p-(n+M)-1} - \frac{n+M}{p}\cdot\frac{n\sigma^2}{p-n-1} > \left(1 - \frac{n+2M}{p} + \frac{nM}{p^2}\right)\frac{n\sigma^2}{p-n-1} + \frac{M\sigma^2}{p-M-1}.
$$

The comparison implies that $\hat{c}_1^{\text{(concurrent)}} < \hat{c}_1^{\text{(sequential)}}$, $\hat{c}_2^{\text{(concurrent)}} > \hat{c}_2^{\text{(sequential)}}$ and $\hat{\text{noise}}_F^{\text{(concurrent)}}(\sigma) > \hat{\text{noise}}_F^{\text{(sequential)}}(\sigma)$. By further calculation, we obtain the following conclusion:

$$
F_2^{\text{(concurrent)}} > F_2^{\text{(sequential)}} \quad \textbf{if and only if} \quad \xi_1\|\boldsymbol{w}_1^* - \boldsymbol{w}_2^*\|^2 + \xi_2\sigma^2 > \|\boldsymbol{w}_1^*\|^2,
$$

where $\xi_1 = \frac{\frac{nM}{p}\left(\frac{1}{p-n-M-1} + \frac{1}{p}\right)}{\frac{nM}{p^2}\left(1 - \frac{n}{p}\right)}$ and $\xi_2 = \frac{\left(\frac{n+M}{p-n-M-1} - \left(1 - \frac{M}{p} + \frac{nM}{p^2}\right)\frac{n}{p-n-1} - \frac{M}{p-M-1}\right)}{\frac{nM}{p^2}\left(1 - \frac{n}{p}\right)}$. To illustrate this conclusion better, we provide the following two special cases.

- If the noise $\sigma$ is 0, and the task similarity is low enough (i.e., $\|\boldsymbol{w}_1^* - \boldsymbol{w}_2^*\|^2$ is large enough), sequential rehearsal achieves a lower forgetting. More specifically, $F_2^{\text{(concurrent)}} \geq F_2^{\text{(sequential)}}$ **if and only if** $\|\boldsymbol{w}_1^* - \boldsymbol{w}_2^*\|^2 \geq \frac{(p-n)(p-n-M-1)}{p^2+p(p-n-M-1)}\|\boldsymbol{w}_1^*\|^2$,

- If task difference $\|\boldsymbol{w}_1^* - \boldsymbol{w}_2^*\|^2 = 0$ and the noise $\sigma$ is large enough, sequential rehearsal achieves a lower forgetting. More specifically, $F_2^{\text{(concurrent)}} \geq F_2^{\text{(sequential)}}$ **if and only if**

$$\sigma \geq \frac{\frac{nM}{p^2}\left(1 - \frac{n}{p}\right)}{\frac{n+M}{p-n-M-1} - \left(1 - \frac{M}{p} + \frac{nM}{p^2}\right)\frac{n}{p-n-1} - \frac{M}{p-M-1}} \|\boldsymbol{w}_1^*\|^2 \, .$$

### D.2. Proof of Coefficients $\hat{d}_1, \hat{d}_2$ in Lemma 5.2 and Generalization error in Theorem 5.3

By observing Equation (37) and Equation (39), the expressions of generalization error for both rehearsal methods share the same structure:

$$G_2 = \hat{d}_1(\|\boldsymbol{w}_1^*\|^2 + \|\boldsymbol{w}_2^*\|^2) + \hat{d}_2 \|\boldsymbol{w}_1^* - \boldsymbol{w}_2^*\|^2 + \hat{\text{noise}}_G(\sigma).$$

We first compare the coefficients $\hat{d}_1, \hat{d}_2$ and the noise term $\hat{\text{noise}}_G(\sigma)$. In the below inequalities, the expressions for concurrent rehearsal are provided on the left while the expressions for sequential rehearsal are on the right.

$$\hat{d}_1 : \qquad\qquad \left(1 - \frac{n+M}{p}\right)\left(1 - \frac{n}{p}\right) < \left(1 - \frac{M}{p}\right)\left(1 - \frac{n}{p}\right)^2$$

$$\hat{d}_2 : \quad \frac{2n+M}{p} + \frac{2nM}{p(p-n-M-1)} - \frac{n(n+M)}{p^2} > \frac{2n+M}{p} - \frac{n(n+2M)}{p^2} + \frac{n^2 M}{p^3},$$

$$\hat{\text{noise}}_G : \quad \frac{(n+M)\sigma^2}{p-(n+M)-1} + \left(1 - \frac{n+M}{p}\right)\frac{n\sigma^2}{p-n-1} > \left(1 - \frac{M}{p}\right)\left(2 - \frac{n}{p}\right)\frac{n\sigma^2}{p-n-1} + \frac{M\sigma^2}{p-M-1},$$

which implies that $\hat{d}_1^{\text{(concurrent)}} < \hat{d}_1^{\text{(sequential)}}$ and $\hat{d}_2^{\text{(concurrent)}} > \hat{d}_2^{\text{(sequential)}}$, $\hat{\text{noise}}_G^{\text{(concurrent)}}(\sigma) > \hat{\text{noise}}_G^{\text{(sequential)}}(\sigma)$. By further calculation, we obtain the following conclusion:

$$G_2^{\text{(concurrent)}} \geq G_2^{\text{(sequential)}} \quad \text{if and only if} \quad \mu_1 \|\boldsymbol{w}_1^* - \boldsymbol{w}_2^*\|^2 + \mu_2 \sigma^2 > \|\boldsymbol{w}_1^*\|^2,$$

where $\mu_1 = \frac{\frac{nM}{p}\left(\frac{2}{p-n-M-1} + \frac{1}{p} - \frac{n}{p^2}\right)}{\frac{nM}{p^2}\left(1 - \frac{n}{p}\right)}$ and $\mu_2 = \frac{\frac{n+M}{p-n-M-1} - \left(1 - \frac{M}{p} + \frac{nM}{p^2}\right)\frac{n}{p-n-1} - \frac{M}{p-M-1}}{\frac{nM}{p^2}\left(1 - \frac{n}{p}\right)}$. To illustrate this conclusion better, we provide the following two special cases.

- If the noise $\sigma$ is 0, and the task similarity is small enough (i.e., $\|\boldsymbol{w}_1^* - \boldsymbol{w}_2^*\|^2$ is big enough), sequential rehearsal has a smaller generalization error. More specifically, $G_2^{\text{(concurrent)}} \geq G_2^{\text{(sequential)}}$ **if and only if** $\|\boldsymbol{w}_1^* - \boldsymbol{w}_2^*\|^2 \geq \frac{(p-n)(p-n-M-1)}{2p^2+(p-n)(p-n-M-1)}\left(\|\boldsymbol{w}_1^*\|^2 + \|\boldsymbol{w}_2^*\|^2\right)$.

- If the task difference $\|\boldsymbol{w}_1^* - \boldsymbol{w}_2^*\|^2 = 0$ and the noise $\sigma$ is big, sequential rehearsal has a smaller generalization error. More specifically, $G_2^{\text{(concurrent)}} \geq G_2^{\text{(sequential)}}$ **if and only if**

$$\sigma^2 \geq \frac{\frac{nM}{p^2}\left(1 - \frac{n}{p}\right)}{\frac{n+M}{p-n-M-1} - \left(1 - \frac{M}{p} + \frac{nM}{p^2}\right)\frac{n}{p-n-1} - \frac{M}{p-M-1}}\left(\|\boldsymbol{w}_1^*\|^2 + \|\boldsymbol{w}_2^*\|^2\right)$$

## E. Comparison between Concurrent and Sequential Rehearsal Methods When $T = 3$

Recall that $M_{2,1} = M$ and $M_{3,1} = M_{3,2} = \frac{M}{2}$ under our equal memory allocation assumption. In this section, we assume $\sigma = 0$. According to Equations (31) and (32), we write out the performance of the concurrent rehearsal method when $T = 3$ as follows.

$$F_3^{\text{(concurrent)}}$$
$$= \frac{1}{2}(\mathbb{E}\|\boldsymbol{w}_3 - \boldsymbol{w}_1^*\|^2 - \mathbb{E}\|\boldsymbol{w}_1 - \boldsymbol{w}_1^*\|^2 + \mathbb{E}\|\boldsymbol{w}_3 - \boldsymbol{w}_2^*\|^2 - \mathbb{E}\|\boldsymbol{w}_2 - \boldsymbol{w}_2^*\|^2)$$
$$= \frac{1}{2}\left(-\frac{2(n+M)}{p} + \frac{(n+M)^2}{p^2}\right)\left(1 - \frac{n}{p}\right)\|\boldsymbol{w}_1^*\|^2 + \frac{1}{2}\left(-\frac{n+M}{p}\right)\left(1 - \frac{n+M}{p}\right)\left(1 - \frac{n}{p}\right)\|\boldsymbol{w}_2^*\|^2$$

$$+ \frac{1}{2}\left[\left(1 - \frac{2(n+M)}{p}\right)\frac{nM}{p(p-n-M-1)} + \frac{M^2}{2p(p-n-M-1)} + \frac{n+M}{p}\left(1 - \frac{n}{p}\right)\left(1 - \frac{n+M}{p}\right)\right]\|\boldsymbol{w}_1^* - \boldsymbol{w}_2^*\|^2$$

$$+ \frac{1}{2}\left[\frac{n}{p} + \frac{nM}{p(p-n-M-1)}\right]\|\boldsymbol{w}_1^* - \boldsymbol{w}_3^*\|^2 + \frac{1}{2}\left[\frac{n}{p} + \frac{nM}{p(p-n-M-1)}\right]\|\boldsymbol{w}_2^* - \boldsymbol{w}_3^*\|^2, \tag{40}$$

and

$$G_3^{(\text{concurrent})}$$

$$= \frac{1}{3}(\mathbb{E}\|\boldsymbol{w}_3 - \boldsymbol{w}_1^*\|^2 + \mathbb{E}\|\boldsymbol{w}_3 - \boldsymbol{w}_2^*\|^2 + \mathbb{E}\|\boldsymbol{w}_3 - \boldsymbol{w}_3^*\|^2)$$

$$= \frac{1}{3}\left(1 - \frac{n+M}{p}\right)^2\left(1 - \frac{n}{p}\right)(\|\boldsymbol{w}_1^*\|^2 + \|\boldsymbol{w}_2^*\|^2 + \|\boldsymbol{w}_3^*\|^2)$$

$$+ \frac{1}{3}\left[\left(3 - \frac{3(n+M)}{p}\right)\frac{nM}{p(p-n-M-1)} + \frac{3M^2}{4p(p-n-M-1)}\right.$$

$$\left. + \frac{n+M}{p}\left(2 - \frac{3n}{p} - \frac{M}{p} + \frac{n(n+M)}{p^2}\right)\right]\|\boldsymbol{w}_1^* - \boldsymbol{w}_2^*\|^2$$

$$+ \frac{1}{3}\left[\frac{n}{p}\left(2 - \frac{2(n+M)}{p} + \frac{(n+M)^2}{p^2}\right) + \frac{M}{p}\left(1 - \frac{n+M}{p}\right) + \frac{M}{2p} + \frac{3nM}{2p(p-n-M-1)}\right]\|\boldsymbol{w}_1^* - \boldsymbol{w}_3^*\|^2$$

$$+ \frac{1}{3}\left[\frac{n}{p}\left(2 - \frac{n+M}{p}\right) + \frac{M}{2p} + \frac{3nM}{2p(p-n-M-1)}\right]\|\boldsymbol{w}_2^* - \boldsymbol{w}_3^*\|^2. \tag{41}$$

According to Equations (34) and (35), we write out the performance of sequential rehearsal when $T = 3$ as follows.

$$F_3^{(\text{sequential})} = \frac{1}{2}(\mathbb{E}\|\boldsymbol{w}_3 - \boldsymbol{w}_1^*\|^2 - \mathbb{E}\|\boldsymbol{w}_1 - \boldsymbol{w}_1^*\|^2 + \mathbb{E}\|\boldsymbol{w}_3 - \boldsymbol{w}_2^*\|^2 - \mathbb{E}\|\boldsymbol{w}_2 - \boldsymbol{w}_2^*\|^2)$$

$$= \frac{1}{2}\left[\left(1 - \frac{n}{p}\right)^3\left(1 - \frac{M}{p}\right)\left(1 - \frac{M}{2p}\right)^2 - \left(1 - \frac{n}{p}\right)\right]\|\boldsymbol{w}_1^*\|^2$$

$$+ \frac{1}{2}\left[\left(1 - \frac{n}{p}\right)^3\left(1 - \frac{M}{p}\right)\left(1 - \frac{M}{2p}\right)^2 - \left(1 - \frac{n}{p}\right)^2\left(1 - \frac{M}{p}\right)\right]\|\boldsymbol{w}_2^*\|^2$$

$$+ \frac{1}{2}\left[\left(1 - \frac{n}{p}\right)\left(1 - \frac{M}{p}\right)\frac{n}{p}\left(\left(1 - \frac{M}{2p}\right)^2\left(2 - \frac{n}{p}\right) - 1\right) + \left(1 - \frac{M}{2p}\right)^2\left(1 - \frac{n}{p}\right)\frac{M}{p} - \frac{M^2}{4p^2}\right]\|\boldsymbol{w}_1^* - \boldsymbol{w}_2^*\|^2$$

$$+ \frac{1}{2}\left(1 - \frac{M}{2p}\right)^2\frac{n}{p}\|\boldsymbol{w}_1^* - \boldsymbol{w}_3^*\|^2 + \left(1 - \frac{M}{2p}\right)^2\frac{n}{p}\|\boldsymbol{w}_2^* - \boldsymbol{w}_3^*\|^2. \tag{42}$$

And also, we have

$$G_3^{(\text{concurrent})} = \frac{1}{3}(\mathbb{E}\|\boldsymbol{w}_3 - \boldsymbol{w}_1^*\|^2 + \mathbb{E}\|\boldsymbol{w}_3 - \boldsymbol{w}_2^*\|^2 + \mathbb{E}\|\boldsymbol{w}_3 - \boldsymbol{w}_3^*\|^2)$$

$$= \frac{1}{3}\left(1 - \frac{n}{p}\right)^3\left(1 - \frac{M}{p}\right)\left(1 - \frac{M}{2p}\right)^2(\|\boldsymbol{w}_1^*\|^2 + \|\boldsymbol{w}_2^*\|^2 + \|\boldsymbol{w}_3^*\|^2)$$

$$+ \frac{1}{3}\left\{\left(1 - \frac{n}{p}\right)\left(1 - \frac{M}{2p}\right)^2\left[\left(1 - \frac{M}{p}\right)\left(2 - \frac{n}{p}\right)\frac{n}{p} + \frac{M}{p}\right] + \frac{M}{p} - \frac{M^2}{4p^2}\right\}\|\boldsymbol{w}_1^* - \boldsymbol{w}_2^*\|^2$$

$$+ \frac{1}{3}\left[\left(1 - \frac{M}{2p}\right)^2\frac{n}{p} + \left(1 - \frac{M}{2p}\right)^2\left(1 - \frac{n}{p}\right)\frac{M}{p} + \left(1 - \frac{n}{p}\right)^2\left(1 - \frac{M}{p}\right)\left(1 - \frac{M}{2p}\right)^2\frac{n}{p}\right.$$

$$\left. + \left(1 - \frac{M}{2p}\right)\frac{M}{2p}\right]\|\boldsymbol{w}_1^* - \boldsymbol{w}_3^*\|^2$$

$$+ \frac{1}{3}\left\{\left(1 - \frac{M}{2p}\right)^2\frac{n}{p}\left[\left(1 - \frac{M}{p}\right)\left(1 - \frac{n}{p}\right) + 1\right] + \frac{M}{2p}\right\}\|\boldsymbol{w}_2^* - \boldsymbol{w}_3^*\|^2. \tag{43}$$

### E.1. Comparison of Forgetting When $T = 3$

By observing Equation (40) and Equation (42), the expressions of forgetting for both rehearsal methods share the same structure:

$$F_3 = \frac{1}{2}\hat{c}_1 \|\boldsymbol{w}_1^*\|^2 + \frac{1}{2}\hat{c}_2 \|\boldsymbol{w}_2^*\|^2 + \frac{1}{2}\hat{c}_3 \|\boldsymbol{w}_1^* - \boldsymbol{w}_2^*\|^2 + \frac{1}{2}\hat{c}_4 \|\boldsymbol{w}_1^* - \boldsymbol{w}_3^*\|^2 + \frac{1}{2}\hat{c}_5 \|\boldsymbol{w}_2^* - \boldsymbol{w}_3^*\|^2 .$$

By comparing Equation (40) and Equation (42), we have the following conclusions: $1.\hat{c}_1^{(\text{concurrent})} < \hat{c}_1^{(\text{sequential})}$; $2.\hat{c}_2^{(\text{concurrent})} < \hat{c}_2^{(\text{sequential})}$; $3.\hat{c}_3^{(\text{concurrent})} > \hat{c}_3^{(\text{sequential})}$, when $p > \frac{5n+4M}{2}$; $4.\hat{c}_4^{(\text{concurrent})} > \hat{c}_4^{(\text{sequential})}$; $5.\hat{c}_5^{(\text{concurrent})} > \hat{c}_5^{(\text{sequential})}$. The proof of these conclusions is provided as follows.

*Proof.* 1. $\hat{c}_1^{(\text{concurrent})} < \hat{c}_1^{(\text{sequential})}$. We have:

$$\hat{c}_1^{(\text{sequential})} = \left[\left(1 - \frac{n}{p}\right)^3 \left(1 - \frac{M}{p}\right)\left(1 - \frac{M}{2p}\right)^2 - \left(1 - \frac{n}{p}\right)\right]$$

$$= \left[\left(1 - \frac{n}{p}\right)^2 \left(1 - \frac{M}{p}\right)\left(1 - \frac{M}{2p}\right)^2 - 1\right]\left(1 - \frac{n}{p}\right)$$

$$> \left[\left(1 - \frac{n}{p}\right)^2 \left(1 - \frac{M}{p}\right)^2 - 1\right]\left(1 - \frac{n}{p}\right)$$

$$> \left[\left(1 - \frac{n+M}{p}\right)^2 - 1\right]\left(1 - \frac{n}{p}\right)$$

$$= \hat{c}_1^{(\text{concurrent})}.$$

2. $\hat{c}_2^{(\text{concurrent})} < \hat{c}_2^{(\text{sequential})}$. Consider:

$$\hat{c}_2^{(\text{sequential})} = \left[\left(1 - \frac{n}{p}\right)^3 \left(1 - \frac{M}{p}\right)\left(1 - \frac{M}{2p}\right)^2 - \left(1 - \frac{n}{p}\right)^2\left(1 - \frac{M}{p}\right)\right]$$

$$> \left[\left(1 - \frac{n}{p}\right)^3 \left(1 - \frac{M}{p}\right)^2 - \left(1 - \frac{n}{p}\right)^2\left(1 - \frac{M}{p}\right)\right]$$

$$= \left(1 - \frac{n}{p}\right)\left[\left(1 - \frac{n}{p}\right)\left(1 - \frac{M}{p}\right) - 1\right]\left(1 - \frac{n}{p}\right)\left(1 - \frac{M}{p}\right)$$

$$= \left(1 - \frac{n}{p}\right)\left[\frac{nM}{p^2} - \frac{n+M}{p}\right]\left(1 - \frac{n+M}{p} + \frac{nM}{p^2}\right)$$

$$= \left(1 - \frac{n}{p}\right)\left[\frac{nM}{p^2} - \frac{n+M}{p}\right]\left(1 - \frac{n+M}{p} + \frac{nM}{p^2}\right)$$

$$= \left(1 - \frac{n}{p}\right)\left[-\frac{n+M}{p}\right]\left(1 - \frac{n+M}{p}\right) + \left(1 - \frac{n}{p}\right)\frac{nM}{p^2}\left(1 - \frac{2(n+M)}{p} + \frac{nM}{p^2}\right)$$

$$> \left(1 - \frac{n}{p}\right)\left[-\frac{n+M}{p}\right]\left(1 - \frac{n+M}{p}\right)$$

$$= \hat{c}_2^{(\text{concurrent})}.$$

3. $\hat{c}_3^{(\text{concurrent})} > \hat{c}_3^{(\text{sequential})}$ when $p > \frac{5n+4M}{2}$. We first lower bound $\hat{c}_3^{(\text{concurrent})}$ as follows.

$$\hat{c}_3^{(\text{concurrent})} = \left(1 - \frac{2(n+M)}{p}\right)\frac{nM}{p(p-n-M-1)} + \frac{M^2}{2p(p-n-M-1)} + \frac{n+M}{p}\left(1 - \frac{n}{p}\right)\left(1 - \frac{n+M}{p}\right)$$

$$> \left(1 - \frac{2(n+M)}{p}\right)\frac{nM}{p^2} + \frac{M^2}{2p^2} + \frac{n+M}{p}\left(1 - \frac{n}{p}\right)\left(1 - \frac{n+M}{p}\right)$$

$$= \frac{n+M}{p}\left(1-\frac{n}{p}\right)\left(1-\frac{M}{p}\right) - \frac{n^2}{p^2} + \frac{n^3 - n^2 M - 2nM^2}{p^3}.$$

On the other hand, we upper bound $\hat{c}_3^{(\text{sequential})}$ as follows.

$$\hat{c}_3^{(\text{sequential})} = \left(1-\frac{n}{p}\right)\left(1-\frac{M}{p}\right)\frac{n}{p}\left(\left(1-\frac{M}{2p}\right)^2\left(2-\frac{n}{p}\right)-1\right) + \left(1-\frac{M}{2p}\right)^2\left(1-\frac{n}{p}\right)\frac{M}{p} - \frac{M^2}{4p^2}$$

$$= \left(1-\frac{n}{p}\right)\left(1-\frac{M}{p}\right)\frac{n}{p}\left(\left(1-\frac{M}{p}\right)\left(2-\frac{n}{p}\right)-1\right) + \left(1-\frac{M}{p}\right)\left(1-\frac{n}{p}\right)\frac{M}{p}$$

$$\quad + \frac{M^2}{4p^2}\left[\left(2-\frac{n}{p}\right)\left(1-\frac{M}{p}\right)\left(1-\frac{n}{p}\right)\frac{n}{p} + \left(1-\frac{n}{p}\right)\frac{M}{p}-1\right]$$

$$< \left(1-\frac{n}{p}\right)\left(1-\frac{M}{p}\right)\frac{n}{p}\left(\left(1-\frac{M}{p}\right)\left(2-\frac{n}{p}\right)-1\right) + \left(1-\frac{M}{p}\right)\left(1-\frac{n}{p}\right)\frac{M}{p} + \frac{M^2}{4p^2}\left[\frac{2n}{p}+\frac{M}{p}-1\right]$$

$$\overset{(i)}{<} \left(1-\frac{n}{p}\right)\left(1-\frac{M}{p}\right)\frac{n}{p}\left(\left(1-\frac{M}{p}\right)\left(2-\frac{n}{p}\right)-1\right) + \left(1-\frac{M}{p}\right)\left(1-\frac{n}{p}\right)\frac{M}{p}$$

$$= \left(1-\frac{n}{p}\right)\left(1-\frac{M}{p}\right)\frac{n}{p}\left(1-\frac{2M}{p}-\frac{n}{p}+\frac{nM}{p^2}\right) + \left(1-\frac{M}{p}\right)\left(1-\frac{n}{p}\right)\frac{M}{p}$$

$$= \left(1-\frac{n}{p}\right)\left(1-\frac{M}{p}\right)\frac{n+M}{p} + \left(1-\frac{n}{p}\right)\left(1-\frac{M}{p}\right)\frac{n}{p}\left(-\frac{n+2M}{p}+\frac{nM}{p^2}\right)$$

$$\overset{(ii)}{<} \left(1-\frac{n}{p}\right)\left(1-\frac{M}{p}\right)\frac{n+M}{p} + \left(1-\frac{n+M}{p}\right)\frac{n}{p}\left(-\frac{n+2M}{p}+\frac{nM}{p^2}\right)$$

$$< \left(1-\frac{n}{p}\right)\left(1-\frac{M}{p}\right)\frac{n+M}{p} - \frac{n^2+2nM}{p^2} + \frac{n^3+4n^2 M+2nM^2}{P^3},$$

where $(i)$ follows from the face that $p > \frac{5n+4M}{2}$ and $(ii)$ follows from the fact that $-\frac{n+2M}{p}+\frac{nM}{p^2} < 0$. Since $p > \frac{5n+4M}{2}$, we have:

$$-\frac{n^2}{p^2} + \frac{n^3 - n^2 M - 2nM^2}{p^3} > -\frac{n^2+2nM}{p^2} + \frac{n^3+4n^2 M+2nM^2}{P^3}$$

which implies $\hat{c}_3^{(\text{concurrent})} > \hat{c}_3^{(\text{sequential})}$ and completes the proof.

4.$\hat{c}_4^{(\text{concurrent})} > \hat{c}_4^{(\text{sequential})}$. Consider:

$$\hat{c}_4^{(\text{concurrent})} = \frac{n}{p} + \frac{nM}{p(p-n-M-1)} > \frac{n}{p} > \left(1-\frac{M}{2p}\right)^2\frac{n}{p} = \hat{c}_4^{(\text{sequential})}.$$

5. The proof of $\hat{c}_5^{(\text{concurrent})} > \hat{c}_5^{(\text{sequential})}$ is the same as $\hat{c}_4^{(\text{concurrent})} > \hat{c}_4^{(\text{sequential})}$.

### E.2. Comparison of Generalization Error When $T = 3$

By observing Equation (41) and Equation (43), the expressions of generalization error for both rehearsal methods share the same structure:

$$G_3 = \frac{1}{3}\hat{d}_1(\|\boldsymbol{w}_1^*\|^2 + \|\boldsymbol{w}_2^*\|^2 + \|\boldsymbol{w}_3^*\|^2) + \frac{1}{3}\hat{d}_2\|\boldsymbol{w}_1^* - \boldsymbol{w}_2^*\|^2 + \frac{1}{3}\hat{d}_3\|\boldsymbol{w}_1^* - \boldsymbol{w}_3^*\|^2 + \frac{1}{3}\hat{d}_4\|\boldsymbol{w}_2^* - \boldsymbol{w}_3^*\|^2.$$

By comparing Equation (41) and Equation (43), we have the following conclusions: 1.$\hat{d}_1^{(\text{concurrent})} < \hat{d}_1^{(\text{sequential})}$; 2.$\hat{d}_2^{(\text{concurrent})} > \hat{d}_2^{(\text{sequential})}$ when $p > \frac{4n+3M}{2}$; 3.$\hat{d}_3^{(\text{concurrent})} > \hat{d}_3^{(\text{sequential})}$; 4.$\hat{d}_4^{(\text{concurrent})} > \hat{d}_4^{(\text{sequential})}$. The proof of these relationships is provided as follows.

1. $\hat{d}_1^{(\text{concurrent})} < \hat{d}_1^{(\text{sequential})}$:

$$\hat{d}_1^{(\text{sequential})} = \left(1-\frac{n}{p}\right)^3\left(1-\frac{M}{p}\right)\left(1-\frac{M}{2p}\right)^2$$

$$> \left(1 - \frac{n}{p}\right)^3 \left(1 - \frac{M}{p}\right)^2$$

$$> \left(1 - \frac{n+M}{p}\right)^2 \left(1 - \frac{M}{p}\right)$$

$$= \hat{d}_1^{(\text{concurrent})}.$$

2. $\hat{d}_2^{(\text{concurrent})} > \hat{d}_2^{(\text{sequential})}$ when $p > \frac{4n+3M}{2}$. We first lower bound $\hat{d}_2^{(\text{concurrent})}$ as follows.

$\hat{d}_2^{(\text{concurrent})}$

$$= \left(3 - \frac{3(n+M)}{p}\right) \frac{nM}{p(p-n-M-1)} + \frac{3M^2}{4p(p-n-M-1)} + \frac{n+M}{p}\left(2 - \frac{3n}{p} - \frac{M}{p} + \frac{n(n+M)}{p^2}\right)$$

$$> \left(3 - \frac{3(n+M)}{p}\right) \frac{nM}{p^2} + \frac{3M^2}{4p^2} + \frac{n+M}{p}\left(2 - \frac{3n}{p} - \frac{M}{p} + \frac{n(n+M)}{p^2}\right)$$

$$> 3\left(1 - \frac{n+M}{p}\right) \frac{nM}{p^2} + \frac{2(n+M)}{p} + \frac{n+M}{p}\left(-\frac{3n}{p} - \frac{n}{p} + \frac{n(n+M)}{p^2}\right)$$

$$= \frac{2(n+M)}{p} - \frac{3n^2 + nM + M^2}{p^2} + \frac{n^3 - n^2M - 2nM^2}{p^3}.$$

On the other hand, we upper bound $\hat{d}_2^{(\text{sequential})}$ as follows.

$\hat{d}_2^{(\text{sequential})}$

$$= \left(1 - \frac{n}{p}\right)\left(1 - \frac{M}{2p}\right)^2 \left[\left(1 - \frac{M}{p}\right)\left(2 - \frac{n}{p}\right)\frac{n}{p} + \frac{M}{p}\right] + \frac{M}{p} - \frac{M^2}{4p^2}$$

$$= \left(1 - \frac{n}{p}\right)\left(1 - \frac{M}{p} + \frac{M^2}{4p^2}\right)\left[\left(1 - \frac{M}{p}\right)\left(2 - \frac{n}{p}\right)\frac{n}{p} + \frac{M}{p}\right] + \frac{M}{p} - \frac{M^2}{4p^2}$$

$$= \left(1 - \frac{n}{p}\right)\left(1 - \frac{M}{p}\right)\left[\left(1 - \frac{M}{p}\right)\left(2 - \frac{n}{p}\right)\frac{n}{p} + \frac{M}{p}\right] + \frac{M}{p} + \frac{M^2}{4p^2}\left[\left(1 - \frac{M}{p}\right)\left(2 - \frac{n}{p}\right)\frac{n}{p} + \frac{M}{p}\right] - \frac{M^2}{4p^2}$$

$$< \left(1 - \frac{n}{p}\right)\left(1 - \frac{M}{p}\right)\left[\left(1 - \frac{M}{p}\right)\left(2 - \frac{n}{p}\right)\frac{n}{p} + \frac{M}{p}\right] + \frac{M}{p} + \frac{M^2}{4p^2}\left[\frac{2n}{p} + \frac{M}{p} - 1\right]$$

$$< \left(1 - \frac{n}{p}\right)\left(1 - \frac{M}{p}\right)\left[\left(2 - \frac{n}{p}\right)\frac{n}{p} + \frac{M}{p}\right] + \frac{M}{p}$$

$$= \left(1 - \frac{n}{p}\right)\left(1 - \frac{M}{p}\right)\frac{2n}{p} - \left(1 - \frac{n}{p}\right)\left(1 - \frac{M}{p}\right)\frac{n^2}{p^2} + \frac{2M}{p} + \left(-\frac{n+M}{p} + \frac{nM}{p^2}\right)\frac{M}{p}$$

$$= \frac{2(n+M)}{p} - \frac{3n^2 + 3nM + M^2}{p^2} + \frac{n^3 + 3n^2M + nM^2}{p^3} - \frac{n^3M}{p^4}$$

$$< \frac{2(n+M)}{p} - \frac{3n^2 + 3nM + M^2}{p^2} + \frac{n^3 + 3n^2M + nM^2}{p^3}.$$

Since $p > \frac{4n+3M}{2}$, we have:

$$-\frac{3n^2 + nM + M^2}{p^2} + \frac{n^3 - n^2M - 2nM^2}{p^3} > -\frac{3n^2 + 3nM + M^2}{p^2} + \frac{n^3 + 3n^2M + nM^2}{p^3}$$

which implies $\hat{d}_2^{(\text{concurrent})} > \hat{d}_2^{(\text{sequential})}$ and completes the proof.

3. $\hat{d}_3^{(\text{concurrent})} > \hat{d}_3^{(\text{sequential})}$. We first lower bound $\hat{d}_3^{(\text{concurrent})}$ as follows.

$$\hat{d}_3^{(\text{concurrent})} = \frac{n}{p}\left(2 - \frac{2(n+M)}{p} + \frac{(n+M)^2}{p^2}\right) + \frac{M}{p}\left(1 - \frac{n+M}{p}\right) + \frac{M}{2p} + \frac{3nM}{2p(p-n-M-1)}$$

$$> \frac{n}{p}\left(2 - \frac{2(n+M)}{p} + \frac{(n+M)^2}{p^2}\right) + \frac{M}{p}\left(1 - \frac{n+M}{p}\right) + \frac{M}{2p} + \frac{3nM}{2p^2}.$$

On the other hand, we upper bound $\hat{d}_3^{(\text{sequential})}$ as follows.

$$\hat{d}_3^{(\text{sequential})}$$

$$= \left(1 - \frac{M}{2p}\right)^2 \frac{n}{p} + \left(1 - \frac{M}{2p}\right)^2\left(1 - \frac{n}{p}\right)\frac{M}{p} + \left(1 - \frac{n}{p}\right)^2\left(1 - \frac{M}{p}\right)\left(1 - \frac{M}{2p}\right)^2\frac{n}{p} + \left(1 - \frac{M}{2p}\right)\frac{M}{2p}$$

$$< \left(1 - \frac{M}{2p}\right)^2 \frac{n}{p}\left[1 + \left(1 - \frac{n}{p}\right)^2\left(1 - \frac{M}{p}\right)\right] + \left(1 - \frac{n}{p}\right)\left(1 - \frac{M}{p}\right)\frac{M}{p} + \frac{M^3}{4p^3} + \frac{M}{2p}$$

$$< \frac{n}{p}\left(2 - \frac{2n+M}{p} + \frac{n^2 + 2nM}{p^2}\right) + \frac{n}{p}\left(-\frac{M}{p} + \frac{M^2}{4p^2}\right) + \frac{M}{p}\left(1 - \frac{n+M}{p}\right) + \frac{nM^2}{p} + \frac{M^3}{4p^3} + \frac{M}{2p}$$

$$= \frac{n}{p}\left(2 - \frac{2n+2M}{p} + \frac{n^2 + 2nM + M^2}{p^2}\right) + \frac{M}{p}\left(1 - \frac{n+M}{p}\right) + \frac{nM^2 + M^3}{4p^3} + \frac{M}{2p}$$

$$< \frac{n}{p}\left(2 - \frac{2(n+M)}{p} + \frac{(n+M)^2}{p^2}\right) + \frac{M}{p}\left(1 - \frac{n+M}{p}\right) + \frac{M}{2p} + \frac{3nM}{2p^2}.$$

By combining the above equations, we complete the proof.

4. $\hat{d}_4^{(\text{concurrent})} > \hat{d}_4^{(\text{sequential})}$. Consider:

$$\hat{d}_4^{(\text{sequential})} = \left(1 - \frac{M}{2p}\right)^2 \frac{n}{p}\left[\left(1 - \frac{M}{p}\right)\left(1 - \frac{n}{p}\right) + 1\right] + \frac{M}{2p}$$

$$< \frac{n}{p}\left[\left(1 - \frac{M}{p}\right)\left(1 - \frac{n}{p}\right) + 1\right] + \frac{M}{2p}$$

$$= \frac{n}{p}\left[2 - \frac{n+M}{p}\right] + \frac{M}{2p} + \frac{n^2 M}{p^3}$$

$$< \frac{n}{p}\left[2 - \frac{n+M}{p}\right] + \frac{M}{2p} + \frac{3nM}{2p(p - n - M - 1)}$$

$$< \hat{d}_4^{(\text{concurrent})}.$$

## F. Proof of Lemma 5.4

In this section, we prove Lemma 5.4, which helps to further compare the performance between concurrent and sequential rehearsal methods for general $T$. We assume that $M \geq 2$. We first prove coefficients $d_{0T}, d_{ijkT}$ in Appendix F.1 , and then prove the coefficients $c_i, c_{ijk}$ in Appendix F.2.

### F.1. Proof of Coefficients $d_{0T}, d_{ijkT}$ in Lemma 5.4

In this subsection, we will compare the coefficients $d_{0T}, d_{ijkT}$ under different rehearsal methods. We first fix the index $i$, which implies that we consider the generalization error on the previous task $i$.

1. We first prove $d_{0T}^{(\text{concurrent})} < d_{0T}^{(\text{sequential})}$. According to Lemma B.11, we have:

$$d_{0T}^{(\text{concurrent})} = \left(1 - \frac{n}{p}\right)\left(1 - \frac{n+M}{p}\right)^{T-1}$$

$$< \left(1 - \frac{n}{p}\right)\prod_{l=0}^{T-2}\left[\left(1 - \frac{M}{(T-l-1)p}\right)^{T-l-1}\left(1 - \frac{n}{p}\right)\right]$$

$$= d_{0T}^{(\text{sequential})}$$

2. Now, we prove $d_{i1iT}^{(\text{concurrent})} > d_{i1iT}^{(\text{sequential})}$ if $p > 2T^4(n+M)nM$. We first consider:

$$\frac{n}{p} \prod_{l=0}^{T-2} \left[ \left(1 - \frac{M}{(T-l-1)p}\right)^{T-l-1} \left(1 - \frac{n}{p}\right) \right] \overset{(i)}{<} \frac{n}{p} \left(1 - \frac{n+M}{p} + \frac{(n+M)M}{p^2}\right)^{T-1}$$

$$\overset{(ii)}{<} \frac{n}{p} \left(1 - \frac{n+M}{p}\right)^{T-1} + \frac{T^2(n+M)nM}{p^3}, \qquad (44)$$

where $(i)$ follows from Lemma B.12 and $(ii)$ follows from Lemma B.13. We also notice that:

$$\sum_{l=0}^{T-2} \prod_{k=0}^{l-1} \left[ \left(1 - \frac{M}{(T-k-1)p}\right)^{T-k-1} \left(1 - \frac{n}{p}\right) \right] \left(1 - \frac{M}{(T-l-1)p}\right)^{T-l-2} \frac{M}{(T-l-1)p}$$

$$= \sum_{l=0}^{T-3} \prod_{k=0}^{l-1} \left[ \left(1 - \frac{M}{(T-k-1)p}\right)^{T-k-1} \left(1 - \frac{n}{p}\right) \right] \left(1 - \frac{M}{(T-l-1)p}\right)^{T-l-2} \frac{M}{(T-l-1)p}$$

$$+ \prod_{k=0}^{T-3} \left[ \left(1 - \frac{M}{(T-k-1)p}\right)^{T-k-1} \left(1 - \frac{n}{p}\right) \right] \left(1 - \frac{M}{p}\right) \frac{M}{p}$$

$$\overset{(i)}{<} \left(1 - \frac{1}{Tp}\right) \sum_{l=0}^{T-3} \left(1 - \frac{n+M}{p}\right)^{l} \frac{M}{(T-l-1)p} + \left(1 - \frac{n+M}{p} + \frac{(n+M)M}{p^2}\right)^{T-2} \left(1 - \frac{M}{p}\right) \frac{M}{p}$$

$$\overset{(ii)}{<} \left(1 - \frac{1}{Tp}\right) \sum_{l=0}^{T-3} \left(1 - \frac{n+M}{p}\right)^{l} \frac{M}{(T-l-1)p} + \left[ \left(1 - \frac{n+M}{p}\right)^{T-2} + \frac{T^2(n+M)M}{p^2} \right] \left(1 - \frac{M}{p}\right) \frac{M}{p}$$

$$< \sum_{l=0}^{T-2} \left(1 - \frac{n+M}{p}\right)^{l} \frac{M}{(T-l-1)p} - \frac{M}{T^2p^2} + \frac{T^2(n+M)M^2}{p^3}, \qquad (45)$$

where $(i)$ follows from Lemmas B.12 and B.15 and $(ii)$ follows from Lemma B.13. By combining Equations (44) and (45), we can conclude:

$$d_{i1iT}^{(\text{sequential})} < \frac{n}{p} \left(1 - \frac{n+M}{p}\right)^{T-1} + \sum_{l=0}^{T-2} \left(1 - \frac{n+M}{p}\right)^{l} \frac{M}{(T-l-1)p} + \frac{T^2(n+M)nM}{p^3} - \frac{M}{T^2p^2} + \frac{T^2(n+M)M^2}{p^3}$$

$$\overset{(i)}{<} \frac{n+M}{p} \left(1 - \frac{n+M}{p}\right)^{T-1} + \sum_{l=0}^{T-2} \left(1 - \frac{n+M}{p}\right)^{l} \frac{M}{(T-l-1)p}$$

$$= d_{i1iT}^{(\text{concurrent})}, \qquad (46)$$

where $(i)$ follows from the fact that $p > 2T^4(n+M)nM$.

3. Next, we prove $d_{ijiT}^{(\text{concurrent})} > d_{ijiT}^{(\text{sequential})}$ if $p > T^4(n+M)M$, for $j = 2, 3, ..., T-1$. We first have:

$$\sum_{l=0}^{T-j-1} \prod_{k=0}^{l-1} \left[ \left(1 - \frac{M}{(T-k-1)p}\right)^{T-k-1} \left(1 - \frac{n}{p}\right) \right] \left(1 - \frac{M}{(T-l-1)p}\right)^{T-j-l-1} \frac{M}{(T-l-1)p}$$

$$= \sum_{l=0}^{T-j-2} \prod_{k=0}^{l-1} \left[ \left(1 - \frac{M}{(T-k-1)p}\right)^{T-k-1} \left(1 - \frac{n}{p}\right) \right] \left(1 - \frac{M}{(T-l-1)p}\right)^{T-j-l-1} \frac{M}{(T-l-1)p}$$

$$+ \prod_{k=0}^{T-j-2} \left[ \left(1 - \frac{M}{(T-k-1)p}\right)^{T-k-1} \left(1 - \frac{n}{p}\right) \right] \frac{M}{jp}$$

$$\overset{(i)}{<} \left(1 - \frac{1}{Tp}\right) \sum_{l=0}^{T-j-2} \left(1 - \frac{n+M}{p}\right)^{l} \frac{M}{(T-l-1)p} + \left(1 - \frac{n+M}{p} + \frac{(n+M)M}{p^2}\right)^{T-j-1} \frac{M}{jp}$$

$$\overset{(ii)}{<} \left(1 - \frac{1}{Tp}\right) \sum_{l=0}^{T-j-2} \left(1 - \frac{n+M}{p}\right)^{l} \frac{M}{(T-l-1)p} + \left(1 - \frac{n+M}{p}\right)^{T-j-1} \frac{M}{jp} + \frac{T^2(n+M)M^2}{jp^3}$$

$$< \sum_{l=0}^{T-j-1} \left(1 - \frac{n+M}{p}\right)^l \frac{M}{(T-l-1)p} - \frac{M}{T^2 p^2} + \frac{T^2(n+M)M^2}{p^3} \tag{47}$$

where $(i)$ follows from Lemmas B.12 and B.15, $(ii)$ follows Lemma B.13. Therefore, if $p > T^4(n+M)M$, we have:

$$\sum_{l=0}^{T-j-1} \prod_{k=0}^{l-1} \left[\left(1 - \frac{M}{(T-k-1)p}\right)^{T-k-1}\left(1 - \frac{n}{p}\right)\right]\left(1 - \frac{M}{(T-l-1)p}\right)^{T-j-l-1} \frac{M}{(T-l-1)p}$$

$$< \sum_{l=0}^{T-j-1} \left(1 - \frac{n+M}{p}\right)^l \frac{M}{(T-l-1)p}. \tag{48}$$

Furthermore, we have:

$$\prod_{k=0}^{T-j-1}\left[\left(1 - \frac{M}{(T-l-1)p}\right)^{T-k-1}\left(1 - \frac{n}{p}\right)\right]\left(1 - \frac{M}{(j-1)p}\right)^{j-1}\frac{n}{p} \overset{(i)}{<} \left(1 - \frac{n+M}{p}\right)^{T-j}\frac{n}{p} \tag{49}$$

where $(i)$ follows from Lemmas B.12 and B.15. Therefore, by combining Equations (48) and (49), we have:

$$d_{ijiT}^{\text{(sequential)}} < \sum_{l=0}^{T-j-1}\left(1 - \frac{n+M}{p}\right)^l \frac{M}{(T-l-1)p} + \left(1 - \frac{n+M}{p}\right)^{T-j}\frac{n}{p} \le d_{ijiT}^{\text{(concurrent)}}. \tag{50}$$

4. Last, we prove $d_{iTiT}^{\text{(concurrent)}} > d_{iTiT}^{\text{(sequential)}}$. The proof is straightforward:

$$d_{iTiT}^{\text{(sequential)}} = \left(1 - \frac{M}{(T-1)p}\right)^{T-1}\frac{n}{p} < \frac{n}{p} \le d_{iTiT}^{\text{(concurrent)}}.$$

5. Moreover, for the other choices of $j, k$ we have $d_{ijkT}^{\text{(concurrent)}} \ge 0$ and $d_{ijkT}^{\text{(sequential)}} = 0$.

### F.2. Proof of Coefficients $c_i, c_{ijk}$ in Lemma 5.4

In this subsection, we will compare the coefficients $c_i, c_{ijk}$ under different rehearsal methods. Before we start, we first provide some important observation about the terms $\beta_1$ and $\beta_2$ in Equation (32). We split the term $\beta_1$ into two parts.

$$\beta_1 = \left.\sum_{l=0}^{T-i-1}\left(1 - \frac{n+M}{p}\right)^l \sum_{j=1}^{T-l-2}\sum_{k=j+1}^{T-l-1}\frac{(\frac{M}{T-l-1})^2}{p(p-n-M-1)}\left\|w_j^* - w_k^*\right\|^2\right\}\beta_1^+$$

$$+ \left.\begin{aligned}&\sum_{l=T-i}^{T-2}\left(1 - \frac{n+M}{p}\right)^l \sum_{j=1}^{T-l-2}\sum_{k=j+1}^{T-l-1}\frac{(\frac{M}{T-l-1})^2}{p(p-n-M-1)}\left\|w_j^* - w_k^*\right\|^2\\ &-\sum_{l=0}^{i-2}\left(1 - \frac{n+M}{p}\right)^l \sum_{j=1}^{i-l-2}\sum_{k=j+1}^{i-l-1}\frac{(\frac{M}{i-l-1})^2}{p(p-n-M-1)}\left\|w_j^* - w_k^*\right\|^2\end{aligned}\right\}\beta_1^-. \tag{51}$$

Since $T - i - 1 \ge 0$, we notice that $\beta_1^+$ consists of terms $\delta_{j,k}^+\left\|w_j^* - w_k^*\right\|^2$ where $\delta_{j,k}^+ \ge \left(1 - \frac{n+M}{p}\right)\frac{M^2}{T^2 p^2}$ for any $j, k \in [T-1]$ and $j < k$. For the term $\beta_1^-$, we have:

$$\beta_1^- = \sum_{l=0}^{i-2}\left(1 - \frac{n+M}{p}\right)^{T-i+l}\sum_{j=1}^{i-l-2}\sum_{k=j+1}^{i-l-1}\frac{(\frac{M}{i-l-1})^2}{p(p-n-M-1)}\left\|w_j^* - w_k^*\right\|^2$$

$$-\sum_{l=0}^{i-2}\left(1 - \frac{n+M}{p}\right)^l \sum_{j=1}^{i-l-2}\sum_{k=j+1}^{i-l-1}\frac{(\frac{M}{i-l-1})^2}{p(p-n-M-1)}\left\|w_j^* - w_k^*\right\|^2$$

$$=\sum_{l=0}^{i-2}\left[\left(1 - \frac{n+M}{p}\right)^{T-i} - 1\right]\left(1 - \frac{n+M}{p}\right)^l \sum_{j=1}^{i-l-2}\sum_{k=j+1}^{i-l-1}\frac{(\frac{M}{i-l-1})^2}{p(p-n-M-1)}\left\|w_j^* - w_k^*\right\|^2$$

$$\geq -\frac{T(n+M)}{p} \sum_{l=0}^{i-2} \sum_{j=1}^{i-l-2} \sum_{k=j+1}^{i-l-1} \frac{(\frac{M}{i-l-1})^2}{p(p-n-M-1)} \left\| \boldsymbol{w}_j^* - \boldsymbol{w}_k^* \right\|^2 . \tag{52}$$

The above argument shows that $\beta_1^-$ consists of terms $\delta_{j,k}^- \left\| \boldsymbol{w}_j^* - \boldsymbol{w}_k^* \right\|^2$ where $\delta_{j,k}^- \geq -\frac{T^2(n+M)M^2}{p^3}$ for $j,k \in [i-1]$ and $j < k$. Therefore, if $p > (T^4+1)(n+M)$, we can conclude that $\beta_1$ consists of terms $\delta_{j,k} \left\| \boldsymbol{w}_j^* - \boldsymbol{w}_k^* \right\|^2$ for $j,k \in [T-1]$ and $j < k$, where

$$\delta_{j,k} \geq \left(1 - \frac{n+M}{p}\right) \frac{M^2}{T^2 p^2} - \frac{T^2(n+M)M^2}{p^3} > 0, \tag{53}$$

By the same argument, we have:

$$
\begin{aligned}
\beta_2 = &\left. \sum_{l=0}^{T-i-1} \left(1 - \frac{n+M}{p}\right)^l \sum_{j=1}^{T-l-1} \frac{\frac{nM}{T-l-1}}{p(p-n-M-1)} \left\| \boldsymbol{w}_j^* - \boldsymbol{w}_{T-l}^* \right\|^2 \right\} \beta_2^+ \\
&\left. \begin{aligned} + &\sum_{l=T-i}^{T-2} \left(1 - \frac{n+M}{p}\right)^l \sum_{j=1}^{T-l-1} \frac{\frac{nM}{T-l-1}}{p(p-n-M-1)} \left\| \boldsymbol{w}_j^* - \boldsymbol{w}_{T-l}^* \right\|^2 \\ - &\sum_{l=0}^{i-2} \left(1 - \frac{n+M}{p}\right)^l \sum_{j=1}^{i-l-1} \frac{\frac{nM}{i-l-1}}{p(p-n-M-1)} \left\| \boldsymbol{w}_j^* - \boldsymbol{w}_{i-l}^* \right\|^2 \end{aligned} \right\} \beta_2^- ,
\end{aligned}
\tag{54}
$$

where $\beta_2^+$ consists of terms $\eta_{j,k}^+ \left\| \boldsymbol{w}_j^* - \boldsymbol{w}_k^* \right\|^2$ with $\eta_{j,k}^+ \geq \left(1 - \frac{n+M}{p}\right) \frac{nM}{Tp^2}$ for $j,k \in [T-1]$ and $j < k$ and $\beta_2^-$ consists of terms $\eta_{j,k}^- \left\| \boldsymbol{w}_j^* - \boldsymbol{w}_k^* \right\|^2$ with $\eta_{j,k}^- \geq -\frac{T^2(n+M)nM}{p^3}$ for $j,k \in [i-1]$ and $j < k$. Therefore, if $p > (T^4+1)(n+M)M$, we conclude that $\beta_2$ consists of terms $\eta_{j,k} \left\| \boldsymbol{w}_j^* - \boldsymbol{w}_k^* \right\|^2$ for $j \in [k-1], k = 2,3,..,i$ where

$$\eta_{j,k} \geq \left(1 - \frac{n+M}{p}\right) \frac{nM}{Tp^2} - \frac{T^2(n+M)nM}{p^3} > 0. \tag{55}$$

Now, we start to coefficients $c_i, c_{ijk}$ in Lemma 5.4. We first fix the index $i$, which means that we consider the forgetting on the previous task $i$. The proof of $c_i^{(\text{concurrent})} < c_i^{(\text{sequential})}$ if $p > 2T^3(n+M)^2$ follows from Lemma B.16 directly.

The proof of $c_{ijk}^{(\text{concurrent})} > c_{ijk}^{(\text{sequential})}$ are as follows.

1. We prove $c_{i1i}^{(\text{concurrent})} > c_{i1i}^{(\text{sequential})}$ if $p > 3T^4(n+M)nM$. We first upper bound part of the coefficient $c_{i1i}^{(\text{sequential})}$:

$$
\begin{aligned}
&\frac{n}{p} \left\{ \prod_{l=0}^{T-2} \left[ \left(1 - \frac{M}{(T-l-1)p}\right)^{T-l-1} \left(1 - \frac{n}{p}\right) \right] - \prod_{l=0}^{i-2} \left[ \left(1 - \frac{M}{(i-l-1)p}\right)^{i-l-1} \left(1 - \frac{n}{p}\right) \right] \right\} \\
&\overset{(i)}{<} \frac{n}{p} \left[ \left(1 - \frac{n+M}{p}\right)^{T-1} - \left(1 - \frac{n+M}{p}\right)^{i-1} \right] + \frac{T^2(n+M)nM}{p^3}
\end{aligned}
\tag{56}
$$

where $(i)$ follows from Lemma B.17. We then upper bound the other part of $c_{i1i}^{(\text{sequential})}$ as follows.

$$
\begin{aligned}
&\sum_{l=0}^{T-2} \prod_{k=0}^{l-1} \left[ \left(1 - \frac{M}{(T-k-1)p}\right)^{T-k-1} \left(1 - \frac{n}{p}\right) \right] \left(1 - \frac{M}{(T-l-1)p}\right)^{T-l-2} \frac{M}{(T-l-1)p} \\
&- \sum_{l=0}^{i-2} \prod_{k=0}^{l-1} \left[ \left(1 - \frac{M}{(i-k-1)p}\right)^{i-k-1} \left(1 - \frac{n}{p}\right) \right] \left(1 - \frac{M}{(i-l-1)p}\right)^{i-l-2} \frac{M}{(i-l-1)p} \\
&= \sum_{l=0}^{T-i-1} \prod_{k=0}^{l-1} \left[ \left(1 - \frac{M}{(T-k-1)p}\right)^{T-k-1} \left(1 - \frac{n}{p}\right) \right] \left(1 - \frac{M}{(T-l-1)p}\right)^{T-l-2} \frac{M}{(T-l-1)p} \\
&+ \sum_{l=T-i}^{T-2} \prod_{k=0}^{l-1} \left[ \left(1 - \frac{M}{(T-k-1)p}\right)^{T-k-1} \left(1 - \frac{n}{p}\right) \right] \left(1 - \frac{M}{(T-l-1)p}\right)^{T-l-2} \frac{M}{(T-l-1)p}
\end{aligned}
$$

$$
-\sum_{l=0}^{i-2}\prod_{k=0}^{l-1}\left[\left(1-\frac{M}{(i-k-1)p}\right)^{i-k-1}\left(1-\frac{n}{p}\right)\right]\left(1-\frac{M}{(i-l-1)p}\right)^{i-l-2}\frac{M}{(i-l-1)p}
$$

$$
=\sum_{l=0}^{T-i-1}\prod_{k=0}^{l-1}\left[\left(1-\frac{M}{(T-k-1)p}\right)^{T-k-1}\left(1-\frac{n}{p}\right)\right]\left(1-\frac{M}{(T-l-1)p}\right)^{T-l-2}\frac{M}{(T-l-1)p}
$$

$$
+\sum_{l=0}^{i-2}\prod_{k=0}^{l-i+T-1}\left[\left(1-\frac{M}{(T-k-1)p}\right)^{T-k-1}\left(1-\frac{n}{p}\right)\right]\left(1-\frac{M}{(i-l-1)p}\right)^{i-l-2}\frac{M}{(i-l-1)p}
$$

$$
-\sum_{l=0}^{i-2}\prod_{k=0}^{l-1}\left[\left(1-\frac{M}{(i-k-1)p}\right)^{i-k-1}\left(1-\frac{n}{p}\right)\right]\left(1-\frac{M}{(i-l-1)p}\right)^{i-l-2}\frac{M}{(i-l-1)p}
$$

$$
\overset{(i)}{<}\sum_{l=0}^{T-i-1}\left(1-\frac{n+M}{p}\right)^{l}\frac{M}{(T-l-1)p}-\frac{M}{T^2p^2}+\frac{T^2(n+M)M^2}{p^3}
$$

$$
+\sum_{l=0}^{i-2}\left[\left(1-\frac{n+M}{p}+\frac{(n+M)M}{p^2}\right)^{l-i+T}-\left(1-\frac{n+M}{p}\right)^{l}\right]\left(1-\frac{M}{(i-l-1)p}\right)^{i-l-2}\frac{M}{(i-l-1)p}
$$

$$
\overset{(ii)}{<}\sum_{l=0}^{T-i-1}\left(1-\frac{n+M}{p}\right)^{l}\frac{M}{(T-l-1)p}-\frac{M}{T^2p^2}+\frac{T^2(n+M)M^2}{p^3}
$$

$$
+\sum_{l=0}^{i-2}\left[\left(1-\frac{n+M}{p}\right)^{l-i+T}+\frac{T^2(n+M)M}{p^2}-\left(1-\frac{n+M}{p}\right)^{l}\right]\frac{M}{(i-l-1)p}
$$

$$
<\sum_{l=0}^{T-1}\left(1-\frac{n+M}{p}\right)^{l}\frac{M}{(T-l-1)p}-\sum_{l=0}^{i-1}\left(1-\frac{n+M}{p}\right)^{l}\frac{M}{(i-l-1)p}-\frac{M}{T^2p^2}+\frac{2T^2(n+M)M^2}{p^3}, \tag{57}
$$

where $(i)$ follows from Equation (45) and Lemmas B.11 and B.12, $(ii)$ follows from Lemma B.13. By combining Equations (56) and (57),

$$
c_{i1i}^{(\text{sequential})}<\frac{n}{p}\left[\left(1-\frac{n+M}{p}\right)^{T-1}-\left(1-\frac{n+M}{p}\right)^{i-1}\right]+\sum_{l=0}^{T-1}\left(1-\frac{n+M}{p}\right)^{l}\frac{M}{(T-l-1)p}
$$

$$
-\sum_{l=0}^{i-1}\left(1-\frac{n+M}{p}\right)^{l}\frac{M}{(i-l-1)p}+\frac{T^2(n+M)nM}{p^3}-\frac{M}{T^2p^2}+\frac{2T^2(n+M)M^2}{p^3} \tag{58}
$$

$$
\overset{(i)}{<}\frac{n}{p}\left[\left(1-\frac{n+M}{p}\right)^{T-1}-\left(1-\frac{n+M}{p}\right)^{i-1}\right]+\sum_{l=0}^{T-1}\left(1-\frac{n+M}{p}\right)^{l}\frac{M}{(T-l-1)p}
$$

$$
-\sum_{l=0}^{i-1}\left(1-\frac{n+M}{p}\right)^{l}\frac{M}{(i-l-1)p}
$$

$$
\overset{(ii)}{\leq}c_{i1i}^{(\text{concurrent})} \tag{59}
$$

where $(i)$ follows from the fact that $p>3T^4(n+M)nM$, $(ii)$ follows from our observation in Equations (51) to (55).

2. Next, we prove $c_{iji}^{(\text{concurrent})}>c_{iji}^{(\text{sequential})}$ if $p>3T^4(n+M)nM$, for $j=2,3,...,i-1$. We observe that $c_{iji}^{(\text{sequential})}$ consists of two parts, where the first part can be upper bounded by

$$
\sum_{l=0}^{T-j-1}\prod_{k=0}^{l-1}\left[\left(1-\frac{M}{(T-k-1)p}\right)^{T-k-1}\left(1-\frac{n}{p}\right)\right]\left(1-\frac{M}{(T-l-1)p}\right)^{T-j-l-1}\frac{M}{(T-l-1)p}
$$

$$
-\sum_{l=0}^{i-j-1}\prod_{k=0}^{l-1}\left[\left(1-\frac{M}{(i-k-1)p}\right)^{i-k-1}\left(1-\frac{n}{p}\right)\right]\left(1-\frac{M}{(i-l-1)p}\right)^{i-j-l-1}\frac{M}{(i-l-1)p}
$$

$$\overset{(i)}{<} \sum_{l=0}^{T-j-1} \left(1 - \frac{n+M}{p}\right)^l \frac{M}{(T-l-1)p} - \sum_{l=0}^{i-j-1} \left(1 - \frac{n+M}{p}\right)^l \frac{M}{(i-l-1)p} - \frac{M}{T^2 p^2} + \frac{2T^2(n+M)M^2}{p^3}, \quad (60)$$

The other part of $c_{iji}^{(\text{sequential})}$ can be upper bounded by:

$$
\prod_{k=0}^{T-j-1} \left[ \left(1 - \frac{M}{(T-k-1)p}\right)^{T-k-1} \left(1 - \frac{n}{p}\right) \right] \left(1 - \frac{M}{(j-1)p}\right)^{j-1} \frac{n}{p}
$$
$$
- \prod_{k=0}^{i-j-1} \left[ \left(1 - \frac{M}{(i-k-1)p}\right)^{i-k-1} \left(1 - \frac{n}{p}\right) \right] \left(1 - \frac{M}{(j-1)p}\right)^{j-1} \frac{n}{p}
$$
$$
\overset{(i)}{<} \left\{ \left(1 - \frac{n+M}{p}\right)^{i-j-1} \left[ \left(1 - \frac{n+M}{p}\right)^{T-i} - 1 \right] + \frac{T^2(n+M)M}{p^2} \right\} \left(1 - \frac{M}{(j-1)p}\right)^{j-1} \frac{n}{p}
$$
$$
< \left\{ \left(1 - \frac{n+M}{p}\right)^{i-j-1} \left[ \left(1 - \frac{n+M}{p}\right)^{T-i} - 1 \right] \right\} + \frac{T^2(n+M)nM}{p^3}, \quad (61)
$$

where $(i)$ follows from Lemma B.17. By combining Equations (60) and (61), we have

$$
c_j^{(\text{sequential})} < \sum_{l=0}^{T-j-1} \left(1 - \frac{n+M}{p}\right)^l \frac{M}{(T-l-1)p} - \sum_{l=0}^{i-j-1} \left(1 - \frac{n+M}{p}\right)^l \frac{M}{(i-l-1)p}
$$
$$
+ \left\{ \left(1 - \frac{n+M}{p}\right)^{i-1} \left[ \left(1 - \frac{n+M}{p}\right)^{T-i} - 1 \right] \right\} + \frac{T^2(n+M)nM}{p^3} - \frac{M}{T^2 p^2} + \frac{2T^2(n+M)M^2}{p^3}
$$
$$
\overset{(i)}{<} \sum_{l=0}^{T-j-1} \left(1 - \frac{n+M}{p}\right)^l \frac{M}{(T-l-1)p} - \sum_{l=0}^{i-j-1} \left(1 - \frac{n+M}{p}\right)^l \frac{M}{(i-l-1)p}
$$
$$
+ \left\{ \left(1 - \frac{n+M}{p}\right)^{i-j-1} \left[ \left(1 - \frac{n+M}{p}\right)^{T-i} - 1 \right] \right\}
$$
$$
\overset{(ii)}{\leq} c_{iji}^{(\text{concurrent})}, \quad (62)
$$

where $(i)$ follows from the fact that $p > 3T^4(n+M)nM$, $(ii)$ follows from our observation in Equations (51) to (55).

3. We prove $c_{iji}^{(\text{concurrent})} > c_{iji}^{(\text{sequential})}$ for $j = i, i+1, ..., T-1$ if $p > T^4(n+M)M$. According to the same derivation as Equations (47) and (49), we have

$$
c_{iji}^{(\text{sequential})} < \sum_{l=0}^{T-j-1} \left(1 - \frac{n+M}{p}\right)^l \frac{M}{(T-l-1)p} \left(1 - \frac{n+M}{p}\right)^{T-j} \frac{n}{p} - \frac{M}{T^2 p^2} + \frac{T^2(n+M)M^2}{p^3}
$$
$$
< \sum_{l=0}^{T-j-1} \left(1 - \frac{n+M}{p}\right)^l \frac{M}{(T-l-1)p} \left(1 - \frac{n+M}{p}\right)^{T-j} \frac{n}{p}
$$
$$
\overset{(i)}{\leq} c_{iji}^{(\text{concurrent})},
$$

where $(i)$ follows from our observation in Equations (51) to (55).

4. Last, we prove $c_{iTi}^{(\text{concurrent})} > c_{iTi}^{(\text{sequential})}$ if $p > T^2(n+M)M$. Consider:

$$
c_{iTi}^{(\text{sequential})} = \left(1 - \frac{M}{(T-1)p}\right)^{T-1} \frac{n}{p} < \left(1 - \frac{M}{(T-1)p}\right) \frac{n}{p} < \frac{n}{p} - \frac{nM}{p^2}
$$
$$
\overset{(i)}{<} \frac{n}{p}
$$

$$\overset{(ii)}{\leq} c_{iTi}^{(\text{concurrent})}, \tag{63}$$

where $(i)$ follows from the fact that $p > T^2(n+M)M$, $(ii)$ follows from our observation in Equations (51) to (55).

5. As discussed in Equations (51) to (55), we have

## G. Proof of Theorem 5.5

In this section, we prove Theorem 5.5 where we provide a particular example in which sequential rehearsal has lower forgetting and generalization than concurrent rehearsal. We first prove the forgetting part in Theorem 5.5. Recall $F_T = \frac{1}{T-1} \sum_{i=1}^{T-1} \mathbb{E}(\mathcal{L}_i(\boldsymbol{w}_T) - \mathcal{L}_i(\boldsymbol{w}_i))$. Therefore, it suffices to prove

$$[\mathcal{L}_i(\boldsymbol{w}_T) - \mathcal{L}_i(\boldsymbol{w}_i)]^{(\text{concurrent})} > [\mathcal{L}_i(\boldsymbol{w}_T) - \mathcal{L}_i(\boldsymbol{w}_i)]^{(\text{sequential})}$$

if $p > 2T^2(n+M)nM$ for each $i \in [T-1]$. Since $\boldsymbol{w}_i^*$ are orthonormal, we have $\|\boldsymbol{w}_i^*\|^2 = 1$ and $\|\boldsymbol{w}_i^* - \boldsymbol{w}_j^*\|^2 = 2$ for $i \neq j$. Recall the discussion about $\beta_2$ in Equation (54). Then, we consider

$$\begin{aligned}
2\beta_2^+ &= \sum_{l=0}^{T-i-1} \left(1 - \frac{n+M}{p}\right)^l \frac{2nM}{p(p-n-M-1)} \\
&= \frac{2nM}{p(p-n-M-1)} \cdot \frac{[1 - (1 - \frac{n+M}{p})^{T-i}]}{1 - (1 - \frac{n+M}{p})} \\
&> \frac{2nM}{p^2} \cdot \frac{-\sum_{k=1}^{T-i} \binom{T-i}{k}(-\frac{n+M}{p})^k}{\frac{n+M}{p}}
\end{aligned} \tag{64}$$

We note that for any $k \in [3, T-i-1]$ and $k$ is odd, we have

$$\begin{aligned}
&\binom{T-i}{k}\left(-\frac{n+M}{p}\right)^k + \binom{T-i}{k+1}\left(-\frac{n+M}{p}\right)^{k+1} \\
&= \frac{(T-i)!}{k!(T-i-k-1)!}\left(-\frac{n+M}{p}\right)^k \left[\frac{1}{T-i-k} + \frac{1}{k+1}\left(-\frac{n+M}{p}\right)\right] \\
&< \frac{(T-i)!}{k!(T-i-k-1)!}\left(-\frac{n+M}{p}\right)^k \left[\frac{1}{T} - \frac{n+M}{p}\right] \\
&\overset{(i)}{<} 0,
\end{aligned}$$

where $(i)$ follows from the fact that $p > T(n+M)$. By simply discussing when $T-i$ is odd or even, we can have

$$\begin{aligned}
-\sum_{k=1}^{T-i} \binom{T-i}{k}\left(-\frac{n+M}{p}\right)^k &> -\binom{T-i}{1}\left(-\frac{n+M}{p}\right) - \binom{T-i}{2}\left(-\frac{n+M}{p}\right)^2 \\
&= \frac{(T-i)(n+M)}{p} - \frac{(T-i)(T-i-1)(n+M)^2}{2p^2}.
\end{aligned}$$

By substituting the above equation into Equation (64), we can have

$$\begin{aligned}
2\beta_2^+ &> \frac{2nM}{p(n+M)} \cdot \left[\frac{(T-i)(n+M)}{p} - \frac{(T-i)(T-i-1)(n+M)^2}{2p^2}\right] \\
&= \frac{2(T-i)nM}{p^2} - \frac{(T-i)(T-i-1)(n+M)nM}{p^3} \\
&\overset{(i)}{\geq} \frac{(T-i)(n+M)M}{p^2} + \frac{M}{p^2} - \frac{T^2(n+M)nM}{p^3}
\end{aligned} \tag{65}$$

where $(i)$ follows from the fact that $n \geq M + 1$. Now, we can conclude:

$$
[\mathcal{L}_i(\boldsymbol{w}_T) - \mathcal{L}_i(\boldsymbol{w}_i)]^{(\text{concurrent})}
$$

$$
= c_0^{(\text{concurrent})} + 2 \sum_{j=1}^{T} c_j^{(\text{concurrent})}
$$

$$
\overset{(i)}{>} \left(1 - \frac{n}{p}\right) \left[\left(1 - \frac{n+M}{p}\right)^{T-1} - \left(1 - \frac{n+M}{p}\right)^{i-1}\right] + 2 \sum_{j=1}^{T} c_j^{(\text{sequential})} + 2\beta_1^+ + 2\beta_2^+
$$

$$
\geq \left(1 - \frac{n}{p}\right) \left[\left(1 - \frac{n+M}{p}\right)^{T-1} - \left(1 - \frac{n+M}{p}\right)^{i-1}\right] + 2 \sum_{j=1}^{T} c_j^{(\text{sequential})} + 2\beta_2^+ \tag{66}
$$

where $(i)$ follows from Equations (58), (62) and (63). On the other hand, we have:

$$
[\mathcal{L}_i(\boldsymbol{w}_T) - \mathcal{L}_i(\boldsymbol{w}_i)]^{(\text{sequential})}
$$

$$
\overset{(i)}{<} \left(1 - \frac{n}{p}\right) \left[\left(1 - \frac{n+M}{p}\right)^{T-1} - \left(1 - \frac{n+M}{p}\right)^{i-1}\right] + 2 \sum_{j=1}^{T} c_j^{(\text{sequential})}
$$

$$
+ \frac{(T-i)(n+M)M}{p^2} + \frac{T^3(n+M)^2 M^2}{p^4}, \tag{67}
$$

where $(i)$ follows from Lemma B.18. By combining Equations (65) to (67) and the fact that $p > 2T^2(n+M)nM$, we have

$$
[\mathcal{L}_i(\boldsymbol{w}_T) - \mathcal{L}_i(\boldsymbol{w}_i)]^{(\text{concurrent})} > [\mathcal{L}_i(\boldsymbol{w}_T) - \mathcal{L}_i(\boldsymbol{w}_i)]^{(\text{sequential})},
$$

which completes the proof.

Next, we prove the generalization error part in Theorem 5.5. Recall $G_T = \frac{1}{T} \sum_{i=1}^{T} \mathbb{E}\mathcal{L}_i(\boldsymbol{w}_T)$, it suffices to prove

$$
\mathcal{L}_i^{(\text{concurrent})}(\boldsymbol{w}_T) > \mathcal{L}_i^{(\text{sequential})}(\boldsymbol{w}_T)
$$

if $p > 2T^4(n+M+1)^2 M$ for each $i \in [T]$. Since $\boldsymbol{w}_i^*$ are orthonormal, we have $\|\boldsymbol{w}_i^*\|^2 = 1$ and $\|\boldsymbol{w}_i^* - \boldsymbol{w}_j^*\|^2 = 2$ for $i \neq j$. Therefore, we have

$$
\sum_{l=0}^{t-2} \left(1 - \frac{n+M}{p}\right)^{l} \sum_{j=1}^{t-l-1} \frac{\frac{nM}{t-l-1}}{p(p-n-M-1)} \|\boldsymbol{w}_j^* - \boldsymbol{w}_{t-l}^*\|^2
$$

$$
= \sum_{l=0}^{T-2} \left(1 - \frac{n+M}{p}\right)^{l} \sum_{j=1}^{T-l-1} \frac{\frac{2nM}{T-l-1}}{p^2}
$$

$$
> (T-1) \left(1 - \frac{n+M}{p}\right)^{T} \frac{2nM}{p^2}
$$

$$
> \left(1 - \frac{T(n+M)}{p}\right) \frac{2(T-1)nM}{p^2}
$$

$$
\overset{(i)}{\geq} \left(1 - \frac{T(n+M)}{p}\right) \frac{(T-1)(n+M+1)M}{p^2}, \tag{68}
$$

where $(i)$ follows from the fact that $n \geq M + 1$. Therefore, by combining Equations (31) and (68), we have:

$$
\mathcal{L}_i^{(\text{concurrent})}(\boldsymbol{w}_T) > \left(1 - \frac{n}{p}\right) \left(1 - \frac{n+M}{p}\right)^{T-1}
$$

$$
+ 2 \left\{ \left(1 - \frac{n+M}{p}\right)^{T-1} \frac{n}{p} + \sum_{l=0}^{T-2} \left(1 - \frac{n+M}{p}\right)^{l} \frac{M}{(T-l-1)p} \right\}
$$

$$+ 2 \sum_{j=2}^{T-1} \left\{ \sum_{l=0}^{T-j-1} \left( 1 - \frac{n+M}{p} \right)^l \frac{M}{(T-l-1)p} + \left( 1 - \frac{n+M}{p} \right)^{T-j} \frac{n}{p} \right\} + \frac{2n}{p}$$

$$+ \left( 1 - \frac{T(n+M)}{p} \right) \frac{(T-1)(n+M+1)M}{p^2}. \tag{69}$$

On the other hand, we have:

$$\mathcal{L}_i^{(\text{sequential})}(\boldsymbol{w}_T) \stackrel{(i)}{<} \left( 1 - \frac{n}{p} \right) \left( 1 - \frac{n+M}{p} + \frac{(n+M)M}{p^2} \right)^{T-1}$$

$$+ 2 \left\{ \left( 1 - \frac{n+M}{p} \right)^{T-1} \frac{n}{p} + \sum_{l=0}^{T-2} \left( 1 - \frac{n+M}{p} \right)^l \frac{M}{(T-l-1)p} \right\}$$

$$+ 2 \sum_{j=2}^{T-1} \left\{ \sum_{l=0}^{T-j-1} \left( 1 - \frac{n+M}{p} \right)^l \frac{M}{(T-l-1)p} + \left( 1 - \frac{n+M}{p} \right)^{T-j} \frac{n}{p} \right\} + \frac{2n}{p}$$

$$\stackrel{(ii)}{<} \left( 1 - \frac{n}{p} \right) \left( 1 - \frac{n+M}{p} \right)^{T-1}$$

$$+ 2 \left\{ \left( 1 - \frac{n+M}{p} \right)^{T-1} \frac{n}{p} + \sum_{l=0}^{T-2} \left( 1 - \frac{n+M}{p} \right)^l \frac{M}{(T-l-1)p} \right\}$$

$$+ 2 \sum_{j=2}^{T-1} \left\{ \sum_{l=0}^{T-j-1} \left( 1 - \frac{n+M}{p} \right)^l \frac{M}{(T-l-1)p} + \left( 1 - \frac{n+M}{p} \right)^{T-j} \frac{n}{p} \right\} + \frac{2n}{p}$$

$$+ \left( \frac{(T-1)(n+M)M}{p^2} + \frac{T^3(n+M)^2 M^2}{2p^4} \right) \tag{70}$$

where $(i)$ follows from Lemma B.12 and Equations (46) and (50), $(ii)$ follows from Lemma B.14 and the fact that $1 - \frac{n}{p} < 1$. To build the relationship between Equations (69) and (70), we have:

$$\left( 1 - \frac{T(n+M)}{p} \right) \frac{(T-1)(n+M+1)M}{p^2} - \left( \frac{(T-1)(n+M)M}{p^2} + \frac{T^3(n+M)^2 M^2}{2p^4} \right)$$

$$= \frac{(T-1)M}{p^2} - \frac{T(T-1)(n+M)(n+M+1)M}{p^3} - \frac{T^3(n+M)^2 M^2}{2p^4}$$

$$\stackrel{(i)}{>} 0 \tag{71}$$

where $(i)$ follows from the fact that $p > 2T^2(n+M+1)^2 M$. By combining Equations (69) to (71), we can conclude: $\mathcal{L}_i^{(\text{concurrent})}(\boldsymbol{w}_T) > \mathcal{L}_i^{(\text{sequential})}(\boldsymbol{w}_T)$.

## H. Proof of Hybrid Rehearsal in Proposition H.1

First of all, recall that the memory set $\mathcal{M}_t = \mathcal{M}_t^{\text{sim}} \bigcup \mathcal{M}_t^{\text{dis}}$, where $\mathcal{M}_t^{\text{sim}} = \bigcup_{h \in \mathcal{I}_t^{(\text{sim})}} \mathcal{M}_{t,h}$ and $\mathcal{M}_t^{\text{dis}} = \bigcup_{h \in \mathcal{I}_t^{(\text{dis})}} \mathcal{M}_{t,h}$. We present the explicit expressions of coefficients and the noise term in Theorem 5.1 for hybrid rehearsal method in the following proposition.

**Proposition H.1.** *Under the problem setups considered in this work, the coefficients that express the expected value of the forgetting $F_t$ and generalization error $G_t$ obtained by Algorithm 1 take the following forms.*

$$d_{0t}^{(hybrid)} = r_0 \Gamma_t(t-1), \quad c_i^{(hybrid)} = d_{0T}^{(hybrid)} - d_{0i}^{(hybrid)}$$

$$d_{ijkt}^{(hybrid)} = \begin{cases} \sum_{l=0}^{t-j-1}(1-B_{l,t})^{|\mathcal{I}_{t-l}^{(dis)}|}\Gamma_t(l)B_{l,t}\mathbf{1}_{t-l,j}^{(sim)} + \sum_{l=0}^{t-2}(1-B_{l,t})^{|\mathcal{I}_{t-l}^{(dis)}|}\Gamma_t(l)pB_{l,t}K_{l,t}\mathbf{1}_{t-l,j}^{(sim)}\mathbf{1}_{t-l,k}^{(sim)} \\ \quad +(1-B_{t-k,t})^{|\mathcal{I}_k^{(dis)}|}n\Gamma_t(t-k)K_{t-k,t}\mathbf{1}_{k,j}^{(sim)} + (1-B_{t-j,t})^{|\mathcal{I}_j^{(dis)}|}\Gamma_t(t-j)(1-r_0) \\ \quad + \sum_{l=0}^{t-j-1}\Gamma_t(l)(1-B_{l,t})^{|\mathcal{I}_{t-l}^{(dis)}|-f_{t-l}(j)}B_{l,t}\mathbf{1}_{\{j\in\mathcal{I}_{t-l}^{(dis)}\}} & if \quad j\in[t-1], k=i \\ (1-r_0) + n\Gamma_t(t-k)K_{t-k,t}\mathbf{1}_{\{i\in\mathcal{I}_t^{(sim)}\}} & if \quad j=t, k=i \\ \sum_{l=0}^{t-2}(1-B_{l,t})^{|\mathcal{I}_{t-l}^{(dis)}|}\Gamma_t(l)pB_{l,t}K_{l,t}\mathbf{1}_{\{j,k\in\mathcal{I}_t^{(sim)}\}} & if \quad j<k \ and \ j,k\neq i,t \\ (1-B_{t-k,t})^{|\mathcal{I}_k^{(dis)}|}n\Gamma_t(t-k)K_{t-k,t}\mathbf{1}_{\{j\in\mathcal{I}_t^{(sim)}\}} & if \quad j<k \ and \ j,k\neq i \end{cases}$$

$$c_{ijk}^{(hybrid)} = d_{ijkT}^{(hybrid)} - d_{ijki}^{(hybrid)},$$

$$noise_t^{(hybrid)}(\sigma) = \sum_{l=0}^{t-2}\Gamma_t(l)\left(\sum_{j=1}^{|\mathcal{I}_{t-l}^{(dis)}|}(1-B_{l,t})^{|\mathcal{I}_{t-l}^{(dis)}|-f_t(j)}\Lambda_{\frac{M}{t-l-1},\sigma} + (1-B_{l,t})^{|\mathcal{I}_{t-l}^{(dis)}|}\Lambda_{n+p|\mathcal{I}_{t-l}^{(sim)}|B_{l,t},\sigma}\right)$$

where $r_a := 1-\frac{n+a}{p}$, $B_{l,t} := \begin{cases} \frac{M}{(t-l-1)p} & if\, l\neq t-1 \\ 0 & o.w. \end{cases}$, $K_{l,t} = \frac{B_{l,t}}{p-n-\frac{M}{t-l-1}|\mathcal{I}^{(sim)}|-1}$, $f_t(j) = s$ such that (the $s^{th}$ element

of $\mathcal{I}_t^{(dis)}) = j$, $\mathbf{1}_{t,j}^{(sim)} := \mathbf{1}_{\{j\in\mathcal{I}_t^{(sim)}\}}$, $\mathbf{1}_{t,j}^{(dis)} := \mathbf{1}_{\{j\in\mathcal{I}_t^{(dis)}\}}$, $\Gamma_t(a) = \prod_{l=0}^{a-1}\left[(1-B_{l,t})^{|\mathcal{I}_{t-l}^{(dis)}|}r_{\frac{M}{t-l-1}|\mathcal{I}_{t-l}^{(sim)}|}\right]$, $\Lambda_{a,\sigma} = \frac{a\sigma^2}{p-a-1}$.

In Algorithm 1, the training process of CONCURRENTTRAIN$(\mathcal{D}_t\bigcup\mathcal{M}_t^{\text{sim}})$ is equivalent to solve the following optimization problem:

$$\hat{\boldsymbol{w}}_t^{(0)} = \min_{\boldsymbol{w}}\|\boldsymbol{w}-\boldsymbol{w}_{t-1}\|^2 \quad s.t. \ \boldsymbol{X}_t^\top\boldsymbol{w} = \boldsymbol{Y}_t,$$
$$\boldsymbol{X}_{t,h}^\top\boldsymbol{w} = \boldsymbol{Y}_{t,h}, \ \ h\in\mathcal{I}_t^{(\text{sim})}. \tag{72}$$

Then, the training process of SEQUENTIALTRAIN$(\mathcal{M}_{t,h})$ for $h : \mathcal{M}_{t,h}\in\mathcal{M}_t^{\text{dis}}$ is equivalent to solve:

$$\hat{\boldsymbol{w}}_t^{(f_t(h))} = \min_{\boldsymbol{w}}\left\|\boldsymbol{w}-\hat{\boldsymbol{w}}_t^{(f_t(h)-1)}\right\|_2^2 \quad s.t. \ \boldsymbol{X}_{t,h}^\top\boldsymbol{w} = \boldsymbol{Y}_{t,h}, \ \ h=1,2,...,|\mathcal{I}_t|^{(\text{dis})}, \tag{73}$$

where the $f_t(h)^{th}$ element of $\mathcal{I}_t^{(\text{dis})}$ is $h$. The final convergent point of task $t$ is denoted as $\boldsymbol{w}_t = \hat{\boldsymbol{w}}_t^{(|\mathcal{I}_t|^{(\text{dis})})}$. By the same argument as concurrent rehearsal (or sequential rehearsal), it suffices to prove Lemma C.1 to derive the explicit expressions for both forgetting and generalization error. Consider an arbitrary $i$ s.t. $i\leq T$ and fix it. The expected value of model error $\mathbb{E}[\mathcal{L}_i(\boldsymbol{w}_t)]$ are derived as follows including concurrent part and sequential part.

The first part is concurrent part. Define $\boldsymbol{V}_t^{\text{sim}}$ as the concatenation of $\boldsymbol{X}_t$ and $\boldsymbol{X}_{t,h}$ for all $h\in\mathcal{I}_t^{(\text{sim})}$. Similarly, define $\bar{z}_t^{\text{sim}}$, as the concatenation of $\boldsymbol{z}_t$ and $\boldsymbol{z}_{t,h}$ for all $h\in\mathcal{I}_t^{(\text{sim})}$. By following the same argument as concurrent rehearsal method, we have:

$$\mathbb{E}\left\|\hat{\boldsymbol{w}}_t^{(0)}-\boldsymbol{w}_i^*\right\|^2$$

$$\overset{(i)}{=} \left(1-\frac{n+|\mathcal{I}_t^{(\text{sim})}|\frac{M}{t-1}}{p}\right)\mathbb{E}\|\boldsymbol{w}_{t-1}-\boldsymbol{w}_i^*\|^2 + \mathbb{E}\left\|\boldsymbol{V}_t^{\text{sim}\dagger}\begin{bmatrix}\cdots \\ \boldsymbol{X}_{t,h}^\top(\boldsymbol{w}_h^*-\boldsymbol{w}_i^*) \\ \cdots \\ \boldsymbol{X}_t^\top(\boldsymbol{w}_t^*-\boldsymbol{w}_i^*)\end{bmatrix}\right\|^2 + \frac{(n+|\mathcal{I}_t^{(\text{sim})}|\frac{M}{t-1})\sigma^2}{p-n-|\mathcal{I}_t^{(\text{sim})}|\frac{M}{t-1}-1},$$

$$\overset{(ii)}{=} \left(1-\frac{n+|\mathcal{I}_t^{(\text{sim})}|\frac{M}{t-1}}{p}\right)\mathbb{E}\|\boldsymbol{w}_{t-1}-\boldsymbol{w}_i^*\|^2$$

$$+ \sum_{j\in\mathcal{I}_t^{(\text{sim})}}\frac{M}{(t-1)p}\left(1+\frac{n+\frac{M}{t-1}(|\mathcal{I}_t^{(\text{sim})}|-1)}{p-n-|\mathcal{I}_t^{(\text{sim})}|\frac{M}{t-1}-1}\right)\|\boldsymbol{w}_j^*-\boldsymbol{w}_i^*\|^2$$

$$
+ \frac{n}{p}\left(1 + \frac{|\mathcal{I}_t^{(\text{sim})}|\frac{M}{t-1}}{p - n - |\mathcal{I}_t^{(\text{sim})}|\frac{M}{t-1} - 1}\right)\|\boldsymbol{w}_t^* - \boldsymbol{w}_i^*\|^2
$$

$$
+ \sum_{j,k\in\mathcal{I}_t^{(\text{sim})},\, k>j} \frac{(\frac{M}{t-1})^2}{p(p - n - |\mathcal{I}_t^{(\text{sim})}|\frac{M}{t-1} - 1)}\left(\|\boldsymbol{w}_j^* - \boldsymbol{w}_k^*\|^2 - \|\boldsymbol{w}_j^* - \boldsymbol{w}_i^*\|^2 - \|\boldsymbol{w}_k^* - \boldsymbol{w}_i^*\|^2\right)
$$

$$
+ \sum_{j\in\mathcal{I}_t^{(\text{sim})}} \frac{\frac{nM}{t-1}}{p(p - n - |\mathcal{I}_t^{(\text{sim})}|\frac{M}{t-1} - 1)}\left(\|\boldsymbol{w}_j^* - \boldsymbol{w}_t^*\|^2 - \|\boldsymbol{w}_j^* - \boldsymbol{w}_i^*\|^2 - \|\boldsymbol{w}_t^* - \boldsymbol{w}_i^*\|^2\right)
$$

$$
+ \frac{(n + |\mathcal{I}_t^{(\text{sim})}|\frac{M}{t-1})\sigma^2}{p - n - |\mathcal{I}_t^{(\text{sim})}|\frac{M}{t-1} - 1}, \tag{74}
$$

where $(i)$ follows from the same argument of Equation (29) and $(ii)$ follows from the same argument of Equation (30). The second part is sequential part. According to Lemmas B.1 to B.4, we have:

$$
\mathbb{E}\left\|\hat{\boldsymbol{w}}_t^{(f_t(h))} - \boldsymbol{w}_i^*\right\|^2 = \left(1 - \frac{M}{(t-1)p}\right)\mathbb{E}\left\|\hat{\boldsymbol{w}}_t^{(f_t(h)-1)} - \boldsymbol{w}_i^*\right\|^2 + \frac{M}{(t-1)p}\left\|\boldsymbol{w}_j^* - \boldsymbol{w}_i^*\right\|^2 + \frac{\frac{M}{(t-1)}\sigma^2}{p - \frac{M}{(t-1)} - 1}.
$$

By iterating the above equation, we have:

$$
\mathbb{E}\left\|\boldsymbol{w}_t - \boldsymbol{w}_i^*\right\|^2 = \left(1 - \frac{M}{(t-1)p}\right)^{|\mathcal{I}_t^{(\text{dis})}|}\mathbb{E}\left\|\hat{\boldsymbol{w}}_t^{(0)} - \boldsymbol{w}_i^*\right\|^2
$$

$$
+ \sum_{j\in\mathcal{I}_t^{(\text{dis})}}\left(1 - \frac{M}{(t-1)p}\right)^{|\mathcal{I}_t^{(\text{dis})}|-f_t(j)}\frac{M}{(t-1)p}\mathbb{E}\left\|\boldsymbol{w}_j^* - \boldsymbol{w}_i^*\right\|^2
$$

$$
+ \sum_{j=1}^{|\mathcal{I}_t^{(\text{dis})}|}\left(1 - \frac{M}{(t-1)p}\right)^{|\mathcal{I}_t^{(\text{dis})}|-f_t(j)}\frac{\frac{M}{t-1}\sigma^2}{p - \frac{M}{t-1} - 1}. \tag{75}
$$

By combining Equations (74) and (75) and repeating the process, we derive the expected value of model error $\mathcal{L}_i(\boldsymbol{w}_t)$ with hybrid rehearsal method as follows.

$$
\mathbb{E}\left\|\boldsymbol{w}_t - \boldsymbol{w}_i^*\right\|^2
$$

$$
= \left(1 - \frac{M}{(t-1)p}\right)^{|\mathcal{I}_t^{(\text{dis})}|}\left(1 - \frac{n + |\mathcal{I}_t^{(\text{sim})}|\frac{M}{t-1}}{p}\right)\|\boldsymbol{w}_{t-1} - \boldsymbol{w}_i^*\|^2
$$

$$
+ \left(1 - \frac{M}{(t-1)p}\right)^{|\mathcal{I}_t^{(\text{dis})}|}\sum_{j=1}^{t-1}\mathbf{1}_{t,j}^{(\text{sim})}\frac{M}{(t-1)p}\|\boldsymbol{w}_j^* - \boldsymbol{w}_i^*\|^2
$$

$$
+ \left(1 - \frac{M}{(t-1)p}\right)^{|\mathcal{I}_t^{(\text{dis})}|}\frac{n}{p}\|\boldsymbol{w}_t^* - \boldsymbol{w}_i^*\|^2
$$

$$
+ \left(1 - \frac{M}{(t-1)p}\right)^{|\mathcal{I}_t^{(\text{dis})}|}\sum_{1=j<k\leq t-1}\frac{\mathbf{1}_{t,j}^{(\text{sim})}\mathbf{1}_{t,k}^{(\text{sim})}(\frac{M}{t-1})^2}{p(p - n - |\mathcal{I}_t^{(\text{sim})}|\frac{M}{t-1} - 1)}\|\boldsymbol{w}_j^* - \boldsymbol{w}_k^*\|^2
$$

$$
+ \left(1 - \frac{M}{(t-1)p}\right)^{|\mathcal{I}_t^{(\text{dis})}|}\sum_{j=1}^{t-1}\frac{\mathbf{1}_{t,j}^{(\text{sim})}\frac{nM}{t-1}}{p(p - n - |\mathcal{I}_t^{(\text{sim})}|\frac{M}{t-1} - 1)}\|\boldsymbol{w}_j^* - \boldsymbol{w}_t^*\|^2
$$

$$
+ \sum_{j=1}^{t-1}\mathbf{1}_{t,j}^{(\text{dis})}\left(1 - \frac{M}{(t-1)p}\right)^{|\mathcal{I}_t^{(\text{dis})}|-f_t(j)}\frac{M}{(t-1)p}\|\boldsymbol{w}_j^* - \boldsymbol{w}_i^*\|^2
$$

$$
+ \left(1 - \frac{M}{(t-1)p}\right)^{|\mathcal{I}_t^{(\text{dis})}|}\frac{(n + |\mathcal{I}_t^{(\text{sim})}|\frac{M}{t-1})\sigma^2}{p - n - |\mathcal{I}_t^{(\text{sim})}|\frac{M}{t-1} - 1} + \sum_{j=1}^{|\mathcal{I}_t^{(\text{dis})}|}\left(1 - \frac{M}{(t-1)p}\right)^{|\mathcal{I}_t^{(\text{dis})}|-f_t(j)}\frac{\frac{M}{t-1}\sigma^2}{p - \frac{M}{t-1} - 1}
$$

$$\overset{(i)}{=} \left( \prod_{h=0}^{t-2} \left( 1 - \frac{M}{(t-h-1)p} \right)^{|\mathcal{I}_{t-h}^{(\mathrm{dis})}|} \left( 1 - \frac{n + |\mathcal{I}_{t-h}^{(\mathrm{sim})}|\frac{M}{t-h-1}}{p} \right) \right) \left( 1 - \frac{n}{p} \right) \|\boldsymbol{w}_i^*\|^2$$

$$+ \sum_{j=1}^{t-1} \left[ \sum_{l=0}^{t-j-1} \left( 1 - \frac{M}{(t-l-1)p} \right)^{|\mathcal{I}_{t-l}^{(\mathrm{dis})}|} \prod_{h=0}^{l-1} \left( 1 - \frac{M}{(t-h-1)p} \right)^{|\mathcal{I}_{t-h}^{(\mathrm{dis})}|} \left( 1 - \frac{n + |\mathcal{I}_{t-h}^{(\mathrm{sim})}|\frac{M}{t-h-1}}{p} \right) \frac{\mathbf{1}_{t-l,j}^{(\mathrm{sim})}M}{(t-l-1)p} \right.$$

$$+ \left( 1 - \frac{M}{(j + \mathbf{1}_{\{j=1\}} - 1)p} \right)^{|\mathcal{I}_j^{(\mathrm{dis})}|} \prod_{h=0}^{t-j-1} \left( 1 - \frac{M}{(t-h-1)p} \right)^{|\mathcal{I}_{t-h}^{(\mathrm{dis})}|} \left( 1 - \frac{n + |\mathcal{I}_{t-h}^{(\mathrm{sim})}|\frac{M}{t-h-1}}{p} \right) \frac{n}{p}$$

$$+ \sum_{j=1}^{t-1} \sum_{l=0}^{t-j-1} \prod_{h=0}^{l-1} \left( 1 - \frac{M}{(t-h-1)p} \right)^{|\mathcal{I}_{t-h}^{(\mathrm{dis})}|} \left( 1 - \frac{n + |\mathcal{I}_{t-h}^{(\mathrm{sim})}|\frac{M}{t-h-1}}{p} \right) \left( 1 - \frac{M}{(t-l-1)p} \right)^{|\mathcal{I}_{t-l}^{(\mathrm{dis})}|-f_{t-l}(j)} \left. \frac{\mathbf{1}_{t-l,j}^{(\mathrm{dis})}M}{(t-l-1)p} \right]$$

$$\cdot \left\| \boldsymbol{w}_j^* - \boldsymbol{w}_i^* \right\|^2$$

$$+ \sum_{l=0}^{t-2} \prod_{h=0}^{l-1} \left( 1 - \frac{M}{(t-h-1)p} \right)^{|\mathcal{I}_{t-h}^{(\mathrm{dis})}|} \left( 1 - \frac{n + |\mathcal{I}_{t-h}^{(\mathrm{sim})}|\frac{M}{t-h-1}}{p} \right) \left\{ \sum_{\substack{j=1\\k>j}}^{t-l-1} \frac{\mathbf{1}_{t-l,j}^{(\mathrm{sim})}\mathbf{1}_{t-l,k}^{(\mathrm{sim})}(\frac{M}{t-l-1})^2}{p(p - n - |\mathcal{I}_{t-l}^{(\mathrm{sim})}|\frac{M}{t-l-1} - 1)} \left\| \boldsymbol{w}_j^* - \boldsymbol{w}_k^* \right\|^2 \right.$$

$$+ \left( 1 - \frac{M}{(t-l-1)p} \right)^{|\mathcal{I}_t^{(\mathrm{dis})}|} \sum_{j=1}^{t-l-1} \frac{\mathbf{1}_{t-l,j}^{(\mathrm{sim})}\frac{nM}{t-l-1}}{p(p - n - |\mathcal{I}_{t-l}^{(\mathrm{sim})}|\frac{M}{t-l-1} - 1)} \left. \left\| \boldsymbol{w}_j^* - \boldsymbol{w}_{t-l}^* \right\|^2 \right\}$$

$$+ \mathrm{noise}_t^{(\mathrm{hybrid})}(\sigma)$$

where $(i)$ follows from the iteration and Equation (28) and

$$\mathrm{noise}_t^{(\mathrm{hybrid})}(\sigma) = \sum_{l=0}^{t-2} \prod_{h=0}^{l-1} \left( 1 - \frac{M}{(t-h-1)p} \right)^{|\mathcal{I}_{t-h}^{(\mathrm{dis})}|} \left( 1 - \frac{n + |\mathcal{I}_{t-h}^{(\mathrm{sim})}|\frac{M}{t-h-1}}{p} \right)$$

$$\cdot \left[ \left( 1 - \frac{M}{(t-l-1)p} \right)^{|\mathcal{I}_{t-l}^{(\mathrm{dis})}|} \frac{(n + |\mathcal{I}_{t-l}^{(\mathrm{sim})}|\frac{M}{t-l-1})\sigma^2}{p - n - |\mathcal{I}_{t-l}^{(\mathrm{sim})}|\frac{M}{t-l-1} - 1} + \sum_{j=1}^{|\mathcal{I}_{t-l}^{(\mathrm{dis})}|} \left( 1 - \frac{M}{(t-l-1)p} \right)^{|\mathcal{I}_{t-l}^{(\mathrm{dis})}|-f_{t-l}(j)} \frac{\frac{M}{t-l-1}\sigma^2}{p - \frac{M}{t-l-1} - 1} \right]$$

$$+ \prod_{h=0}^{t-2} \left( 1 - \frac{M}{(t-h-1)p} \right)^{|\mathcal{I}_{t-h}^{(\mathrm{dis})}|} \left( 1 - \frac{n + |\mathcal{I}_{t-h}^{(\mathrm{sim})}|\frac{M}{t-h-1}}{p} \right) \frac{n\sigma^2}{p - n - 1}.$$

By rearranging the terms and substituting $t = T$, we complete the poof for $d_{0T}^{(\mathrm{hybrid})}$ and $d_{ijkT}^{(\mathrm{hybrid})}$. Furthermore, the expressions of $c_i^{(\mathrm{hybrid})}$ and $c_{ijk}^{(\mathrm{hybrid})}$ in Proposition H.1 can be derived directly based on $d_{0T}^{(\mathrm{hybrid})}$ and $d_{ijkT}^{(\mathrm{hybrid})}$ and the definition of forgetting.

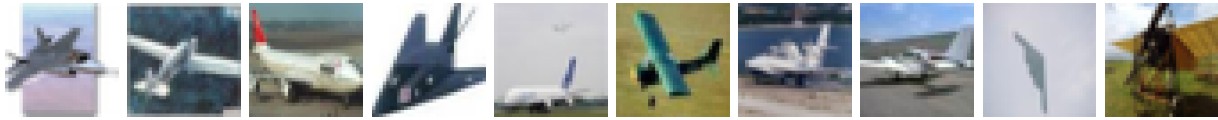

(a) Sample images without corruption.

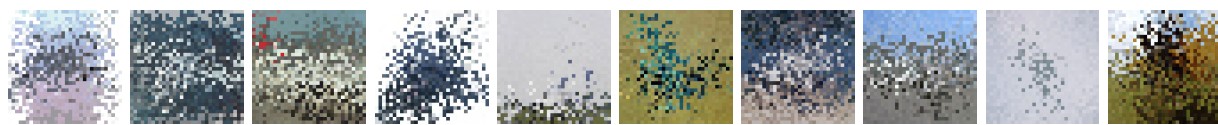

(b) Glass Corruption: the images are transformed to simulate the effect of viewing through frosted glass, inducing localized blurring and pixel displacement.

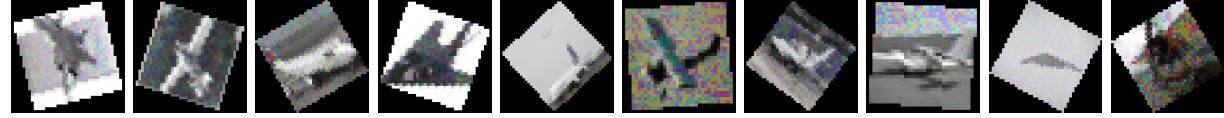

(c) Color-swapping and Rotation Corruption: the images are randomly rotated by arbitrary angles, and a subset of pixels undergoes random permutation of RGB channels.

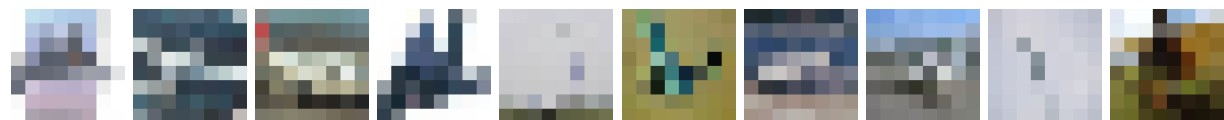

(d) Elastic and Pixelate Image Corruption: the images are subjected to smooth, non-linear spatial deformations followed by pixelation, resulting in a low-resolution appearance.

*Figure 4.* Sample images for demonstrating the employed corruption schemes.

