# OpenReview forum: "Unlocking the Power of Rehearsal in Continual Learning: A Theoretical Perspective"
_ICML.cc/2025/Conference — ICML 2025 poster_

### Official Review · Reviewer_p4ve · 2025-03-04

**Overall Recommendation:** 4

**Summary:**

This paper explores a different scheme for rehearsal training in continual learning. It also provides theoretical framework for forgetting and generalization error of concurrent and sequential rehearsal. And the authors show under conditions where the difference between the sequential tasks is large, sequential rehearsal is provably better than concurrent rehearsal. This paper includes experimental results that verify its theory.

**Claims And Evidence:**

The claims are easy to understand and with rich illustration.

**Essential References Not Discussed:**

No

**Experimental Designs Or Analyses:**

The experiments and analyses are adequate.

**Methods And Evaluation Criteria:**

About the sequential rehearsal, did you try training with the rehearsal samples for several rounds? For example, in figure 1, after training $M_{t, t-1}$, go back to train on $M_{t,1}$. What would it be?

**Other Comments Or Suggestions:**

No

**Other Strengths And Weaknesses:**

No

**Questions For Authors:**

Are there any risk of overfitting in sequential rehearsal?

**Relation To Broader Scientific Literature:**

No

**Theoretical Claims:**

The theoretical claims are supported with abundant proof.

---

> ### Author Rebuttal · Authors · 2025-04-01
>
> We thank the reviewer for providing the valuable review. Please note that all our new experiment results (i.e., the tables we quote below) and our codes can be accessed via the link https://anonymous.4open.science/r/repo-c14014
>
>
> **Q:** Are there any risks of overfitting in sequential rehearsal?
>
> **A:** We appreciate the reviewer’s question. During training, the risk of overfitting indeed exists due to the limited size of memory data. To reduce the overfitting risk, our experiment has adopted a more conservative learning rate if a ‘dissimilar task’ is learned separately, as presented in Table 3 of the paper.
>
> **Q:** About the sequential rehearsal, did you try training with the rehearsal samples for several rounds? For example, in figure 1, after training $M_{t,t-1}$, go back to train on $M_{t,1}$.  What would it be?
>
> **A:** We appreciate the reviewer’s insightful question. Despite the limited time, we made every effort to expand our experiments to train dissimilar tasks for multiple rounds, as presented in Table R3 in the anonymous link. The Averaged Final Accuracy(Acc) of multiple-round training is not changed significantly, while the Forgetting(Fgt) is slightly improved compared to single-round training. It could be because the same memory data is repeatedly learned when sequential training is conducted over multiple rounds, which may help recover more previous knowledge while introducing an unavoidable risk of overfitting.

---

### Official Review · Reviewer_GGhR · 2025-03-12

**Overall Recommendation:** 2

**Summary:**

In the context of Continual Learning, the authors propose a new rehearsal method, which is sequential. Then the authors present a theoretical analysis of both sequential and concurrent rehearsal methods. The authors derive a closed form expression of generalisation and forgetting for both methods. The main takeaway is that sequential rehearsal outperforms concurrent rehearsal when tasks are dissimilar. Based on these findings, they propose a hybrid rehearsal CL algorithm. For every new task, the memory dataset is split into similar and dissimilar samples. The similar samples are merged with the current task and the dissimilar points are finetuned afterwards.

**Claims And Evidence:**

- "our hybrid approach can perform better than concurrent rehearsal and the advantage is more apparent when tasks are more dissimilar"
One question I have is whether the samples are seen the same number of times in the final benchmarks, when comparing the hybrid vs the concurrent rehearsal. (i.e. Are the experiments set with training steps or epochs as a hyperparameter ?)

The following two claims are supported theoretically :
- The first explicit closed-form expressions for the expected value of forgetting and generalization error for both concurrent rehearsal strategy and sequential rehearsal strategy under an overparameterized linear regression setting.
- Sequential rehearsal outperforms concurrent rehearsal if tasks in CL are dissimilar, and the performance improvement is larger when the tasks are more dissimilar.

**Essential References Not Discussed:**

The proposed sequential rehearsal method may share some similarities with other curriculum learning methods. I am not familiar with the literature, but It would be worth mentioning the related works in Curriculum Learning.

**Experimental Designs Or Analyses:**

I reviewed all the experimental designs and analyses in the main paper.

Some questions below :
- Are the samples are seen the same number of times in the final benchmarks, when comparing the hybrid vs the concurrent rehearsal. (i.e. Are the experiments set with training steps or epochs as a hyperparameter ?)
- In Table 2, could you add the std to conclude on the significance of the improvement ?
- Is the method effective if the corruption is applied to the labels ?
- L 370 : Actually I think that the task ordering for the buffer division is within scope and it would be valuable to see the improvement wrt the number of splits and the ordering of the dissimilar samples. Especially since that the improvements are not very significant in the current minimal setup. It would be valuable to see if it's because of this simplification, or if it's still the case for a more optimised buffer division and ordering.
- Figure 2 : Could you clarify the unit of the y axis ?

**Methods And Evaluation Criteria:**

The benchmarks and metrics are the standard ones used across most CL literature.

**Other Comments Or Suggestions:**

Suggestions :
- Theorem 5.1 : I think it would add clarity to share an intuition about the coefficients in the interpretation below the theorem.
- L 223 to 237 : "By letting M = 0", "When p→∞" both quantities don't appear in the form of the theorem presented in the main paper.

**Other Strengths And Weaknesses:**

Strengths :
- Clarity of the presentation and ease to follow the paper
- Clear motivation and positioning wrt the literature
- Informative analysis which investigates rehearsal methods which are widely used
- Clear experiments and results

Weaknesses :
- The linear model is very simplistic and it's unclear to which extends it transfers to more complex data.
- The results look promising on the small scale MNIST and CIFAR-10 datasets, but the improvement is marginal on the larger CIFAR-100 benchmark.
- The proposed method is evaluated with a single split of the memory buffer. It is unclear to which extend the improvement evolves as a function of the number of splits.
- Other comments in the other sections.

**Questions For Authors:**

- Could the proposed method outperform the multitask learning baseline, provided the permutations of data are optimised and all the dataset is accessible at once ?
- In Theorem 5.1, it look like the only way the task ordering may influence forgetting is through the coefficients. Could you share an intuition about how the task ordering impacts forgetting from the Theorem ?
- Table 1 : On CIFAR-100, the improvement in CF is marginal compared to MNIST and CIFAR-10. Could it be because the data and model are more complex and further from the theoretical assumptions than the other two benchmarks ?

**Relation To Broader Scientific Literature:**

This work relates to the theoretical Continual Learning literature. Several works quantify the impact of task similarity on CF under different task, model and data assumptions : [2], [3], [4], [5], [6].

Another set of related works enforce adapt training using rehearsal samples, in order to avoid interference : [8], [9].

Additionally, the closest work to this paper is [1] which investigates rehearsal-based CL in linear models with concurrent rehearsal.

- [1] Banayeeanzade, Mohammadamin et al. “Theoretical Insights into Overparameterized Models in Multi-Task and Replay-Based Continual Learning.” ArXiv abs/2408.16939 (2024): n. pag.
- [2] Bennani, Mehdi et al. “Generalisation Guarantees for Continual Learning with Orthogonal Gradient Descent.” ArXiv abs/2006.11942 (2020): n. Pag.
- [3] Doan, Thang Van et al. “A Theoretical Analysis of Catastrophic Forgetting through the NTK Overlap Matrix.” International Conference on Artificial Intelligence and Statistics (2020).
- [4] Lee, Sebastian et al. “Continual Learning in the Teacher-Student Setup: Impact of Task Similarity.” International Conference on Machine Learning (2021).
- [5] Evron, Itay et al. “How catastrophic can catastrophic forgetting be in linear regression?” ArXiv abs/2205.09588 (2022): n. Pag.
- [6] Evron, Itay et al. “The Joint Effect of Task Similarity and Overparameterization on Catastrophic Forgetting - An Analytical Model.” ArXiv abs/2401.12617 (2024): n. pag.
- [7] Hiratani, N. (2024). Disentangling and Mitigating the Impact of Task Similarity for Continual Learning. ArXiv, abs/2405.20236.
- [8] Chaudhry, A., Ranzato, M., Rohrbach, M., & Elhoseiny, M. (2018). Efficient Lifelong Learning with A-GEM. ArXiv, abs/1812.00420.
- [9] Lopez-Paz, D., & Ranzato, M. (2017). Gradient Episodic Memory for Continual Learning. Neural Information Processing Systems.

**Theoretical Claims:**

I didn't check the proofs for the theoretical claims.

- Could you clarify the optimisation objective in L 186, is it equivalent to argmin (X \omega - y ) ** 2 plus some regularisation on the weights ?

---

> ### Author Rebuttal · Authors · 2025-04-01
>
> Please note that all our new experiment results (i.e., the tables we quote below) and our codes can be accessed via the link
> https://anonymous.4open.science/r/repo-c14014
>
> **==Questions about Experiments==**
>
> **Effectiveness of the method under label corruption:** We conduct experiments under label corruption (see Table R2 in the anonymous link). Under varying levels of label corruption, our hybrid rehearsal also outperforms concurrent rehearsal, demonstrating the effectiveness of our method under label corruption.
>
> **Experiments on Tiny-Imagenet200:** Results are in Table R1 via the anonymous link. For this more complex dataset, hybrid rehearsal substantially outperforms concurrent replay. On the original dataset, hybrid rehearsal improves accuracy by 2.19% and reduces forgetting by 3.66%. On the corrupted dataset, it improves accuracy by 2.38% and reduces forgetting by 13.32%.
>
> **Add std in Table 2:** We marked standard deviation in Table R4 via the anonymous link. As can be observed, the improvement of most results are well beyond the error bars.
>
> **Impact of number of memory splits and ordering of dissimilar samples:** Thanks for the suggestion. We are currently exploring these issues with full efforts.
>
> **Improvement of CIFAR-100 is marginal compared to MNIST and CIFAR-10:** This may not be due to the more complex dataset, as our new experiment on Tiny-Imagenet200 shows significant improvement. It could be because the random selection of classes in our experiments did not have high dissimilarity among tasks.
>
> **Other questions:** In our experiments, each sample is used the **same** number of times in both hybrid and concurrent rehearsal for a fair comparison. The unit of y-axis is a scalar of 1.
>
> **==Questions about Other Issues==**
>
> **Whether objective in L186 is equivalent to $\text{argmin} (Xw - y )^2$ plus some regularization:** The optimization objective in Line 186 comes from the convergence point of SGD on the linear model with the starting point of $w_{t-1}$ **without any regularization** [2,3,5]. This optimization is not equivalent to $\text{argmin}_w (X^\top w-y)^2+\lambda \|w\|$ because we force $X^\top w-y=0$ (Line 188) while $\text{argmin}_w (X^\top w-y)^2+\lambda \|w\|$ usually leads to a non-zero $(X^\top w-y)^2$ due to the need of balance between $(X^\top w-y)^2$ and $\|w\|$.
>
> **Extension of linear model to more complex data:** A natural next step is to extend our analysis to neural networks in the NTK regime, which is effectively a linearized model. Further, our analysis of rehearsal-based CL can also integrate with recent advances in analyzing over-parameterized neural networks and attention models, to study CL under these more complex models.
>
> **Comparison to multitask learning:** Previous studies [1,4] have shown that conventional concurrent rehearsal outperforms multitask learning. We expect our method to outperform the multitask learning baseline, as it subsumes concurrent replay as a special case.
>
> **Intuition about coefficients in Theorem 5.1:** The coefficients in Theorem 5.1, which are given in (32) and (35) in appendix, suggest the following intuitions about how task ordering impacts forgetting. When memory size $M$ is small, introducing dissimilar tasks early reduces forgetting by encouraging broader feature exploration, aligning with [3] (which has $M=0$). As $M$ increases, delaying dissimilar tasks helps, since early introduction leads to frequent rehearsal, and their conflicting nature can disrupt learning of later tasks.
>
> **L 223 to 237:** $M$ and $p$ affect the theorem via the coefficients $c_i$, $c_{ijk}$, $d_{0T}$ and $d_{ijkT}$ (see Proposition B.2 in appendix).
>
> **Similarity of sequential rehearsal with Curriculum Learning:** Although they appear similar, their underlying training objectives differ significantly. In **curriculum learning**, the model is presented with data in a structured progression, typically organized by increasing difficulty, with the goal of optimizing an overall learning performance. In contrast, **sequential rehearsal** involves data drawn from different (possibly conflicting) tasks, where the model is trained by rehearsing over these tasks with a goal of mitigating forgetting. The forgetting issue is not addressed in curriculum learning.
>
> Thank you for your insightful comments. We hope our responses addressed your concerns and would greatly appreciate your kind consideration in increasing your score.
>
> Reference:
>
> [1] Goodfellow, et al. "An empirical investigation of catastrophic forgetting in gradient-based neural networks." arXiv.
>
> [2] Gunasekar, et al. "Characterizing Implicit Bias in Terms of Optimization Geometry." ICML 2018.
>
> [3] Lin, et al. "Theory on forgetting and generalization of continual learning." ICML 2023.
>
> [4] Wu, Zihao, et al. "Is multi-task learning an upper bound for continual learning?" ICASSP 2023
>
> [5] Zhang, et al. "Understanding deep learning requires rethinking generalization." arXiv 2016.

---

> > ### Comment · Reviewer_GGhR · 2025-04-07
> >
> > Apologies for my late response, I appreciate your time and effort running the additional experiments and clarifying my questions.
> >
> > Some follow-up comments :
> > - In Table R1, I find it susprising that the improvement on AAC is +2.38, while the improvement in forgetting is -13. Does it imply that Hybrid rehearsal somewhat leads to lower accuracies overall ? How could it be explained ?
> > - Sorry, where does [1] show that conventional concurrent rehearsal outperforms multitask learning ?
> > - L 223 to 237: I think that it would be helpful to introduce M and p in the main paper because they are stated in it.
> >
> > Many thanks !

---

> > > ### Author Response · Authors · 2025-04-08
> > >
> > > We thank the reviewer very much for providing further comments.
> > >
> > > Q: In Table R1, I find it surprising that the improvement on AAC is +2.38, while the improvement in forgetting is -13. Does it imply that Hybrid rehearsal somewhat leads to lower accuracies overall? How could it be explained?
> > >
> > > A: Thank you for the insightful observation and question. Below, we first clarify our definitions of Acc and Forgetting, as well as how to interpret our results under these metrics. We then provide our response based on two possible interpretations of the reviewer’s question — we apologize for any misunderstanding, as we are not entirely certain which interpretation aligns with the reviewer’s intended meaning.
> > >
> > > We note that Final Average Accuracy (Acc) evaluates the average testing accuracy **across all tasks** after learning the last task, where a higher value is better. As presented in Table R1, our hybrid method improves Acc by 2.38%, demonstrating that the **overall** testing accuracy of hybrid rehearsal is 2.38% higher than concurrent rehearsal. We also note that forgetting evaluates the average **accuracy drop** of old tasks due to learning new tasks, where a lower value is better. The detailed definition can be found at L362 in the paper. In table R1, a 13.32% lower forgetting in hybrid rehearsal compared to concurrent rehearsal indicates that our method effectively reduces the accuracy degradation of earlier tasks.
> > >
> > > If the reviewer is asking whether Hybrid rehearsal leads to lower accuracy for current task learning compared to Concurrent rehearsal, this can be true because the large amount of current task data dominates the multi-task model training in Concurrent rehearsal, leading to a model that favors the current task. However, the overall objective of CL is to strike a right balance between model stability and plasticity, and the Final Average Accuracy(Acc) is the widely used metric in the CL community to characterize how good an algorithm can handle this balance. In order to achieve a higher Acc, Hybrid rehearsal sacrifices the performance on current task learning to review the knowledge of old tasks (via sequential replay), which can be further verified by the improvement in forgetting. In contrast, Concurrent rehearsal focuses too much on current task learning and cannot retain the knowledge of old tasks.
> > >
> > > If the reviewer is asking whether the accuracy improvement of Hybrid rehearsal can be further improved by sacrificing a certain level of forgetting (given that there is a large improvement in forgetting), this can be possible. Again, how to strike the right balance between model stability and plasticity is the fundamental challenge in CL. The design of Hybrid rehearsal is by no means the optimal scheme to achieve this. However, our purpose here is to demonstrate that Hybrid rehearsal could potentially be a better choice than the widely used Concurrent rehearsal in CL, where the focus is slightly shifted to how to remember old tasks. How to further improve the performance of Hybrid rehearsal deserves an independent and more comprehensive study, which we will explore in the future work.
> > >
> > > Q: Reference about conventional concurrent rehearsal outperforms multitask learning.
> > >
> > > A: We apologize for the mistake about provided references. In [4], they investigate the relationship between multitask learning (MTL) and continual learning (CL), concluding that CL can provide superior performance compared to MTL when tasks are conflicting (i.e., dissimilar).
> > >
> > > We thank the reviewer for suggesting the introduction of $M$ and $p$ in the main paper, and we will revise the paper accordingly.
> > >
> > > Thank you again for your insightful comments. We hope our responses have addressed your questions satisfactorily, and we would greatly appreciate your kind consideration in increasing your score.

---

### Official Review · Reviewer_ptpD · 2025-03-13

**Overall Recommendation:** 3

**Summary:**

This paper theoretically and numerically investigates the effects of concurrent and sequential rehearsal in the context of continual learning. The authors analytically derive that, in a linear regression model, sequential rehearsal leads to better performance than concurrent rehearsal when tasks are more dissimilar. To validate these findings, they propose a hybrid approach in which memories with low dissimilarity are rehearsed concurrently, while those with high dissimilarity are rehearsed sequentially. This hybrid strategy is then applied and tested within a deep neural network model across multiple tasks, showing minimal to modest improvements over a purely concurrent approach.

**Claims And Evidence:**

The authors claim that in a linear regression model, where labels are generated from a noisy teacher network, and under the mean-squared error (MSE) criterion, in the over-parameterized case (i.e., when the input dimension exceeds the number of training examples), the following holds:

1. Sequential rehearsal outperforms concurrent rehearsal when tasks are more dissimilar.
2. This principle extends to deep neural networks.

However, as I will argue below, I have concerns that the authors' performance metric may not be fully justified, and as a result, the theoretical claims may not be valid.

Additionally, throughout their proofs, the authors rely heavily on the statement: "It is known that the convergence point of stochastic gradient descent (SGD) for MSE is the feasible point closest to the initial point with respect to the \mathcal{l}_2-norm, i.e., the minimum-norm solution." Unfortunately, they do not provide a reference for this claim. To my knowledge, there is no theoretical work that confirms this result in the over-parameterized case.

As a consequence, the minimization procedures presented in lines 187 and 212 may not accurately reflect the outcome of running SGD.

**Essential References Not Discussed:**

Robins (1995) was one of the first to study catastrophic forgetting, rehearsal and pseudorehearsal and thus should be included. Similarly, Mc-Closkey & Cohen (1989) and Ratcliff (1990) where among the first to explore and describe the phenonemon of catastrophic forgetting and thus should be referenced.

In my view, it would be important to include Prabhu (2020) to provide a more balanced perspective on the statement: "A large amount of studies have been proposed to address this issue, among which rehearsal-based approaches (Rolnick et al., 2019) have demonstrated state-of-the-art performance." Including this reference would offer a more nuanced view of rehearsal-based methods and their relative performance.

Further, Lee et al. (2021) ("Continual learning in the teacher-student setup: Impact of task similarity."), is closely related to the presented work but not cited. They also study continual learning using a teacher-student paradigm as a function of similarity between teachers.

**Experimental Designs Or Analyses:**

The authors validate their analytical results numerically (albeit without label noise) in Figure 2. Interestingly, the theory and simulation appear to align perfectly. One possibility is that my concern about $\mathcal{L}_i(\mathbf{w})$ being unjustified is mistaken. Another possibility is that resampling items in the memory artificially increases the effective dimensionality of the training data, thereby shifting the model into the underparameterized regime. Or perhaps there is another explanation?

Unfortunately, the authors do not provide their simulation code, making it difficult to verify these possibilities.

In principle, the numerical setup (datasets, evaluation criteria, etc.) in Section 6. appears well-designed to test the hybrid rehearsal training framework. However, the authors state that they "adopt a straightforward relaxation: only one task with the lowest similarity characterization in memory is designated as the ‘dissimilar task.’" This raises a significant concern — if only one task is treated as dissimilar, doesn’t that essentially bypass the role of the threshold $\tau$? If $\tau$ is not properly explored or validated, Algorithm 1 is not actually been implemented and validated and it becomes unclear how the simulation results refer to the theoretical claims. This should also affect how we interpret the trend described in Table 2, which is intended to reflect the theoretical result.

**Methods And Evaluation Criteria:**

The loss over task $i$ for current parameters $w$ is evaluated as $$\mathcal{L}_i(\mathbf{w}) = ||\mathbf{w} - \mathbf{w}_i^*||.$$
However, I believe this measure is not appropriate in the underdetermined case, where the input dimension $p$ exceeds the number of current training examples $n$. In such situations, the current weight vector $\mathbf{w}$ can differ from the target weights $\mathbf{w}_i^*$ while still yielding an MSE of zero. This occurs because, in an underdetermined system, there are multiple weight configurations that can perfectly fit the data, making the comparison between $\mathbf{w}$ and $\mathbf{w}_i^*$ potentially missleading.

For example, let's assume the first task has one input $\mathbf{x} = [1, 0]$ with $\mathbf{w}_1^* = [1, 1]$ then $\mathbf{w} = [1, 0]$ yields a MSE of $0$ but $\mathcal{L}_i(\mathbf{w}) \neq 0$! In fact, $\mathcal{L}_i(\mathbf{w})$ can be arbitraily large or small.

Further, the authors state: "To simplify our theoretical analysis, we focus on the situation in which the memory data are all fresh and have not been used in previous training." However, if the input-output pairs in the memory are resampled after each task, then in the underdetermined case, this would alter the set of $\mathbf{w}_i^*$​ that perfectly solve the corresponding task.

In summary, the theoretical setup does not appear to be fully consistent with the derivations and conclusions drawn.

Numerically, the hybrid rehersal training framework leads to minimal to modest improvements over a purely concurrent approach. However, improvements are often within the margin of error and thus may be simply a result of noise.

**Other Comments Or Suggestions:**

N/A

**Other Strengths And Weaknesses:**

N/A

**Questions For Authors:**

1. Could you clarify why the choice of $\mathcal{L}_i(\mathbf{w})$, and consequently the chosen performance metrics, is justified in the overparameterized setting?
2. Do you plan to make the code (particularly for Figure 2) available (to reviewers)?
3. Could you explain why the influence of $\tau$? was not tested in your experiments?
4. Why do you think the improvements of the numerical results are minimal (often within the margin of error)?

**Relation To Broader Scientific Literature:**

N/A

**Theoretical Claims:**

All theoretical claims are based on the loss function $\mathcal{L}_i(\mathbf{w})$. However, as discussed in the Methods and Evaluation Criteria section, I have concerns about the validity of this measure. Specifically, I believe it may not be appropriate, which raises doubts about the validity of the theoretical claims that rely on it.

---

> ### Author Rebuttal · Authors · 2025-04-01
>
> We thank the reviewer for providing the valuable review. Please note that all our new experiment results (i.e., the tables we quote below) and our codes (including code for Fig. 2) can be accessed via the link
> https://anonymous.4open.science/r/repo-c14014
>
>
> **Q:** Clarify the choice of $\mathcal{L}_i(w)$, and justify the performance metric in the overparameterized setting
>
> **A:**  We first clarify that $\mathcal{L}_i(w) = (w -w^*_i)^2$ is equivalent to the test error $\mathbb{E}[y - Xw]^2$. To see this, we derive $\mathbb{E}[y - Xw]^2 = \mathbb{E}[X(w -w^*_i)]^2 + \sigma^2= \mathbb{E}[(w -w^*_i)^\top X^\top X (w -w^*_i)  ] + \sigma^2 =   (w -w^*_i)^2+ \sigma^2 $, where the last equality follows because $X$ has i.i.d. standard Gaussian entries, and $\sigma$ is noise level. Such a loss has been commonly adopted in recent theoretical studies of CL [1,2].
>
> The example given by the reviewer is not valid, because the loss function (i.e., the test error) needs to take the expected value over the distribution of $X$, not for a specific value of $X$.
>
> Further note that the model parameter $w$ depends on input data $X$, because the model is trained based on data. Thus, the performance of forgetting and generalization defined in (4) and (5) is evaluated using the **expected value** of model errors, which reflects the overall performance across a set of inputs.
>
> **Q:** Provide reference for the statement "the convergence point of SGD for MSE is the feasible point closest to the initial point in $l_2$-norm, i.e., minimum-norm solution."
>
> **A:** In [3,4], it has been mathematically shown that in overparameterized linear models, SGD/GD converges to the minimum-norm solution, i.e., the feasible point closest to the initial point with respect to $l_2$-norm. Further, such a property has been widely used to simplify the theoretical model in CL [1,2].
>
> **Q** Numerical results are within error margin?
>
> **A:** In Table 1 of the paper, the improvement of both generalization and forgetting on **corrupted** datasets (with higher level of task dissimilarity) is substantial and far more outside the error, which validates our theoretical observation. The improvement is marginal on original datasets(Split-MNIST, Split-CIFAR-10, Split-CIFAR-100) because their task dissimilarity level is not significant.
>
> We further conducted **new experiments on Tiny-Imagenet200** (see Table R1 via the anonymous link). For this more complex dataset, where task similarity is much higher, on the original dataset, hybrid rehearsal improves the Averaged Final Accuracy by 2.19% and reduces forgetting by 3.66%. On the corrupted dataset, it improves the Averaged Final Accuracy by 2.38% and reduces forgetting by 13.32%. These results clearly demonstrate the benefits of our hybrid rehearsal.
>
>
> **Q:** Will resampled input-output pairs in memory alter $w_i^*$ that perfectly solves the corresponding task?
>
> **A:** We clarify that in this paper, ​$w_i^*$ represents the ground-truth model parameters, which are **fixed** for each task. Then based on such a ground-truth $w_i^*$, data are generated by $Y_i=X^\top_iw^∗_i+z_i$. Clearly, resampling of input-output pairs will not change ground-truth parameters.
>
> **Q:** Explain influence of $\tau$. Why was it not tested in your experiments? If only one task is treated as dissimilar, doesn’t that bypass the role of $\tau$?
>
> **A:** Thank you for highlighting this point. Our experiment is a simplified implementation of Algorithm 1, where  **at most** one task is designated as the ‘dissimilar task’ for sequential rehearsal. $\tau$ still serves the role of threshold as follows. If multiple tasks have scores below $\tau$, then the most dissimilar task is set for sequential rehearsal. Otherwise, if all tasks have positive scores, then no task is chosen for sequential rehearsal. Note that this is a reasonable experiment setup in CL [5].
>
> **Further note on choice of $\tau$:** Since the task similarity score is defined to be cosine similarity between gradient of each previous task and concurrent task, positive score indicates their alignment, and negative score indicates their confliction. Hence, in our experiment, we set threshold $\tau$ to be $0$ naturally. In practice, $\tau$ can be set negative to ensure the task dissimilarity is high enough.
>
> Thank you again for your insightful comments. We hope our responses addressed your concerns and would greatly appreciate your kind consideration in increasing your score.
>
> References:
>
> [1] Evron, et al. "How catastrophic can catastrophic forgetting be in linear regression?." COLT 2022.
>
> [2] Lin, et al. "Theory on forgetting and generalization of continual learning." ICML 2023.
>
> [3] Gunasekar, et al. "Characterizing Implicit Bias in Terms of Optimization Geometry." ICML 2018.
>
> [4] Zhang, et al. "Understanding deep learning requires rethinking generalization." arXiv 2016.
>
> [5] Lin, et al. "Trgp: Trust region gradient projection for continual learning." ICLR 2022.

---

> > ### Comment · Reviewer_ptpD · 2025-04-01
> >
> > I sincerely appreciate the authors’ detailed response and their efforts in addressing the concerns I raised.
> >
> > In particular, the clarification regarding $\mathcal{L}_i$​ was very helpful. It may be beneficial to explicitly state in the manuscript that $\mathcal{L}_i$​ represents the test error over the distribution of inputs, rather than training error over a fixed set of samples.
> >
> > Additionally, I strongly recommend including references [3,4] to support the equivalence of the constraint optimisation problem and the convergence of SGD.
> >
> > Thank you for addressing my question regarding resampling data in memory. I understand that $\mathbf{w^*}$ is fixed, but I would like to clarify my concern. In the main text, you state: "we focus on the situation in which the memory data are all fresh and have not been used in previous training." Since $w_t$​ explicitly depends on both the training data and the memory data used for optimization, resampling the memory after each task means that more information about the corresponding $\mathbf{w^*}$ accumulates for older tasks compared to more recent ones. Each newly sampled underconstrained i.i.d. $\mathbf{X}$ further constrains learning (which is conceptually similar to performing SGD in a teacher-student model with an increasing number of samples). How does this assumption influence your results? I would encourage verifying numerically (as I understand that this assumption is necessary to simplify the analytical treatment) that using the same data for training and the memory does not change the conclusions of the paper (i.e., Figure 2).
> >
> > I remain of the opinion that the numerical insights from the deep-learning simulations are limited, often falling within error margins. Furthermore, the proposed algorithm, as described, does not appear to be fully implemented (ignoring $\tau$) or tested in its intended form. I suggest either revising the presentation of the algorithm accordingly or providing additional empirical validation.
> >
> > Given the clarification regarding $\mathcal{L}_i$​​, I have updated my overall recommendation accordingly.

---

> > > ### Author Response · Authors · 2025-04-02
> > >
> > > We thank the reviewer very much for the prompt response and for increasing the score.
> > >
> > > We will clarify in our paper that $\mathcal L_i$​ represents the test error over the distribution of inputs, not the training error over a fixed set of samples. We will also include references [3,4] to support the equivalence of the constraint optimisation problem and the convergence of SGD. Thank you for the suggestions.
> > >
> > > **Regarding resampling data in memory**, we appreciate the reviewer’s further explanation and we now get the point, which is quite insightful. As suggested by the reviewer, we conducted a new numerical simulation where the memory data is selected from training data of previous tasks (not the fresh resampled data). All other experiment parameters are set to be the same as in the paper (i.e., Figure 2 of the paper). As can be observed in Figure R1 via https://anonymous.4open.science/r/repo-c14014/rebuttal_table.pdf, our conclusion still holds. Namely, sequential rehearsal has smaller forgetting value and test error (and hence more advantageous) than concurrent rehearsal when task dissimilarity becomes large, and such an advantage of sequential rehearsal becomes more obvious as task dissimilarity enlarges.
> > >
> > > Regarding **insights from the deep-learning simulations**, we thank the reviewer for re-iterating this issue. The improvements within error margins primarily occur in the small-scale datasets studied in our original submission, where task dissimilarity may not be substantial enough to fully showcase the benefits of our hybrid algorithm. In our new experiment on a larger dataset **Tiny-ImageNet200**, as can be observed in Table R1 via the anonymous link: https://anonymous.4open.science/r/repo-c14014/rebuttal_table.pdf, the hybrid rehearsal method achieves an accuracy of **$63.29 (\pm 0.47) $**%, compared to **$61.10 (\pm 0.28) $**% for concurrent rehearsal. This represents a $2.19 $\% improvement, which is around three times the error margin. We hope this additional evidence helps demonstrate the potential of our approach in scenarios where task dissimilarity is more significant.
> > >
> > > We also thank the reviewer for the suggestion regarding Algorithm 1. We will clarify our implementation in the experiments and revise our presentation of the algorithm accordingly. Meanwhile, we are actively working on incorporating more dissimilar tasks based on $\tau$ in sequential rehearsal in our experiments.

---

### Official Review · Reviewer_ngv9 · 2025-03-18

**Overall Recommendation:** 3

**Summary:**

The paper studies rehearsal in continual learning for overparameterized linear models. Next to concurrent rehearsal, which is the commonly used setting, they also look into sequential rehearsal, where different task data is revisited sequentially. From the theoretical analysis, they conclude that for highly dissimilar tasks, sequential rehearsal may be better. Interestingly, they next turn this into a practical algorithm, where the most dissimilar task is learnt sequentially while the other tasks are learnt concurrently. Although the effect is minor, a positive effect is observed on standard small continual learning benchmarks such as MNIST, Cifar 10 and Cifar-100.

**Claims And Evidence:**

I did not find any problematic claims.

**Essential References Not Discussed:**

I can't think of any references that should have been discussed but are missing.

**Ethical Review Concerns:**

/

**Experimental Designs Or Analyses:**

Results with more / larger datasets (e.g. (mini)ImageNet), and under different settings could have strenghtened the paper further, for instance going beyond the task-incremental setting, to class or domain incremental ones.

**Methods And Evaluation Criteria:**

The proposed method is simple, but this is mostly a theoretical paper. The strategy for the proofs in the paper make sense.
Evaluation criteria are the ones commonly used in this context.

**Other Comments Or Suggestions:**

/

**Other Strengths And Weaknesses:**

The key finding of the paper, that sequential rehearsal can in some cases outperform concurrent rehearsal, is interesting and somewhat surprising. The paper's strength is that this is not only observed empirically, but also analyzed theoretically, albeit on an overparameterized linear model only.
Neither of the two parts of the paper (theoretical analysis, empirical study) would have been sufficient on their own, but combined I think there's sufficient evidence.
I found the paper well structured and clear.

I doubt whether the observed differences, which have only been shown in a task-incremental setting, are worth the extra complexity in a practical setting, and concurrent rehearsal may remain the go-to solution. Nevertheless, the paper shows that concurrent rehearsal may not be the optimal setting, and I think this is worth sharing.

**Questions For Authors:**

1. Did you try experiments beyond the task-incremental setting ? Is there a reason why you did not ?

**Relation To Broader Scientific Literature:**

The authors did a good job in contextualizing their work in a broader context, including both the standard continual learning literature as well as papers focusing on a more theoretical analysis.

**Theoretical Claims:**

I did not check all proofs completely, but could not find any issues in the proof outlines and other parts I checked.

---

> ### Author Rebuttal · Authors · 2025-04-01
>
> We thank the reviewer for providing the valuable review. Please note that all our new experiment results (i.e., the tables we quote below) and our codes can be accessed via the link
> https://anonymous.4open.science/r/repo-c14014
>
>
> **Q1:** Results with more / larger datasets (e.g. (mini)ImageNet), and under different settings could have strengthened the paper further.
>
> **A1:** Thank you for your constructive suggestion. Despite the limited time, we made every effort to expand our experiments to Tiny-Imagenet200, as presented in Table R1 via the anonymous link. Our experiments on Tiny-ImageNet demonstrate that hybrid rehearsal achieves a substantial improvement over conventional concurrent rehearsal. On the original dataset, hybrid rehearsal improves the Averaged Final Accuracy by 2.19% and reduces forgetting by 3.66%. On the corrupted dataset, it improves the Averaged Final Accuracy by 2.38% and reduces forgetting by 13.32%, which also aligns with our theoretical observation that benefits of our hybrid rehearsal increase as task dissimilarity rises. Furthermore, we observe that the advantage of our method is even more obvious in Tiny-Imagenet200 compared to MNIST, CIFAR-10 and CIFAR-100. This is because Tiny-Imagenet200 is a more complex dataset and exhibits a higher level of task dissimilarity, and hence benefits of our hybrid rehearsal are more salient.
>
> **Q2.**  I doubt whether the observed differences, which have only been shown in a task-incremental setting, are worth the extra complexity in a practical setting, and concurrent rehearsal may remain the go-to solution.
>
> **A2:**  We sincerely appreciate the reviewer’s insightful observation. We agree that hybrid rehearsal can introduce additional complexity in practical deployments, and that conventional concurrent rehearsal may offer greater simplicity and convenience. We also appreciate the reviewer’s recognition that one goal of our paper is to highlight, from a scientific perspective, that concurrent rehearsal, while practical, can be suboptimal in terms of performance, as we have demonstrated in task-incremental settings. Our findings suggest that alternative strategies, such as hybrid rehearsal, warrant further exploration for their potential benefits.
>
>
> We thank the reviewer again for your inspiring comments. We hope that our responses resolved your concerns. If so, we wonder if the reviewer could kindly consider to increase your score. Certainly, we are more than happy to answer your further questions.

---

### Decision · Program_Chairs · 2025-05-01

**Decision:**

Accept (poster)

**Comment:**

The paper studies the impact of rehearsal buffers in continual learning for overparametrized models. It observes an interesting phenomenon, namely that, depending on the extent of overlaps between tasks, serial rather than concurrent rehearsal is preferable: this was supported by both a theoretical analysis and experimental validation, and led to the suggestion of a new and performant continual learning algorithm.

Reviewers found these observations and the consequent algorithm interesting and worth reporting to the ICML community, while also pointing out that experimental evidence could be strengthened. In response to this, the authors included several additional experiments, including on Tiny-ImageNet200, that they are strongly encouraged to include in the final version of the paper.

The authors should also incorporate several additional comments by the reviewers, as they have stated that they will do during the author-reviewer discussions.  The issues raised by Reviewer ptpD and satisfactorily addressed during the rebuttal period suggest corresponding edits that will help clarify the test error notion used, the proper context ([3,4]) for SGD equivalence to the constrained optimization, as well as sampling. One additional edit specifically requested by Reviewer GGhR is to add breakdown similar to Fig. 4 here in Mirzadeh et al., that reports accuracy performance on a per task basis.